# SNAP-UQ: Self-supervised Next-Activation Prediction for Single-Pass Uncertainty in TinyML

**Ismail Lamaakal**[1]*, **Chaymae Yahyati**[1]*, **Khalid El Makkaoui**[1], **Ibrahim Ouahbi**[1], **Yassine Maleh**[2]

[1]Multidisciplinary Faculty of Nador, Mohammed First University, Oujda 60000, Morocco
[2]Laboratory LaSTI, ENSAK, Sultan Moulay Slimane University, Khouribga 54000, Morocco

{ismail.lamaakal,chaymae.yahyati,k.elmakkaoui,i.ouahbi}@ump.ac.ma
y.maleh@usms.ma

## Abstract

Reliable uncertainty estimation is a key missing piece for *on-device* monitoring in TinyML: microcontrollers must detect failures, distribution shift, or accuracy drops under strict flash/latency budgets, yet common uncertainty approaches (deep ensembles, MC dropout, early exits, temporal buffering) typically require multiple passes, extra branches, or state that is impractical on milliwatt hardware. This paper proposes a novel and practical method, **SNAP-UQ**, for single-pass, label-free uncertainty estimation based on *depth-wise next-activation prediction*. SNAP-UQ taps a small set of backbone layers and uses tiny `int8` heads to predict the mean and scale of the next activation from a low-rank projection of the previous one; the resulting standardized prediction error forms a depth-wise surprisal signal that is aggregated and mapped through a lightweight monotone calibrator into an actionable uncertainty score. The design introduces no temporal buffers or auxiliary exits and preserves state-free inference, while increasing deployment footprint by only a few tens of kilobytes. Across vision and audio backbones, SNAP-UQ reduces flash and latency relative to early-exit and deep-ensemble baselines (typically ∼40–60% smaller and ∼25–35% faster), with several competing methods at similar accuracy often exceeding MCU memory limits. On corrupted streams, it improves accuracy-drop event detection by multiple AUPRC points and maintains strong failure detection (AUROC ≈ 0.9) in a single forward pass. By grounding uncertainty in layer-to-layer dynamics rather than solely in output confidence, SNAP-UQ offers a novel, resource-efficient basis for robust TinyML monitoring. Our code is available at: `https://github.com/Ism-ail11/SNAP-UQ`

## 1 Introduction

TinyML models increasingly ship on battery-powered microcontrollers (MCUs) (David et al., 2021) to deliver private, low-latency perception for vision and audio (Banbury et al., 2021). Once deployed, inputs seldom match the training distribution: sensors drift, lighting and acoustics vary, and streams interleave in-distribution (ID), corrupted-in-distribution (CID), and out-of-distribution (OOD) samples (Hendrycks & Dietterich, 2019). Under such shifts, modern networks are notoriously *overconfident* (Minderer et al., 2021) even when they appear calibrated on held-out ID data (Guo et al., 2017; Ovadia et al., 2019), complicating on-device monitoring and safe fallback (Lai et al., 2018; Chen et al., 2018). Addressing this on MCUs is challenging: memory and compute budgets preclude multi-pass inference, large ensembles (Lakshminarayanan et al., 2017), or heavy feature stores.

**This work:** We introduce **SNAP-UQ** (*Self-supervised Next-Activation Prediction for single-pass Uncertainty*)[1], a label-free uncertainty mechanism tailored to MCU deployments. Instead of sampling-based uncertainty (e.g., MC Dropout (Gal & Ghahramani, 2016)) or branching with auxiliary exits

---

[1]* Equally contributed and led the project.
[1]Throughout, we use 'self-supervised' to mean a *label-free auxiliary regression* that predicts next-layer activation statistics solely from the network's own features (for uncertainty estimation), rather than self-supervised *representation learning* of the backbone.

(Teerapittayanon et al., 2017), SNAP-UQ attaches *two or three tiny heads* at chosen depths. Each head predicts the next-layer activation statistics from a low-rank projection of the previous layer; the mismatch between the realized and predicted activation yields a *depth-wise surprisal* score (Sensoy et al., 2018). Aggregating these per-layer surprisals produces a single-pass uncertainty proxy that (optionally) blends with an instantaneous confidence term. The approach requires no extra forward passes, no temporal buffers, and no architectural changes to the backbone. All arithmetic is integer-friendly: the heads are quantized to int8, covariance is diagonal (with an optional low-rank correction), and exponentials are replaced by a small look-up table for $\exp(-\frac{1}{2}\log\sigma^2)$.

**Why depth-wise surprisal?**  Confidence at the softmax often degrades late and can remain peaky under CID, whereas *inter-layer dynamics* shift earlier: features become atypical *relative to the network's own transformation* even before class posteriors flatten. SNAP-UQ explicitly models this conditional evolution $a_{\ell-1} \mapsto a_\ell$ and scores how surprising $a_\ell$ is under the head's predictive distribution. The resulting score acts as an early, label-free indicator of trouble while preserving MCU budgets. Unlike classwise Mahalanobis (Lee et al., 2018) or energy-based OOD methods (Liu et al., 2021), which compare against unconditional feature statistics or log-sum-exp energies, SNAP-UQ is *conditional-on-depth* and thus sensitive to distortions that break the mapping between layers.

**Relation to prior work:**  Post-hoc calibration improves ID confidence but generally fails under shift (Guo et al., 2017; Ovadia et al., 2019). Early-exit ensembles and TinyML variants reduce cost by reusing a backbone (Qendro et al., 2021; Ghanathe & Wilton, 2024), yet still add inference-time heads and memory bandwidth, and depend on softmax-derived signals that are brittle under CID. Sampling-based uncertainty (MC Dropout, Deep Ensembles) increases compute and flash substantially (Gal & Ghahramani, 2016; Lakshminarayanan et al., 2017). Classical OOD detectors such as ODIN/G-ODIN (Liang et al., 2020; Hsu et al., 2020) can be strong on larger backbones but transfer less reliably to ultra-compact models typical of TinyML. Beyond ensembles and stochastic sampling, several *single-pass* deterministic UQ methods have been proposed, e.g., DUQ, DDU, evidential/posterior/prior networks, and recent fixes for early-exit overconfidence, but many rely on architectural changes, specialized output layers, OOD exposure during training, or heavier heads that clash with MCU constraints (Van Amersfoort et al., 2020; Mukhoti et al., 2022; Sensoy et al., 2018; Malinin & Gales, 2018; Charpentier et al., 2020; Deng et al., 2023; Meronen et al., 2023). Tiny Deep Ensemble (Ahmed et al., 2024): forms a lightweight ensemble by sharing the entire backbone and duplicating only normalization layers (with different running stats/affine params), so multiple "members" are produced with minimal extra flash/RAM. This yields ensemble-style uncertainty (better ID/CID error and OOD separation than single heads) while keeping single-pass compute close to a vanilla model, well-suited for microcontrollers and edge accelerators. In contrast, SNAP-UQ occupies a different point: *single pass, no state, tiny heads*, with a score derived from the network's own depth-wise dynamics rather than auxiliary classifiers or repeated sampling.

**Contributions:**  SNAP-UQ introduces a self-supervised, depth-wise surprisal signal from tiny predictors attached to a few layers and trained with a lightweight auxiliary loss, yielding *single-pass* uncertainty at inference with no temporal buffers, auxiliary exits, or ensembles (see Figure 1). The aggregate surprisal is an affine transform of a depth-wise negative log-likelihood (equivalently a conditional Mahalanobis energy) and is invariant to BN-like per-channel rescaling; we also derive robust (Student-$t$/Huber) and low-rank+diag variants (Appx. H, I). We provide an MCU-ready implementation using $1\times1$ projectors with global average pooling, int8 heads, LUT-based scales for $\log\sigma^2$, and a tiny monotone mapper, adding only a few-tens of KB of flash and $\lesssim 2\%$ extra MACs. Empirically, across MNIST, CIFAR-10, TinyImageNet, and SpeechCommands on two MCU tiers, SNAP-UQ improves accuracy-drop detection under CID, is competitive or better on ID✓ — ID✗ and ID✓ — OOD failure detection, and strengthens ID calibration, while fitting on the *Small-MCU* where heavier baselines are out-of-memory.

## 2  SNAP-UQ EXPLAINED

We consider a depth-$D$ backbone that maps an input $x$ to activations $\{a_\ell\}_{\ell=0}^D$ with $a_0 = x$, final features $f(x) = a_D$, and class posteriors $p_\phi(y \mid x) = \mathrm{softmax}(g(a_D)) \in \Delta^{L-1}$. Let $\hat{y} = \arg\max_\ell p_\phi(y = \ell \mid x)$ be the predicted class, $C_\phi(x) = \max_\ell p_\phi(y = \ell \mid x)$ the maximum confidence, and $m^{\mathrm{mg}}(x) = p_\phi^{(1)}(x) - p_\phi^{(2)}(x)$ the probability margin where $p_\phi^{(1)}(x) \geq p_\phi^{(2)}(x) \geq \cdots$ are sorted class probabilities. Unlike temporal uncertainty methods that rely on repeated evaluations,

buffering, or online aggregation, SNAP-UQ constructs a label-free uncertainty signal from the network's internal *depth-wise dynamics* in a single forward pass. The core intuition is that in-distribution inputs tend to induce predictable, smooth transitions from one layer to the next, while corrupted or shifted inputs often cause atypical intermediate transformations even when the final softmax remains overconfident. SNAP-UQ operationalizes this by tapping a small set of layers, predicting each tapped activation from the previous one using a lightweight conditional model, and measuring the surprisal (negative log-likelihood) of the realized activation under that model. Because all quantities are computed during the standard forward pass, SNAP-UQ does not introduce auxiliary exits, multiple passes, or temporal state, and it remains compatible with constant-memory MCU inference.

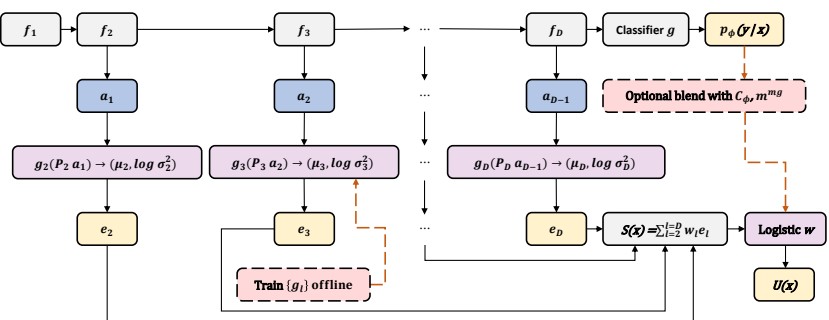

Figure 1: **SNAP-UQ pipeline.** A standard backbone $f_1, \ldots, f_D$ is tapped at a small set of layers to expose activations $a_\ell$. For each tap, a lightweight projector $P_\ell a_{\ell-1}$ feeds a tiny predictor $g_\ell$ that outputs next-activation statistics $(\mu_\ell, \log \sigma_\ell^2)$. The per-layer surprisal $e_\ell$ is a standardized squared error between the realized $a_\ell$ and $\mu_\ell$ (diagonal scale $\sigma_\ell^2$), and the single-pass uncertainty proxy aggregates these as $S(x) = \sum_\ell e_\ell$. A monotone logistic head maps $S(x)$ to a calibrated score $U(x) \in [0,1]$; optionally, the head can blend instantaneous classifier confidence signals (maximum probability $C_\phi$ and margin $m^{mg}$) for added separability. Dashed boxes indicate training-only steps where $\{g_\ell\}$ are fitted offline; inference remains one forward pass, state-free, and MCU-friendly.

SNAP-UQ is designed for milliwatt-scale devices where the uncertainty computation must be single-pass, constant-memory, and compatible with integer inference (Appx. M). We assume the backbone is fixed at deployment, intermediate activations $a_\ell$ are naturally available during the forward pass, and the device can afford a small number of additional linear operations per tapped layer together with simple elementwise arithmetic. We also assume no online labels and no long temporal histories are available, so uncertainty must come from instantaneous signals. Under these constraints, SNAP-UQ focuses on measuring whether the network's internal computation follows the feature-transition patterns it learned on in-distribution data, which provides a complementary failure signal to softmax confidence and margin.

## 2.1 Depth-wise next-activation model

We model representation evolution along depth by predicting each tapped activation from its predecessor. Let $a_\ell \in \mathbb{R}^{d_\ell}$ denote the post-activation tensor after block $f_\ell$ (flattened when convenient), and choose a small tap set $\mathcal{S} \subseteq \{2, \ldots, D\}$; in practice, two or three taps suffice, typically a mid network block and the penultimate block. This choice reflects the observation that shifts can manifest differently across the hierarchy: earlier layers may show atypical low-level statistics under corruptions, while later layers may exhibit class-conditional mismatches or feature entanglement under domain shift. By aggregating surprisal from multiple depths, SNAP-UQ captures both effects while remaining lightweight.

For each $\ell \in \mathcal{S}$, SNAP-UQ first compresses the previous activation using a low-rank projector, producing $z_\ell = P_\ell a_{\ell-1} \in \mathbb{R}^{r_\ell}$ with $r_\ell \ll d_{\ell-1}$. For convolutional backbones, $P_\ell$ is implemented as a $1 \times 1$ pointwise convolution followed by global average pooling, which keeps computation and memory low while preserving the channel-wise "state" that best predicts the next block's behavior. For MLP-like backbones, $P_\ell$ is a skinny linear layer. The role of $z_\ell$ is not to reconstruct $a_{\ell-1}$, but

to retain the predictive summary needed to forecast $a_\ell$; this makes the next-activation model both compact and robust to nuisance variability, which is important for on-device stability.

Conditioned on $z_\ell$, a tiny head $g_\ell$ outputs per-channel mean and log-variance, written as $(\mu_\ell, \log \sigma_\ell^2) = g_\ell(z_\ell)$ with $\mu_\ell, \sigma_\ell \in \mathbb{R}^{d_\ell}$. This defines a simple conditional density $p_\theta(a_\ell \mid a_{\ell-1}) = \mathcal{N}(\mu_\ell, \mathrm{diag}(\sigma_\ell^2))$. The diagonal form is deliberate: it yields a closed-form scoring rule, avoids expensive matrix operations, and supports integer-friendly inference via per-channel scales. Intuitively, $\mu_\ell$ captures the most likely next activation under typical in-distribution feature flow, while $\sigma_\ell$ represents expected variability per channel; under distribution shift, either the residual grows (unexpected mean behavior) or the residual becomes inconsistent with the learned scale (unexpected variability), both of which increase surprisal.

When additional expressivity is helpful and resources allow, we optionally refine the diagonal covariance with a low-rank correction $\Sigma_\ell = \mathrm{diag}(\sigma_\ell^2) + B_\ell B_\ell^\top$, where $B_\ell \in \mathbb{R}^{d_\ell \times k_\ell}$ and $k_\ell \ll d_\ell$. This captures a small subspace of cross-channel correlations (e.g., coupled channel groups in later layers) without paying the cost of a full covariance. The required log-determinant and quadratic form can be computed efficiently using the matrix determinant lemma and the Woodbury identity, so inference reduces to solving a $k_\ell \times k_\ell$ system per example (Appx. I); setting $k_\ell = 0$ recovers the diagonal model.

Given $(\mu_\ell, \sigma_\ell^2)$, the per-tap surprisal is computed from a standardized residual $u_\ell(x) = (a_\ell - \mu_\ell) \odot \sigma_\ell^{-1}$ and the quadratic standardized error $q_\ell(x) = \|u_\ell(x)\|_2^2$, with width-normalized $\bar{q}_\ell(x) = \frac{1}{d_\ell} q_\ell(x)$. Under the Gaussian model, the conditional NLL equals $\frac{1}{2} q_\ell(x) + \frac{1}{2} \mathbf{1}^\top \log \sigma_\ell^2$ up to an additive constant, so $q_\ell$ is the energy term that increases when the realized activation is unlikely given the preceding layer. The standardization is essential for stable deployment: it makes the score insensitive to channel-wise scaling effects (for example, due to batch normalization, calibration drift, or quantization scale choices), because the head learns $\mu_\ell$ and $\sigma_\ell$ in the same units as the activation. This yields an invariance property formalized in Prop. 2.3, and empirically reduces the need for architecture-specific tuning.

To form a single uncertainty proxy, we aggregate across taps as $S(x) = \sum_{\ell \in \mathcal{S}} w_\ell \bar{q}_\ell(x)$ with $w_\ell \geq 0$ and $\sum_\ell w_\ell = 1$. Uniform weights work well as a default, while inverse-variance weighting (estimated from a small development split) can down-weight taps whose surprisals are intrinsically noisy. This aggregation produces a scalar that summarizes whether the internal feature evolution is typical across the chosen depths; large values generally indicate that the model's computation is deviating from its learned in-distribution dynamics, which is strongly correlated with error under corruptions and OOD shifts.

The computational overhead is small because $g_\ell$ is tiny and evaluated only at a few taps. For a linear $g_\ell$ producing both $\mu_\ell$ and $\log \sigma_\ell^2$, the parameter count is approximately $2\, d_\ell r_\ell + 2 d_\ell$ (biases included) and the MAC cost is $O(d_\ell r_\ell)$. With $|\mathcal{S}| \leq 3$ and $r_\ell \in [32, 128]$, the extra FLOPs are typically below a few percent of tiny backbones, and the quantized footprint adds only tens of KB in flash. For integer deployment, we store projector and head weights as int8 (with per-tensor or per-channel scales) and use int32 accumulators; we avoid exponentials by keeping $\log \sigma_\ell^2$ and approximating $\exp(-\frac{1}{2} \log \sigma_\ell^2)$ with a small lookup table, clamping $\log \sigma_\ell^2$ to $[\log \sigma_{\min}^2, \log \sigma_{\max}^2]$ to bound dynamic range and preserve monotonicity. These design choices keep inference single-pass and state-free while making the score computation numerically stable under quantization.

When robustness to heavy-tailed deviations is needed, the Gaussian channel penalty can be replaced without changing the tap interface. A Student-$t$ drop-in uses $\mathrm{nll}_\ell^{(t)} = \sum_{i=1}^{d_\ell} \frac{\nu_\ell + 1}{2} \log \big(1 + \frac{(a_{\ell,i} - \mu_{\ell,i})^2}{\nu_\ell \sigma_{\ell,i}^2}\big) + \frac{1}{2} \log \sigma_{\ell,i}^2$, while a Huberized variant uses $\mathrm{nll}_\ell^{(H)} = \sum_{i=1}^{d_\ell} \rho_\delta\big(\frac{a_{\ell,i} - \mu_{\ell,i}}{\sigma_{\ell,i}}\big) + \frac{1}{2} \log \sigma_{\ell,i}^2$ with $\rho_\delta(u) = \frac{1}{2} u^2$ for $|u| \leq \delta$ and $\rho_\delta(u) = \delta|u| - \frac{1}{2}\delta^2$ otherwise. For multi-modality, a small diagonal mixture $p(a_\ell \mid a_{\ell-1}) = \sum_k \pi_k \mathcal{N}(\mu_{\ell,k}, \mathrm{diag}(\sigma_{\ell,k}^2))$ with mixture logits from $z_\ell$ is a drop-in at modest extra cost, requiring only a small log-sum-exp in float16.

## 2.2 TRAINING OBJECTIVE AND REGULARIZATION

Training attaches the projectors and heads at the tapped layers and learns them using a label-free auxiliary objective that encourages predictable depth-wise transitions on in-distribution data. For a

minibatch $\mathcal{B}$ and a tap $\ell \in \mathcal{S}$, define the standardized residual $u_\ell = (a_\ell - \mu_\ell) \odot \sigma_\ell^{-1}$ and the quadratic term $q_\ell = \|u_\ell\|_2^2$. Interpreting the head output as a diagonal Gaussian, the per-tap NLL can be written as $\text{nll}_\ell = \frac{1}{2}q_\ell + \frac{1}{2}\mathbf{1}^\top \log \sigma_\ell^2$. Averaging over examples and taps yields $\mathcal{L}_{\text{SS}} = \frac{1}{|\mathcal{B}|} \sum_{x \in \mathcal{B}} \sum_{\ell \in \mathcal{S}} \text{nll}_\ell$, and the total objective is $\mathcal{L} = \mathcal{L}_{\text{clf}} + \lambda_{\text{SS}} \mathcal{L}_{\text{SS}} + \lambda_{\text{reg}} \mathcal{R}$. Although the classifier is trained with labels via $\mathcal{L}_{\text{clf}}$, the SNAP-UQ heads are trained to model feature transitions in a self-supervised manner; the learned transition model then provides an uncertainty signal at test time without any online supervision.

To avoid degenerate scale predictions and keep the heads well-conditioned, we enforce a variance floor using $\sigma_\ell^2 = \text{softplus}(\xi_\ell) + \epsilon^2$ with a small constant $\epsilon > 0$. We additionally penalize extreme or unstable scales and control head capacity via $\mathcal{R} = \sum_{\ell \in \mathcal{S}} \|\log \sigma_\ell^2\|_1 + \lambda_w \sum_{\ell \in \mathcal{S}} (\|W_{\mu,\ell}\|_2^2 + \|W_{\sigma,\ell}\|_2^2)$, where $W_{\mu,\ell}$ and $W_{\sigma,\ell}$ are the weight matrices producing $\mu_\ell$ and $\log \sigma_\ell^2$. The scale term discourages pathological behavior where the model hides errors by inflating $\sigma_\ell$ or produces overly sharp scales that make the score hypersensitive to quantization noise, which is particularly important for stable int8 inference.

On small datasets or sensitive backbones, we optionally block gradients from the self-supervised term into the backbone by detaching the tapped activation in the residual, using $\text{nll}_\ell^{\text{detach}} = \frac{1}{2}\|(\text{stopgrad}(a_\ell) - \mu_\ell) \odot \sigma_\ell^{-1}\|_2^2 + \frac{1}{2}\mathbf{1}^\top \log \sigma_\ell^2$. This avoids a tug-of-war between improving classification loss and improving transition predictability, while still fitting the tiny heads to the backbone's induced feature dynamics (see Appx. N). To prevent wide layers from dominating the auxiliary loss purely by dimensionality, we balance tap contributions using $\widetilde{\text{nll}}_\ell = \frac{1}{d_\ell}\text{nll}_\ell$ and compute $\mathcal{L}_{\text{SS}} = \frac{1}{|\mathcal{B}|} \sum_{x \in \mathcal{B}} \sum_{\ell \in \mathcal{S}} \widetilde{\text{nll}}_\ell$, which makes each tap contribute on a comparable scale across architectures.

In practice, we tune only a small number of scalars. We sweep a single auxiliary weight $\lambda_{\text{SS}} \in \{10^{-3}, 5 \cdot 10^{-3}, 10^{-2}\}$ on an in-distribution validation split and keep $\lambda_w$ small and fixed. The heads are linear and quantized to `int8`; computing functions of $\log \sigma^2$ uses a small LUT so the implementation remains MCU-friendly. At test time, we do not compute $\mathcal{L}_{\text{SS}}$; we only evaluate the standardized residuals (equivalently, $\bar{q}_\ell(x)$) and aggregate them into the surprisal score used for uncertainty mapping (Appx. J and N.3).

## 2.3 Single-pass surprisal and mapping

At inference, the backbone forward pass produces the activations $\{a_\ell\}$ and each tapped head outputs $(\mu_\ell, \sigma_\ell^2)$ from $z_\ell$. We compute the standardized error per tap as $e_\ell(x) = \|(a_\ell - \mu_\ell) \odot \sigma_\ell^{-1}\|_2^2$ and normalize it as $\bar{e}_\ell(x) = \frac{1}{d_\ell}e_\ell(x)$, then aggregate across taps using $S(x) = \sum_{\ell \in \mathcal{S}} w_\ell \bar{e}_\ell(x)$ with $w_\ell \geq 0$ and $\sum_\ell w_\ell = 1$. This scalar $S(x)$ can be interpreted as a depth-wise energy: it is small when the feature evolution is consistent with the learned in-distribution transition model, and it grows when intermediate transformations become unlikely. Because the computation is local to the forward pass and does not require storing histories, it is naturally state-free and constant-memory.

To translate surprisal into a decision-oriented uncertainty score, we fit a monotone mapping offline using a small labeled development mix (ID plus CID/OOD). We use a logistic map (Platt, 1999) that can optionally incorporate a simple instantaneous confidence proxy. The proxy is defined as $m(x) = \alpha(1 - C_\phi(x)) + (1 - \alpha)(1 - m^{\text{mg}}(x))$ with $\alpha \in [0, 1]$, and the final uncertainty is $U(x) = \sigma(\beta_0 + \beta_1 S(x) + \beta_2 m(x))$. Setting $\beta_2 = 0$ yields a purely label-free mapping from $S(x)$ to uncertainty, while including $m(x)$ often improves separability because it injects complementary information from the classifier's output layer without requiring any additional passes. All parameters $(\beta_0, \beta_1, \beta_2)$ are fit once offline, so inference only requires evaluating $S(x)$ and a tiny scalar map.

Once $U(x)$ is available, decisions can be made using a fixed threshold $U(x) \geq \tau$, by selecting $\tau$ to target a desired risk–coverage operating point on the dev set, or via a budgeted controller that caps the long-run abstention rate at $b$ by choosing the largest $\tau$ such that $\mathbb{E}[\mathbb{I}[U(x) \geq \tau]] \leq b$ under the dev distribution (Appx. K). When budgets are tight or calibration must be especially accurate, isotonic regression (Zadrozny & Elkan, 2002) can replace the logistic map, providing a nonparametric monotone transform from $S(x)$ (or from the pair $(S(x), m(x))$) to estimated error probability (Appx. J). In all cases, calibration is performed offline and does not rely on online labels, preserving the deployment assumptions.

Overall, training attaches lightweight per-layer predictors and optimizes a self-supervised surprisal loss, then fits a monotone map from aggregated surprisal to error probability (Alg. 1). At inference, a single pass computes standardized errors at the taps, aggregates them into $S(x)$, maps to $U(x)$, and thresholds, without extra passes, buffers, or online labels (Alg. 2).

## 2.4 COMPLEXITY, FOOTPRINT, AND MCU IMPLEMENTATION

Let $d_\ell = \dim(a_\ell)$ and $r_\ell = \dim(z_\ell)$. For linear $g_\ell$ with two heads (predicting $\mu$ and $\log \sigma^2$), the parameter count is $\#\theta_\ell \approx 2\,d_\ell r_\ell + 2d_\ell$ (biases included) and the compute is $\mathrm{FLOPs}_\ell = O(d_\ell r_\ell)$. With $|\mathcal{S}| = 2$ taps, $r_\ell \in [32, 128]$, and $d_\ell \in [128, 512]$, the extra FLOPs are typically $< 2\%$ of tiny backbones (e.g., DS-CNN/ResNet-8), and the flash footprint is a few tens of KB at 8-bit weights. Runtime memory stores $\{P_\ell, W_{\mu,\ell}, W_{\sigma,\ell}\}$ plus per-channel scales; there is no $O(W)$ temporal buffer (Appx. L), and all computation is streaming-friendly because it runs at the moment the tapped activation is produced.

We store $P_\ell, W_{\mu,\ell}, W_{\sigma,\ell}$ as int8 with per-tensor scales, compute projections and heads with int32 accumulators, and then compute the standardized residual using integer-friendly arithmetic. To avoid exponentials, we operate on $\log \sigma_\ell^2$ and approximate $\exp(-\frac{1}{2} \log \sigma_\ell^2)$ with a small LUT (per-channel or shared), clamping $\log \sigma_\ell^2$ for stability and applying a small $\epsilon$ floor when forming $\sigma_\ell^{-1}$. This implementation preserves monotonicity of the score with respect to activation deviations, which is critical for reliable thresholding, while keeping the end-to-end pipeline compatible with MCU toolchains.

Taps are chosen at the end of a mid block (capturing shifts in local statistics and mid-level structure) and at the penultimate block (capturing shifts in class-specific abstraction). We set ranks $r_\ell$ so that each tap's compute stays within a small budget (e.g., $\leq 1\%$ of backbone FLOPs), and either keep weights uniform or set $w_\ell \propto 1/\mathrm{Var}(\bar{e}_\ell)$ on dev data to reduce the influence of noisy taps.

## 2.5 THEORY: LINKS TO LIKELIHOOD AND MAHALANOBIS

**Proposition 2.1** (Surprisal–likelihood equivalence under diagonal-Gaussian). *If $p_\theta(a_\ell \mid a_{\ell-1}) = \mathcal{N}(\mu_\ell, \mathrm{diag}(\sigma_\ell^2))$ as above, then $-\log p_\theta(a_\ell \mid a_{\ell-1}) = \frac{1}{2}\,e_\ell(x) + \frac{1}{2}\sum_{i=1}^{d_\ell} \log \sigma_{\ell,i}^2 + const$, so $S(x)$ is (up to additive/multiplicative constants and layer weighting) the depth-wise negative log-likelihood. Higher $S(x)$ implies lower conditional likelihood of the observed activations.*

**Proposition 2.2** (Relation to Mahalanobis scores). *Let the usual Mahalanobis score use unconditional, classwise feature Gaussians at layer $\ell$. If the true feature dynamics are approximately linear, $a_\ell \approx W_\ell a_{\ell-1} + b_\ell + \varepsilon_\ell$ with $\varepsilon_\ell \sim \mathcal{N}(0, \Sigma_\ell)$, then the conditional energy $e_\ell$ equals the squared Mahalanobis distance of $a_\ell$ to the conditional mean $W_\ell a_{\ell-1} + b_\ell$ under covariance $\Sigma_\ell$. Hence SNAP-UQ measures deviations from depth-wise dynamics rather than unconditional, class-averaged statistics, improving sensitivity to distribution shift that alters inter-layer transformations.*

**Proposition 2.3** (Affine invariance (scale)). *Suppose batch-normalized activations admit per-channel affine transforms $a_\ell \mapsto s \odot a_\ell + t$. If $P_\ell$ and $g_\ell$ are trained jointly, the standardized error $e_\ell$ is invariant to such rescalings at optimum because $\mu_\ell$ and $\sigma_\ell$ co-adapt; formally $e_\ell$ is unchanged under $s$ when $\mu_\ell \mapsto s \odot \mu_\ell + t$ and $\sigma_\ell \mapsto |s| \odot \sigma_\ell$.*

Proof sketches are given in Appx. H. Proposition 2.1 follows by expanding the Gaussian NLL; Proposition 2.2 follows by conditioning a linear-Gaussian transition model and relating the quadratic form to a conditional Mahalanobis energy; Proposition 2.3 follows directly from the standardized residual under reparameterization.

## 2.6 VARIANTS AND ABLATIONS

Using $\Sigma_\ell = \mathrm{diag}(\sigma_\ell^2) + B_\ell B_\ell^\top$ with small $k_\ell \in \{4, 8\}$ often tightens the conditional model with negligible extra cost, since inference requires only matrix–vector products with $B_\ell$ and a small system solve. A small $K$-component diagonal mixture can handle multi-modality in depth-wise transitions by letting different modes specialize to distinct feature regimes; logits from $z_\ell$ select mixture weights, and the main additional operation is a log-sum-exp (done in float16). Detaching $a_\ell$ inside $\mathcal{L}_{\mathrm{SS}}$ can improve stability on small datasets by preventing the auxiliary loss from perturbing the backbone, and we report both attached and detached variants in ablations (Appx. N).

For mapping, isotonic regression can replace logistic when the operating point is budgeted and calibration tightness matters, yielding improved risk–coverage behavior (Kull et al., 2019). Quantization-aware training (QAT) can further reduce int8 drift by inserting fake quantization on $P_\ell$ and the heads, and by explicitly quantizing $\log \sigma^2$ to 8-bit with shared scale (Choi et al., 2018). Across these variants, the main conceptual role of $S(x)$ remains the same: it is a conditional, layer-aware energy computed along depth that flags shifts in feature dynamics that plain confidence or margin can miss. SNAP-UQ is complementary to ensembles, MC dropout, and TTA (which increase compute) and to temporal methods (which require buffering); when resources allow, $S(x)$ can also be used as an input feature to richer controllers, while preserving the single-pass MCU-friendly baseline.

# 3 EVALUATION METHODOLOGY

Our objective is to test whether *depth-wise surprisal*, the core of SNAP-UQ, provides a practical, on-device uncertainty signal under TinyML constraints.

**Hardware and toolchain.** We target two common MCU envelopes: a higher-capacity microcontroller with a few MB of flash and several hundred KB of SRAM (**Big-MCU**) and an ultra-low-power part with sub-MB flash and tens of KB SRAM (**Small-MCU**) (STMicroelectronics, 2019; 2018). Builds use the vendor toolchain with `-O3` (Chen et al., 2018); CMSIS-NN kernels are enabled where available (Lai et al., 2018; David et al., 2021). The clock is fixed at the datasheet nominal to avoid DVFS confounds.

**Cost and runtime accounting.** **Flash** is reported from the final ELF after link-time garbage collection. **Peak RAM** comes from the linker map plus the *incremental* buffers for SNAP-UQ's projectors/heads. **Latency** is end-to-end time per inference measured by the on-chip cycle counter with interrupts masked; each figure averages 1,000 runs (std. dev. shown). **Energy** (selected runs) integrates current over time using a shunt on the board rail.

**Backbones and datasets.** Vision: MNIST, CIFAR-10, TinyImageNet (Lecun et al., 1998; Krizhevsky, 2009; Le & Yang, 2015). Audio: SpeechCommands v2 (Warden, 2018). Backbones: a 4-layer DSCNN for SpeechCommands (Zhang et al., 2018), a compact residual net for CIFAR-10 (Banbury et al., 2021), and a MobileNetV2-style model for TinyImageNet (Howard et al., 2017; Cai et al., 2020). Standard augmentation is used; temperature scaling is applied on the ID validation split (Hendrycks et al., 2020). Full dataset and schedule details appear in Appx. A and B.

**SNAP-UQ configuration (inference-friendly).** We attach two taps (end of a mid block and the penultimate block). Each tap uses a $1{\times}1$ projector $P_\ell$ with global average pooling and two int8 heads that output $(\mu_\ell, \log \sigma_\ell^2)$. We set ranks $r_\ell \in \{32, 64, 128\}$ and auxiliary weight $\lambda_{\text{SS}} \in \{10^{-3}, 10^{-2}\}$. To avoid exponentials on-device, $\log \sigma^2$ is clamped and mapped to per-channel multipliers via a 256-entry LUT. A 3-parameter logistic map converts $(S, m)$ to $U$; an isotonic alternative is reported in Appx. J.

**Baselines and tuning.** We compare against single-pass confidence (max-probability, entropy) (Hendrycks & Gimpel, 2018), temperature scaling, classwise Mahalanobis at tapped layers, energy-based scoring (Liu et al., 2021), evidential posteriors when they fit, and, on Big-MCU only, MC Dropout (Gal & Ghahramani, 2016) and Deep Ensembles (Lakshminarayanan et al., 2017) and (Tack et al., 2020; Du et al., 2022; Wang et al., 2022), QUTE (Ghanathe & Wilton, 2024) All methods share backbones and input pipelines; thresholds and any temperature/isotonic parameters are tuned on a common development split. Implementation details and grids appear in Appx. C.

**CID/OOD protocols and streaming setup.** We use MNIST-C, CIFAR-10-C, TinyImageNet-C (Mu & Gilmer, 2019; Hendrycks & Dietterich, 2019). For SpeechCommands we synthesize CID using impulse responses, background noise, reverberation, and time/pitch perturbations (Appx. D). OOD sets are Fashion-MNIST (for MNIST), SVHN (for CIFAR-10), non-keyword/background audio (for SpeechCommands), and disjoint TinyImageNet classes. For *streaming* evaluation, we concatenate clean ID segments with CID segments of rising severity and short OOD bursts (Gama et al., 2014). Events are labeled offline via sliding-window accuracy (window $m{=}100$) falling below an ID band estimated from a held-out run ($\mu_{\text{ID}} \pm 3\sigma_{\text{ID}}$). The monitor never sees labels online. We score event detection by AUPRC and report thresholded detection delay; thresholds are selected offline on the dev split (Appx. E).

## 4 RESULTS

We evaluate *SNAP-UQ* on four axes: deployability on MCUs, monitoring under corrupted streams, failure detection (ID/CID and OOD), and probabilistic quality on ID. Unless noted, results are averaged over three seeds; 95% confidence intervals (CIs) are obtained via $1{,}000\times$ bootstrap over examples. Ablations (tap placement/rank, quantization variants, mapping alternatives, risk–coverage surfaces, reliability diagrams, and error clusters) are deferred to Appx N–F, and O for a single-pass head-to-head and decision-centric risk–coverage analyses.

### 4.1 ON-DEVICE FIT AND RUNTIME

**Setup.** All methods share the same backbones, preprocessing, and integer kernels. Builds use vendor toolchains with `-O3` and CMSIS-NN where available; input I/O is excluded and timing spans from first byte in SRAM to posterior/uncertainty out. *Flash* is read from the final ELF (post link-time GC). *Peak RAM* is computed from the linker map plus scratch buffers required by the method. *Latency* is measured with the MCU cycle counter at datasheet nominal clock (interrupts masked), averaged over 1,000 runs. *Energy* integrates current over time via a calibrated shunt at 20 kHz. All baselines are compiled with the same quantization scheme; when a method does not fit, we report *OOM* and omit latency/energy.

**Findings.** Table 1 summarizes deployability. On **Big-MCU**, SNAP-UQ reduces latency by **35%** (SpeechCmd) and **24%** (CIFAR-10) vs. EE-ens, and by **26–34%** vs. DEEP, with **49–46%** and **37–57%** flash savings, respectively. On **Small-MCU**, both ensembles are *OOM* for CIFAR-10; for SpeechCmd, SNAP-UQ is **33%** faster and **16–24%** smaller, and uses **1.6–2.0×** less peak RAM than EE-ens due to absent exit maps and int8 heads. These trends hold across seeds; CIs are narrow (typically $\pm$1–3% of the mean).

Table 1: **MCU deployability.** Flash (KB) / Peak RAM (KB) / Latency (ms) / Energy (mJ). OOM: method does not fit.

| **Big-MCU** (SpeechCmd) | BASE | EE-ens | DEEP | **SNAP-UQ** |
|---|---|---|---|---|
| Flash ↓ | 220 | 360 | 290 | **182** |
| Peak RAM ↓ | 84 | 132 | 108 | **70** |
| Latency ↓ | 60 | 85 | 70 | **52** |
| Energy ↓ | 2.1 | 3.0 | 2.5 | **1.7** |
| **Big-MCU** (CIFAR-10) | | | | |
| Flash ↓ | 280 | 540 | 680 | **292** |
| Peak RAM ↓ | 128 | 190 | 176 | **120** |
| Latency ↓ | 95 | 110 | 125 | **83** |
| Energy ↓ | 3.7 | 4.1 | 4.6 | **3.3** |
| **Small-MCU** (SpeechCmd) | | | | |
| Flash ↓ | 140 | 320 | 210 | **118** |
| Peak RAM ↓ | 60 | 104 | 86 | **51** |
| Latency ↓ | 170 | 240 | 200 | **113** |
| Energy ↓ | 6.0 | 8.6 | 7.3 | **4.7** |
| **Small-MCU** (CIFAR-10) | | | | |
| Flash ↓ | 180 | OOM | OOM | **158** |
| Peak RAM ↓ | 92 | OOM | OOM | **85** |
| Latency ↓ | 260 | OOM | OOM | **178** |
| Energy ↓ | 9.5 | OOM | OOM | **6.4** |

*Note.* BASE includes on-device softmax+entropy and a small stats buffer; SNAP-UQ replaces this with a single scalar surprisal $S(x)$ and is compiled INT8 with kernel fusion.

### 4.2 MONITORING CORRUPTED STREAMS

**Protocol.** We construct unlabeled streams by concatenating ID segments, CID segments (severities 1–5 from MNIST-C/CIFAR-10-C/TinyImageNet-C or our SpeechCmd-C generator), and short OOD bursts. Ground-truth *events* are labeled offline when a sliding-window accuracy (window $m{=}100$) falls below an ID band estimated from a separate held-out ID run ($\mu_{\mathrm{ID}} - 3\sigma_{\mathrm{ID}}$). Thresholds for each method are fixed on a development stream to maximize the F1 score and then held constant for evaluation.

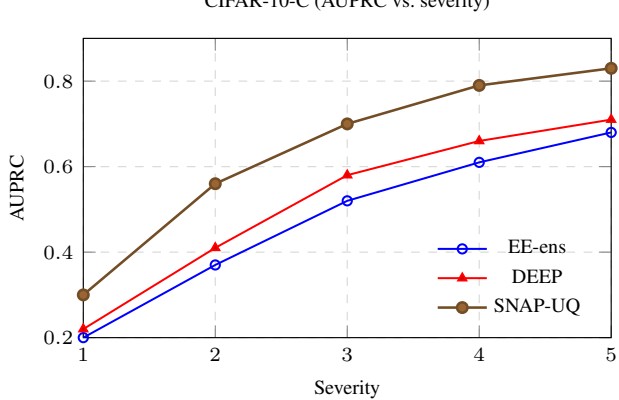

Figure 2: **CIFAR-10-C:** AUPRC vs. corruption severity. SNAP-UQ scales fastest with severity.

**Findings.** SNAP-UQ yields the best average AUPRC and shortest delays on **MNIST-C** and **SpeechCmd-C** (Table 2), and its AUPRC grows fastest with severity on **CIFAR-10-C** (Fig. 2). Qualitatively, depth-wise surprisal reacts earlier than softmax entropy as distortions accumulate, reducing late alarms. False positives on clean ID segments remain low at matched recall (Appx F).

Table 2: **Accuracy-drop detection on CID streams.** AUPRC (higher is better) and median detection delay (frames) at a single dev-chosen threshold.

| | MNIST-C | | SpeechCmd-C | |
|---|---|---|---|---|
| *Context (accuracy)* | Clean: 99.0% — CID: 90.8% (−8.2 pp) | | Clean: 95.2% — CID: 86.0% (−9.2 pp) | |
| Method | AUPRC ↑ | Delay ↓ | AUPRC ↑ | Delay ↓ |
| BASE | 0.54 | 42 | 0.52 | 67 |
| EE-ens | 0.63 | 31 | 0.59 | 55 |
| DEEP | 0.56 | 35 | 0.58 | 57 |
| **SNAP-UQ** | **0.66** | **24** | **0.65** | **41** |

### 4.3 FAILURE DETECTION (ID, CID, OOD)

**Tasks.** We report AUROC for two threshold-free discriminations: **ID✓ — ID×** (correct vs. incorrect among ID + CID) and **ID✓ — OOD** (ID vs. OOD). For operational relevance, we further compare selective risk at fixed coverage and selective NLL in Appx F.

**Findings.** With a single forward pass, SNAP-UQ leads on **ID✓ — ID×** for MNIST and SpeechCmd and remains competitive on CIFAR-10; on **ID✓ — OOD**, it ties the best on SpeechCmd and is close to the strongest semantic OOD detector on CIFAR-10 (Table 3). These gains mirror the monitoring results: when corruptions are label-preserving, depth-wise surprisal provides sharper separation than confidence-only scores.

Table 3: **Failure detection.** AUROC for ID✓ — ID× and ID✓ — OOD.

| Method | ID✓ — ID× | | | ID✓ — OOD | | |
|---|---|---|---|---|---|---|
| | **MNIST** | **SpCmd** | **CIFAR-10** | **MNIST** | **SpCmd** | **CIFAR-10** |
| *Context (clean acc.)* | 99.0% | 95.2% | 83.7% | N/A (OOD sets) | | |
| BASE | 0.75 | 0.90 | 0.84 | 0.07 | 0.90 | 0.88 |
| MCD | 0.74 | 0.89 | **0.87** | 0.48 | 0.89 | 0.89 |
| DEEP | 0.85 | 0.91 | 0.86 | 0.78 | **0.91** | 0.92 |
| EE-ensemble | 0.85 | 0.90 | 0.85 | 0.85 | 0.90 | 0.90 |
| G-ODIN | 0.72 | 0.74 | 0.83 | 0.40 | 0.74 | **0.95** |
| HYDRA | 0.81 | 0.90 | 0.83 | 0.71 | 0.90 | 0.90 |
| QUTE | 0.87 | 0.91 | 0.86 | 0.84 | **0.91** | 0.91 |
| **SNAP-UQ** | **0.90** | **0.94** | **0.87** | **0.86** | **0.92** | 0.94 |

## 4.4 CALIBRATION ON ID

**Findings.** SNAP-UQ improves proper scores on **MNIST** and **SpeechCmd**, lowering both NLL and BS while reducing ECE relative to BASE. On **CIFAR-10**, a capacity-matched variant (SNAP-UQ$^+$) matches DEEP on BS with comparable NLL, while preserving single-pass inference (Table 4). Reliability curves and selective-calibration analyses appear in Appx N.5. **Raw vs. calibrated.** We also evaluate using the raw surprisal $S(x)$ (no calibration) as the uncertainty signal; rankings remain strong, but decision metrics improve with a tiny logistic/isotonic map. See Appx. N.6 for numbers.

Table 4: **ID calibration.** Lower is better. SNAP-UQ$^+$ increases projector rank and adds a low-rank covariance correction.

| MNIST | NLL $\downarrow$ | BS $\downarrow$ | ECE $\downarrow$ |
|---|---|---|---|
| BASE | 0.285 | 0.012 | 0.028 |
| Temp. scaled | 0.242 | 0.010 | 0.022 |
| **SNAP-UQ** | **0.202** | **0.008** | **0.016** |

| SpeechCmd | NLL | BS | ECE |
|---|---|---|---|
| BASE | 0.306 | 0.012 | 0.024 |
| Temp. scaled | 0.228 | 0.009 | 0.021 |
| **SNAP-UQ** | **0.197** | **0.008** | **0.016** |

| CIFAR-10 | NLL | BS | ECE |
|---|---|---|---|
| BASE | 0.415 | 0.021 | 0.031 |
| DEEP | **0.365** | **0.017** | **0.015** |
| SNAP-UQ$^+$ | 0.363 | **0.017** | 0.021 |

---

**Mapping recipe**

We convert surprisal $S$ to a calibrated uncertainty with a tiny *monotone* link.

▷ Isotonic (recommended for fixed coverage monitoring): nonparametric, preserves ordering, adapts to nonlinearities; thresholds remain stable under CID/OOD shifts.

▷ Logistic (recommended for in distribution calibration): use $U = \sigma(\beta_0 + \beta_1 S + \beta_2 m)$, where $m$ may be a confidence cue (e.g., top 2 logit margin or max softmax).

▷ Default include $m$: adding $m$ consistently lowers selective risk at a given coverage with negligible cost; when $m$ is unavailable, set $\beta_2=0$.

▷ Practical recipe: for robust event detection or fixed coverage, fit isotonic on a small dev stream and threshold $U$; for best ID scores (NLL/BS/ECE), fit the logistic map on an ID dev split (logit temperature optional). Because both maps are monotone, AUROC/AUPRC stay nearly unchanged; the choice mostly affects decision centric metrics (selective risk, fixed coverage error) and can be swapped without retraining.

---

## 5 CONCLUSION AND DISCUSSION

**SNAP-UQ** converts depth-wise next-activation prediction into a single-pass, integer-friendly uncertainty signal that drops neatly into CMSIS-NN pipelines and fits within kilobyte-scale MCU budgets. Empirically it improves accuracy-drop monitoring on corrupted streams, remains competitive on ID✓ — ID✗ and ID✓ — OOD failure detection, and strengthens ID calibration without temporal buffers, auxiliary exits, or extra passes. Limitations include reliance on access to intermediate activations, diagonal/low-rank covariance that may miss fine cross-channel structure, and sensitivity to tap placement and projector rank. Future work targets hardware-aware tap/rank selection, richer yet cheap heads (low-rank+diag, mixtures, Student-$t$), self-tuning calibration under budgets, and lightweight fusion with a semantic OOD cue.

## LLM USAGE

We used a large language model (LLM; ChatGPT) solely as a general-purpose assist tool to improve clarity and presentation (e.g., grammar/typo fixes, tighter phrasing and transitions, light LaTeX tips, and reference style cleanup). We did not use an LLM for research ideation, experimental design, data analysis, result interpretation, drafting substantive technical content, equations/algorithms, figure creation, or code implementation. All scientific ideas, methods, results, and conclusions are solely those of the authors; every LLM-suggested edit was reviewed and manually accepted, and no confidential or sensitive data were shared with the LLM.

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

APPENDIX

# A  DATASETS AND PREPROCESSING

For every dataset, we follow the official train/test split and create an additional *development* split from the training portion for calibration, threshold selection, and any dev-tuned hyperparameters, ensuring that the test set is never used during model selection. Unless stated otherwise, we reserve 10% of the training set as dev, using stratified sampling by class and keeping the split fixed across seeds to make comparisons stable and to avoid accidental dev/test mixing. All reported test results use the untouched official test split, while any calibration (e.g., temperature or uncertainty mapping) is fit only on the dev split.

## A.1  VISION

MNIST contains 60k training and 10k test grayscale images at $28\times28$. We keep the native spatial resolution (no interpolation), normalize inputs with $\mu = 0.1307$ and $\sigma = 0.3081$, and apply light geometric augmentation to reduce sensitivity to small pose changes: random affine rotation within $\pm10°$ and translation of up to 2 pixels. These augmentations are intentionally mild so the classifier remains aligned with the standard MNIST setting while still improving robustness to small nuisance shifts. Unless otherwise specified, we use a batch size of 256.

CIFAR-10 consists of 50k training and 10k test RGB images at $32\times32$. We adopt the standard augmentation recipe used for small-image classification, including random crop after 4-pixel reflection padding and random horizontal flipping with probability $p = 0.5$. We optionally apply Cutout with a $16\times16$ mask (disabled in Small-MCU ablations to keep preprocessing minimal and comparable across constrained deployments) and a light color jitter with brightness/contrast/saturation ranges of $\pm0.2$ to encourage invariance to illumination and mild color shifts. Inputs are normalized using per-channel means $(0.4914, 0.4822, 0.4465)$ and standard deviations $(0.2023, 0.1994, 0.2010)$.

TinyImageNet contains 200 classes with 100k training images (500 per class) and a 10k validation set (50 per class), all at $64\times64$. We keep the native resolution and use a random resized crop to $64\times64$ with scale range $[0.8, 1.0]$ to inject mild scale variability while preserving most content. We also apply horizontal flip with probability $p = 0.5$, color jitter (0.2/0.2/0.2), and random grayscale with probability $p = 0.1$ to reduce over-reliance on color cues. Normalization uses ImageNet statistics $\mu = (0.485, 0.456, 0.406)$ and $\sigma = (0.229, 0.224, 0.225)$.

## A.2  AUDIO

For SpeechCommands v2 we use the standard 12-class keyword-spotting setup with classes {yes, no, up, down, left, right, on, off, stop, go, unknown, silence}. Audio is mono at 16 kHz and we extract log-Mel features from 1 s clips using a 25 ms analysis window, 10 ms hop, 512-point FFT, and 40 Mel bands. We apply per-utterance mean/variance normalization (MVN) to reduce sensitivity to channel gain and recording conditions. To avoid introducing artificial silence patterns when utterances are shorter than 1 s, we reflect-pad the waveform to 1 s before feature extraction. Training-time augmentation includes random time shift within $\pm100$ ms, background noise mixing (from the dataset's provided noise clips and, where noted, external noise subsets) with SNR uniformly sampled from $[5, 20]$ dB, light time/frequency masking (SpecAugment) with up to 2 time masks of width 20 frames and 2 frequency masks of width 5 bins, and random gain within $\pm2$ dB to improve robustness to amplitude variation.

## A.3  CORRUPTIONS (CID) AND OOD

For vision, we evaluate corruption-induced distribution shift using MNIST-C, CIFAR-10-C, and TinyImageNet-C. We include all available corruption types, excluding *snow* for MNIST-C (not defined), and evaluate severities $\{1, \dots, 5\}$ individually before averaging to obtain a single CID summary per corruption family. The corruptions span noise (Gaussian, shot, impulse), blur (defocus, glass, motion, zoom), weather (snow, frost, fog), and digital artifacts (contrast, brightness, pixelate, JPEG), providing a broad stress test of both low-level and mid-level invariances.

---

**Algorithm 1** SNAP-UQ training (offline, label-free auxiliary)

---

**Require:** Backbone blocks $f_1, \ldots, f_D$, classifier head $g$, tap set $\mathcal{S}$, projectors $\{P_\ell\}$, predictor heads $\{g_\ell\}$, weights $\lambda_{\mathrm{SS}}, \lambda_{\mathrm{reg}}$

1: **for** epochs **do**
2:     **for** minibatch $\mathcal{B}$ **do**
3:         Forward pass through the backbone to obtain activations $a_1, \ldots, a_D$ and logits for classification
4:         **for** $\ell \in \mathcal{S}$ **do**
5:             $z_\ell \leftarrow P_\ell a_{\ell-1}; \quad (\mu_\ell, \log \sigma_\ell^2) \leftarrow g_\ell(z_\ell)$
6:         **end for**
7:         $\mathcal{L}_{\mathrm{clf}} \leftarrow$ cross-entropy over $(x, y) \in \mathcal{B}$
8:         $\mathcal{L}_{\mathrm{SS}} \leftarrow \frac{1}{|\mathcal{B}|} \sum_{x \in \mathcal{B}} \sum_{\ell \in \mathcal{S}} \left[ \frac{1}{2} \| (a_\ell - \mu_\ell) \odot \sigma_\ell^{-1} \|_2^2 + \frac{1}{2} \mathbf{1}^\top \log \sigma_\ell^2 \right]$
9:         $\mathcal{R} \leftarrow$ regularization terms (variance floor/scale control and weight decay on projector and head parameters)
10:         Update backbone and head parameters by descending $\mathcal{L} = \mathcal{L}_{\mathrm{clf}} + \lambda_{\mathrm{SS}} \mathcal{L}_{\mathrm{SS}} + \lambda_{\mathrm{reg}} \mathcal{R}$
11:     **end for**
12: **end for**
13: Fit a monotone calibration map offline (logistic with parameters $(\beta_0, \beta_1, \beta_2)$ or isotonic regression) to predict error probability from the surprisal score $S$ and the optional confidence proxy $m$

---

For SpeechCommands, we construct SpeechCmd-C as a set of label-preserving degradations that reflect realistic acoustic variability. We apply room impulse responses with RT60 sampled in [0.2, 1.0] s, background noise mixing (from the dataset's noise clips and selected subsets of UrbanSound8K and ESC-50) with SNR in [0, 20] dB, and band-limiting using Butterworth low/high/band-pass filters with random cutoffs. We additionally apply pitch shifts within $\pm 2$ semitones, time stretching by factors in [0.9, 1.1], and reverberation pre-delay in [0, 20] ms. These transformations are mapped to five severities by systematically increasing difficulty (e.g., lowering SNR and increasing perturbation magnitude) while keeping labels unchanged.

For OOD evaluation, we use standard cross-dataset transfers that preserve the input modality but differ in content distribution. For MNIST we use Fashion-MNIST as OOD; for CIFAR-10 we use SVHN; for SpeechCommands we use non-keyword speech and background noise segments; and for TinyImageNet we use a disjoint 200-class subset not present in training. In TinyImageNet experiments, we treat the official validation set as ID and use a curated 200-class slice from ImageNet-1k as OOD when computing ID✓ — OOD; all OOD images are resized to $64 \times 64$ with bicubic interpolation and normalized identically to ID to ensure that differences reflect semantic shift rather than preprocessing mismatch.

### A.4 REPRODUCIBILITY AND BOOKKEEPING

We run three random seeds $\{13, 17, 23\}$ and generate all splits and corruption streams deterministically per seed. Dev/test leakage is prevented by performing any threshold selection, calibration, or operating-point tuning exclusively on the dev split (or dev streams), while the official test split is used only for final reporting. We store per-example metrics and uncertainty scores to enable nonparametric uncertainty estimates, reporting $1{,}000\times$ bootstrap confidence intervals where appropriate.

## B TRAINING, CALIBRATION, AND BUILD DETAILS

### B.1 BACKBONES AND HEADS

For keyword spotting we use a DSCNN with four depthwise-separable convolution blocks at constant width 64, each followed by BN+ReLU6, then global average pooling and a linear classifier. The input feature map is $40 \times 98$ (Mel$\times$time) and the parameter budget is approximately 130k. We tap SNAP-UQ at the end of block 2 (mid-depth) and block 4 (penultimate) so that the surprisal captures both mid-level acoustic patterns and late class-specific structure. Projector ranks are chosen from

---

**Algorithm 2** SNAP-UQ inference (single pass, state-free)

---

**Require:** Frozen backbone $f_1, \ldots, f_D$ with classifier head, tap set $\mathcal{S}$, projectors $\{P_\ell\}$, predictor heads $\{g_\ell\}$, tap weights $\{w_\ell\}$, mapping parameters $(\beta_0, \beta_1, \beta_2)$, threshold $\tau$
 1: Forward pass through the backbone to obtain activations $a_1, \ldots, a_D$ and class posteriors $p_\phi(y \mid x)$
 2: $\hat{y} \leftarrow \arg\max_c p_\phi(y = c \mid x)$
 3: $C_\phi(x) \leftarrow \max_c p_\phi(y = c \mid x)$
 4: $m^{\mathrm{mg}}(x) \leftarrow p_\phi^{(1)}(x) - p_\phi^{(2)}(x)$         $\triangleright\ p_\phi^{(1)}(x) \geq p_\phi^{(2)}(x) \geq \cdots$ are sorted class probabilities
 5: $m(x) \leftarrow \alpha\big(1 - C_\phi(x)\big) + (1 - \alpha)\big(1 - m^{\mathrm{mg}}(x)\big)$     $\triangleright\ \alpha \in [0, 1]$ is fixed from offline tuning
 6: **for** $\ell \in \mathcal{S}$ **do**
 7:     $z_\ell \leftarrow P_\ell a_{\ell-1};$   $(\mu_\ell, \log \sigma_\ell^2) \leftarrow g_\ell(z_\ell)$
 8:     $\bar{e}_\ell \leftarrow \frac{1}{d_\ell} \sum_{i=1}^{d_\ell} \left( \frac{a_{\ell,i} - \mu_{\ell,i}}{\sigma_{\ell,i}} \right)^2$
 9: **end for**
10: $S \leftarrow \sum_{\ell \in \mathcal{S}} w_\ell\, \bar{e}_\ell$
11: $U \leftarrow \sigma\big(\beta_0 + \beta_1 S + \beta_2 m(x)\big)$                    $\triangleright\ \sigma(\cdot)$ is the logistic sigmoid
12: **if** $U \geq \tau$ and budget controller allows **then**
13:     ABSTAIN
14: **else**
15:     Output $\hat{y}$
16: **end if**

---

$r_\ell \in \{32, 64\}$ to keep the extra compute small, and the heads are linear maps producing $\mu_\ell$ and $\log \sigma_\ell^2$ from $z_\ell$.

For CIFAR-10 we use a compact ResNet (ResNet-8/ResNet-10-like) with three stages of widths 16/32/64, stride-2 downsampling at the first convolution in each stage, followed by GAP and a linear classifier. The parameter count is roughly 0.3–0.5M depending on the exact variant. We tap at the end of stage 2 and at the penultimate stage so that the transition model covers both intermediate spatial abstractions and late features near the classifier. Projector ranks are chosen from $r_\ell \in \{64, 128\}$ based on MCU budget.

For TinyImageNet we use a small MobileNetV2 with width multiplier 0.5 and input $64 \times 64$, using inverted residual blocks with expansion factor 6 and strides [2,2,2] across spatial downsampling, followed by GAP and a linear classifier. The parameter count is approximately 1.3M. We tap at the end of a mid inverted-residual block and at the penultimate inverted-residual block, and choose ranks $r_\ell \in \{64, 128\}$ for Small-MCU and $r_\ell \in \{128, 160\}$ for Big-MCU to trade off footprint and separability.

### B.2 OPTIMIZATION AND SCHEDULES

On MNIST we train with Adam (betas 0.9/0.999) using learning rate $1\mathrm{e}{-3}$ with cosine decay to $1\mathrm{e}{-5}$ over 50 epochs, batch size 256, and weight decay $1\mathrm{e}{-4}$. We set $\lambda_{\mathrm{SS}} = 5\mathrm{e}{-3}$ and warm it up over the first 5 epochs to avoid destabilizing early classifier learning, and we use $\lambda_{\mathrm{reg}} = 1\mathrm{e}{-4}$. When noted, we optionally detach $a_\ell$ after epoch 10 if the dev NLL for the auxiliary heads stalls.

On CIFAR-10 we train with SGD (momentum 0.9, weight decay $5\mathrm{e}{-4}$) and cosine learning rate from 0.2 to $5\mathrm{e}{-4}$ over 200 epochs with batch size 128. We use label smoothing 0.1 and enable MixUp with $\alpha = 0.2$ on Big-MCU only, disabling it for Small-MCU reproducibility runs. We set $\lambda_{\mathrm{SS}} = 1\mathrm{e}{-2}$ and linearly ramp it during the first 20 epochs, and apply global gradient clipping at L2 norm 1.0 to stabilize training under stronger augmentation.

On TinyImageNet we use SGD (momentum 0.9, weight decay $1\mathrm{e}{-4}$) with cosine learning rate from 0.15 to $1\mathrm{e}{-4}$ over 220 epochs and batch size 128. We set $\lambda_{\mathrm{SS}} = 5\mathrm{e}{-3}$ with a 20-epoch ramp to gradually introduce the transition modeling loss. When specified, we apply EMA of model weights with $\tau = 0.999$ for final evaluation to reduce variance in the late training regime.

On SpeechCommands we train with AdamW (weight decay $1\mathrm{e}{-3}$) using learning rate $2\mathrm{e}{-3}$ with cosine decay to $1\mathrm{e}{-5}$ over 80 epochs and batch size 256. SpecAugment is enabled as described in

Sec. A. We set $\lambda_{\text{SS}} = 5e{-}3$ and keep detach disabled, as this configuration is empirically stable for KWS with the chosen backbone.

### B.3 SNAP-UQ-SPECIFIC SETTINGS

We parameterize log-variance using $\sigma^2 = \text{softplus}(\xi) + \epsilon^2$ with $\epsilon = 10^{-4}$ and clamp $\log \sigma^2$ to $[\log 10^{-4}, \log 10^2]$ to avoid extreme dynamic range that can amplify quantization error or lead to vanishing standardized residuals. We regularize scale predictions with a small coefficient $\alpha_{\text{var}} = 1e{-}4$ and apply head weight decay $\alpha_{\text{wd}} = 5e{-}4$ to keep auxiliary heads compact and prevent overfitting of transition statistics.

To prevent wider taps from dominating the auxiliary objective, we normalize per-tap losses by $1/d_\ell$ and set tap weights $\omega_\ell$ inversely proportional to the dev-set variance of $\bar{e}_\ell$, then rescale so that $\sum_\ell \omega_\ell = |\mathcal{S}|$. This weighting tends to down-weight taps whose standardized residuals are inherently noisy (e.g., due to highly variable intermediate representations) while preserving contributions from more stable layers.

To improve int8 deployability, we optionally run a short QAT phase during the final 20% of epochs by inserting fake quantization on $P_\ell$ and head weights using symmetric per-tensor int8 scales, while keeping the loss computation in float32 (Esser et al., 2020). We export int8 weights and a 256-entry LUT for $\exp(-\frac{1}{2} \log \sigma^2)$, following standard post-training quantization refinements (Nagel et al., 2020; Li et al., 2021).

### B.4 CALIBRATION AND THRESHOLDS

For in-distribution calibration we fit a temperature $T$ on the ID dev split by minimizing NLL, then report ID calibration metrics (NLL/BS/ECE) using the temperature-scaled logits (Kull et al., 2019). This step is applied uniformly across softmax-based methods to avoid attributing calibration gains to a particular uncertainty score.

For SNAP-UQ we fit either a three-parameter logistic map $U = \sigma(\beta_0 + \beta_1 S + \beta_2 m)$ using class-balanced logistic regression, or isotonic regression on the scalar $\psi = \gamma S + (1 - \gamma)m$ with $\gamma$ tuned on the dev split. Unless stated otherwise, the main text uses the logistic map and we report isotonic variants in Appx. J. Thresholds $\tau$ are selected on the dev split to match a target coverage (e.g., 90%) or, in streaming experiments, to maximize F1 on event frames; the chosen threshold is then held fixed for all test evaluations.

### B.5 BUILD AND MEASUREMENT ON MCUS

We compile with vendor GCC using `-O3` and link-time optimization, relying on CMSIS-NN kernels for int8 convolution and linear operations when available. For Big-MCU builds we optionally enable `--ffast-math` for the auxiliary heads only, verifying that numerical differences remain negligible (RMSE $< 10^{-4}$) relative to the float reference in our checks.

Quantized export stores $P_\ell$, $W_\mu$, and $W_{\log \sigma^2}$ as int8 with per-tensor scaling, aligned with the backbone's activation quantization scheme. The standardized residual uses a LUT on clamped $\log \sigma^2$ values and accumulates in int32 before a single dequantization to float16 (or fixed-point with shared scale) for final aggregation. This design keeps persistent memory minimal and avoids any temporal buffering beyond the activations already produced by the backbone.

Latency is measured using the on-chip DWT cycle counter with interrupts masked, assuming the input is already in SRAM; the timed region spans from the first layer call through posterior computation and final $U(x)$. Energy is measured for selected runs via shunt integration at $20\,\text{kHz}$ with temperature-compensated calibration, and we report mean$\pm$std over 1,000 inferences.

For reproducibility, we store dataset splits and corruption RNG seeds, per-epoch summaries of $\mathbb{E}[\bar{e}_\ell]$ and its variance, fitted mapping parameters $(\beta_0, \beta_1, \beta_2)$ (or the isotonic step function), MCU build flags and commit hash, per-layer operation counts and timings, and raw per-example scores used to compute bootstrap confidence intervals.

## C  BASELINES AND TUNING DETAILS

All baselines are trained and evaluated under the same constraints as SNAP-UQ, sharing the same training data, backbones, input pipelines, and integer kernels. Unless otherwise noted, calibration and any threshold selection use the ID development split (10% of training) and are kept fixed across seeds. For streaming experiments, each method selects its threshold once on a dev stream and then holds it fixed for all test streams; for AUROC/AUPRC we report threshold-free performance.

### C.1  SCORE DEFINITIONS AND MCU CONSIDERATIONS

Max-probability scoring (Conf) uses $S_{\text{conf}}(x) = 1 - \max_\ell p_\phi(y = \ell \mid x)$. On-device, this adds no extra cost beyond the classifier because the softmax probabilities are already computed for classification, and it requires no additional persistent memory.

Entropy scoring uses $S_{\text{ent}}(x) = -\sum_{\ell=1}^L p_\phi(y = \ell \mid x) \log p_\phi(y = \ell \mid x)$. On-device, the additional cost is negligible relative to the backbone, and the log can be implemented via a small LUT or float16 without affecting the overall MCU budget.

Temperature scaling (Temp) computes calibrated probabilities $\tilde{p}(y \mid x) = \text{softmax}(z/T)$ with a scalar $T > 0$ fitted on the ID dev split by minimizing NLL. We apply this calibration uniformly to BASE/Conf/Entropy when reporting ID calibration metrics, and on-device it reduces to dividing logits by $T$ (typically in float16) without increasing model size.

Classwise Mahalanobis scoring (Maha) uses the same tap set $\mathcal{S}$ as SNAP-UQ, fitting class means $\{\mu_{\ell,c}\}$ and a shared diagonal covariance $\hat{\Sigma}_\ell = \text{diag}(\sigma_\ell^2)$ on the ID training set (to avoid dev leakage). The score is $S_{\text{maha}}(x) = \min_c \sum_{\ell \in \mathcal{S}} w_\ell \cdot \frac{1}{d_\ell} \big(a_\ell(x) - \mu_{\ell,c}\big)^\top \hat{\Sigma}_\ell^{-1} \big(a_\ell(x) - \mu_{\ell,c}\big)$. Using a diagonal inverse avoids matrix–vector products and keeps runtime small, but memory grows with $L \cdot \sum_\ell d_\ell$ because class means must be stored; this is feasible for MNIST/SpeechCommands and can be borderline for TinyImageNet, in which case we keep only the penultimate tap for both Mahalanobis and SNAP-UQ to maintain parity.

Energy-based scoring (EBM) uses logit energy $S_{\text{eng}}(x) = -\log \sum_\ell \exp(z_\ell/T_{\text{eng}})$ with $T_{\text{eng}}$ tuned on dev, where higher energy indicates greater uncertainty. On-device, we compute LSE in float16 using the standard max-shift trick; this adds a small constant overhead and no persistent state.

Evidential posteriors (Evid) replace the softmax with nonnegative evidence $e \in \mathbb{R}_+^L$, define $\alpha = e + 1$, and use $\mathbb{E}[p] = \alpha / \sum_j \alpha_j$. The model is trained with a Dirichlet-based objective (NLL plus a regularizer that discourages high evidence on errors). We report uncertainty via $S_{\text{ep}}(x) = 1 - \max_\ell \mathbb{E}[p_\ell]$ and the total-uncertainty proxy $u = \frac{L}{\sum_j \alpha_j}$. This baseline adds parameters (an evidence head and ReLU) and can be out-of-memory on the smallest devices for CIFAR-10/TinyImageNet; when this happens we report it only where it fits.

MC Dropout (MCD) enables dropout at inference and averages over $N$ stochastic forward passes, with uncertainty computed as predictive entropy $H[\bar{p}]$ or mutual information $H[\bar{p}] - \overline{H[p]}$. We use $N \in \{5, 10\}$ and the dropout rate from training. Because latency and energy scale roughly linearly with $N$, we restrict MCD to Big-MCU settings and omit it on Small-MCU.

Deep ensembles (DEEP) train $M$ independently initialized replicas ($M \in \{3, 5\}$) and compute uncertainty as $H[\frac{1}{M} \sum_m p^{(m)}]$. This baseline typically yields strong uncertainty but multiplies either runtime (if evaluated on-device sequentially) or flash footprint (if multiple models are stored), so we deploy it only on Big-MCU and mark cases that exceed memory as out-of-memory.

### C.2  HYPERPARAMETER GRIDS AND SELECTION

All scalar hyperparameters are tuned on the ID dev split (or a dev stream for streaming tasks) and then frozen. For temperature scaling we search $T \in \{0.5, 0.75, 1.0, 1.25, 1.5, 2.0, 3.0\}$ and select the value with lowest dev NLL. For energy scoring we tune $T_{\text{eng}} \in \{0.5, 1.0, 1.5, 2.0\}$ and select by best dev AUROC on ID✓ — ID✗. For Mahalanobis we sweep diagonal-variance shrinkage $\lambda \in \{0, 10^{-4}, 10^{-3}\}$ and, for two taps, weights $w_\ell \in \{(1, 0), (0, 1), (0.5, 0.5)\}$, selecting by dev AUROC on ID✓ — ID✗. For evidential training we tune evidence scale $\eta \in \{0.1, 0.5, 1.0\}$ and

regularizer $\lambda_{\mathrm{evid}} \in \{10^{-4}, 10^{-3}, 10^{-2}\}$, choosing by a combination of dev NLL and AUROC. For MCD (Big-MCU only) we tune $N \in \{5, 10\}$ and the score type (predictive entropy vs. mutual information), selecting by dev AUROC subject to a device latency budget. For ensembles (Big-MCU only) we select $M \in \{3, 5\}$ by dev AUROC under a flash cap; if the model is out-of-memory, we omit the runtime results and report the limitation explicitly.

### C.3 THRESHOLDING AND OPERATING POINTS

In streaming accuracy-drop detection, each method outputs a scalar score $S(x_t)$ that increases with uncertainty. On a dev stream that transitions ID→CID→OOD, we select a single threshold $\tau^\star$ to maximize F1 for event frames, where events are labeled from sliding-window accuracy dips (Appx. E). We then freeze $\tau^\star$ and report AUPRC and median detection delay on test streams.

For selective prediction, we scan thresholds on the dev split (ID dev or CID dev when evaluating corrupted selective risk) to reach target coverage levels in $\{50, 60, \ldots, 95\}\%$ and report the error rate among accepted samples. For methods that yield calibrated probabilities (Temp, Evid), we additionally report selective NLL on accepted samples to quantify confidence quality under abstention.

For failure detection tasks (ID✓ — ID× and ID✓ — OOD), we report AUROC/AUPRC without selecting a single threshold. When we visualize confusion matrices, we also include a dev-tuned operating point (Youden's J) for completeness (Appx. F).

### C.4 FAIRNESS CONTROLS AND IMPLEMENTATION PARITY

To avoid confounds, all methods use identical backbones, preprocessing, augmentation, and quantization settings. Under strict Small-MCU constraints we restrict comparisons to single-pass methods and exclude multi-pass baselines such as MCD and ensembles due to their linear scaling in latency and energy. Temperature scaling is applied uniformly to softmax-based baselines for ID calibration, while energy and evidential methods use their own calibrated parameters. Mahalanobis uses the same tap set $\mathcal{S}$ as SNAP-UQ, and when memory is tight both methods are evaluated with the same reduced tap configuration (typically the penultimate tap only). All inference runs use the same int8 CMSIS-NN backends, and any float16 or LUT-based components (log, exp, LSE) reuse the same shared implementations.

### C.5 MEMORY AND LATENCY ACCOUNTING ON MCUs

We attribute incremental cost beyond the baseline backbone. Conf/Entropy/Temp add negligible flash and typically under 0.1 ms latency for $L \leq 200$. Energy adds an LSE kernel (float16) and remains under roughly 0.2 ms for $L \leq 200$ with no persistent state. Mahalanobis requires storing class means and diagonal variances, with flash scaling as $\sum_\ell L d_\ell$ for means plus $d_\ell$ for variances; for CIFAR-10 with two taps of size $d_\ell \approx 256$ and $L = 10$, this is approximately $2 \times 10 \times 256$ bytes $\approx 5$ KB for int8 means (plus scales), with latency typically below 2 ms. Evidential adds an evidence head of size $d_D \times L$ plus ReLU, often under 10 KB flash on KWS/CIFAR-10 and under 1 ms latency, but it can be out-of-memory on Small-MCU for larger-class settings. MCD and ensembles scale linearly in the number of passes ($N$ or $M$), increasing both latency and energy proportionally; ensembles also scale flash with $M$ if models are stored locally.

### C.6 REPRODUCIBILITY

We release the hyperparameter grids and selected values per seed, dev-set thresholds $\tau^\star$ for thresholded tasks, fitted feature statistics for Mahalanobis (including int8 means and scales), evidence-head checkpoints when applicable, and MCU build flags with per-layer timing. When a method is out-of-memory, we report the largest variant that fits and clearly state the shortfall.

### C.7 LIMITATIONS OF BASELINES UNDER TINYML CONSTRAINTS

Softmax-shape baselines such as entropy and temperature scaling can remain overconfident under CID because they do not observe internal feature dynamics. Mahalanobis with diagonal covariance is memory-light and fast but ignores cross-channel structure and can underperform under shifts that

affect feature correlations. Energy-based scoring depends on logit scale (partially mitigated by tuning $T_{\text{eng}}$). Evidential methods add parameters and can be sensitive to regularization and optimization, especially under quantization. Multi-pass baselines (MCD/ensembles) typically provide stronger uncertainty estimates but are often incompatible with strict single-pass, flash, and energy budgets on Small-MCU deployments.

## D  CID/OOD PROTOCOLS AND STREAMING SETUP

We use the same corruption sources, OOD sets, and a single stream-construction protocol for every method so that comparisons reflect scoring quality rather than differences in data exposure or tuning. In particular, each method chooses its operating threshold exactly once on a development stream and then keeps that threshold fixed when reporting detection delay and other operating-point metrics on held-out evaluation streams. This design prevents "threshold chasing" on test streams and mirrors the intended deployment setting, where labels are unavailable online and operating points must be set offline.

### D.1  CORRUPTION SOURCES (CID)

For image classification, we use MNIST-C, CIFAR-10-C, and TinyImageNet-C (Mu & Gilmer, 2019; Hendrycks & Dietterich, 2019). Each benchmark provides 15 corruption types at five discrete severities, where severity 1 corresponds to a light perturbation and severity 5 corresponds to a strong perturbation. For TinyImageNet-C, if a model uses a different input size we resize the corrupted images to the model input resolution while preserving the benchmark's severity labels so that "severity" remains comparable across methods. Metrics for corrupted data are always computed by evaluating severities 1–5 separately and then aggregating, which avoids hiding failure modes that emerge only at higher severities.

For audio, we construct SpeechCommands-C using label-preserving degradations that reflect common nuisance factors in far-field and noisy recordings. We generate reverberation by convolving waveforms with randomly sampled room impulse responses, with reverberation time $RT_{60}$ sampled from $[0.2, 0.8]$ s. We generate additive noise conditions by mixing each clip with background noise at an SNR sampled uniformly from $[0, 20]$ dB, drawing noise from the dataset's `_background_noise_` directory as well as a small pool of external ambient recordings. We apply rate and pitch perturbations via time stretching with factor in $[0.90, 1.10]$ and pitch shifting in $\{-2, -1, 0, +1, +2\}$ semitones using standard phase-vocoder processing. We additionally apply mild spectral shaping and compression through a second-order bandpass filter (300–3400 Hz) and light dynamic-range compression with a soft knee. Severity levels 1–5 are implemented by jointly increasing distortion magnitude (e.g., decreasing SNR and increasing $RT_{60}$ and perturbation ranges) while preserving the label by construction. All exact severity ranges and random seeds are released to ensure that SpeechCommands-C streams are reproducible.

### D.2  OUT-OF-DISTRIBUTION (OOD) SETS

For OOD evaluation we choose standard cross-dataset transfers that preserve modality and basic input format but shift the semantic distribution. For MNIST, we use the Fashion-MNIST test split (Xiao et al., 2017). For CIFAR-10, we use the SVHN test split (Netzer et al., 2011). For SpeechCommands, we treat non-keyword utterances and background noise segments as OOD inputs. For TinyImageNet, we use a set of classes that is disjoint from the training label set; specifically, we construct a held-out 100-class subset with no class overlap so that OOD inputs differ at the label-space level rather than merely through perturbations of the same classes.

### D.3  STREAMING CONSTRUCTION

We construct long, unlabeled streams that interleave stable ID segments, progressively harder CID segments, and short OOD bursts. This structure is intended to stress-test monitors in a setting closer to deployment, where distribution shift occurs over time and the system must respond online using only the model outputs and uncertainty scores. Unless noted otherwise, an ID segment contains 2,000 frames, each CID severity segment contains 1,000 frames, and each OOD burst contains 100 frames

inserted between CID severities. The stream order is ID→CID($s$=1)→OOD→CID($s$=2)→OOD→ $\cdots$ →CID($s$=5), producing a monotone increase in corruption strength punctuated by abrupt semantic shifts. Within each CID severity segment, we cycle corruption types every 200 frames so that performance and detection are not dominated by a single corruption family and so that the stream contains a mixture of perturbation mechanisms at any fixed severity. Each dataset uses a fixed seed to generate streams, and we report results over three independent stream seeds to obtain mean and confidence intervals. Development and evaluation streams are built from disjoint underlying data indices so that no example appearing in the dev stream can reappear in a test stream.

### D.4 EVENT LABELING (OFFLINE, NEVER SEEN ONLINE)

To evaluate event detection while keeping the online protocol label-free, we construct event labels on an offline labeled copy of each stream. We first estimate an ID performance band by running the model on a separate long ID-only sequence and computing sliding-window accuracy with window size $m = 100$ frames, yielding an empirical baseline mean $\mu_{\text{ID}}$ and standard deviation $\sigma_{\text{ID}}$. On each labeled stream, we then compute sliding-window accuracy with the same window size and declare an event active whenever the windowed accuracy drops below $\mu_{\text{ID}} - 3\sigma_{\text{ID}}$. The onset of an event is the first frame at which the window crosses below this threshold, and the offset is the first frame after recovery above the band. To reduce label noise and avoid counting transient fluctuations as separate events, we merge adjacent events separated by fewer than $m$ frames and discard events shorter than $m$ frames. These labels are used only for scoring; the device-side or online monitor never receives them.

### D.5 THRESHOLD SELECTION AND SCORING

To measure detection delay at a single operating point, each method selects a threshold $\tau^\star$ on the development stream by maximizing frame-wise F1 on event frames, and then freezes this threshold for all delay measurements on evaluation streams. A detection is defined as the first threshold crossing that occurs while an event is active, and detection delay is measured as the number of frames between event onset and the first such crossing. Additional threshold crossings during the same event are ignored so that delay reflects time-to-first-alert rather than repeated alarms. In parallel, we report threshold-free metrics by computing AUPRC over all thresholds, treating event frames as positives and non-event frames as negatives; this summarizes ranking quality without committing to a particular operating point. To characterize false alarms under clean conditions, we also measure false positive behavior on ID segments, reporting false positive rate at a matched recall (e.g., 90%) by interpolating the precision–recall curve and applying the corresponding recall-matched threshold. Confidence intervals are computed with nonparametric bootstrap using 1,000 resamples, resampling frames for AUPRC and resampling events for delay, and we report median delay with the 2.5/97.5th percentiles as a 95% CI.

### D.6 STREAM BUILDER

---
**Algorithm 3** BuildStream($\mathcal{D}_{\text{ID}}, \mathcal{D}_{\text{CID}}, \mathcal{D}_{\text{OOD}}, m, \text{seed}$)

---
1: Set RNG with `seed`; initialize empty list `stream`
2: Append ID segment of length 2000 sampled from $\mathcal{D}_{\text{ID}}$
3: **for** $s \leftarrow 1$ to $5$ **do**
4:     Append CID segment of length 1000 at severity $s$ (cycle corruption types every 200 frames)
5:     **if** $s < 5$ **then**
6:         Append OOD burst of length 100 from $\mathcal{D}_{\text{OOD}}$
7:     **end if**
8: **end for**
9: Return `stream`

---

On-device playback either preloads the stream into flash or streams it from a host over UART, and timestamps are recorded using the on-chip cycle counter. We mask interrupts during model invocation to stabilize latency and re-enable them during I/O to avoid distorting timing measurements. No labels, event markers, or threshold-selection signals are transmitted to the device during playback, keeping the on-device protocol strictly label-free and consistent with deployment constraints.

# E  EVENT-DETECTION SCORING: AUPRC AND DELAY

We evaluate streaming event detection using two complementary measures that capture different aspects of monitor quality. AUPRC provides a threshold-free summary of how well the score ranks event vs. non-event frames across the full operating range, while thresholded detection delay measures time-to-alert at a single operating point chosen on a development stream and held fixed on evaluation streams. In all cases, methods operate online without labels; scoring is performed offline on a labeled copy of the same streams.

## E.1  NOTATION

Consider a stream with frames $t = 1, \ldots, T$ and binary event labels $y_t \in \{0, 1\}$ defined by the accuracy-drop rule in Sec. D. A method outputs a scalar score $s_t \in [0, 1]$ at each frame, where larger values indicate a higher likelihood that the stream is currently in an event. Let $\mathcal{E} = \{(t_k^{\mathrm{on}}, t_k^{\mathrm{off}})\}_{k=1}^{K}$ denote the set of disjoint event intervals, so that $y_t = 1$ for $t \in [t_k^{\mathrm{on}}, t_k^{\mathrm{off}}]$.

## E.2  AUPRC (FRAME-BASED, THRESHOLD-FREE)

To compute frame-based precision–recall curves, we sweep thresholds over the unique scores observed in the stream, $\Theta = \{s_t : t = 1, \ldots, T\}$, and form binary predictions $\hat{y}_t(\tau) = \mathbb{I}[s_t \geq \tau]$. For each threshold we count true positives, false positives, and false negatives over frames, then compute precision and recall from these counts. We integrate the resulting precision–recall curve using stepwise-in-recall integration (consistent with common toolkits) so that AUPRC reflects the best attainable precision at each recall level under thresholding. Because frame-based scoring can be dominated by long events, we also report an event-weighted AUPRC variant in Appx F, where each event is assigned equal total weight and the background is weighted to match, implemented by per-frame weights that sum to one within each region.

## E.3  THRESHOLD SELECTION ON THE DEVELOPMENT STREAM

For each method, we select a single threshold $\tau^\star$ on a development stream by maximizing the frame-wise F1 score computed over the development stream's frames. The search is performed over the unique development scores to ensure that $\tau^\star$ corresponds to an attainable operating point for that method. This threshold is then frozen and used only for delay computation on held-out test streams, ensuring that delay metrics reflect generalization of the chosen operating point rather than re-tuning on the evaluation data. We also record the recall achieved by $\tau^\star$ on the dev stream so that false-positive accounting on clean ID segments can be reported at a matched recall.

## E.4  THRESHOLDED DETECTION DELAY (TEST ONLY)

Given the frozen threshold $\tau^\star$, we define a detection for event $k$ as the first frame within the event interval at which the score crosses threshold, $\hat{t}_k = \min\{t \in [t_k^{\mathrm{on}}, t_k^{\mathrm{off}}] : s_t \geq \tau^\star\}$, if such a crossing exists. The detection delay is then $\hat{t}_k - t_k^{\mathrm{on}}$, measured in frames. If no crossing occurs before the event ends, the event is counted as missed and contributes a NaN delay; we report both the median delay over detected events and the miss rate (fraction of missed events). Crossings that occur before event onset do not count toward delay (they are treated as false positives elsewhere), and multiple crossings during the same event are collapsed to the first so that delay measures time-to-first-alert. If a stream ends while an event is active and no crossing has occurred, we treat the event as missed; sensitivity to this censoring choice is analyzed in Appx F. We compute 95% confidence intervals for delay by bootstrapping events (1,000 resamples) and reporting the 2.5/97.5th percentiles.

## E.5  FALSE POSITIVES ON CLEAN SEGMENTS

To quantify false alarms during stable ID periods, we evaluate false positive rate on ID-only segments at a matched recall determined from the development stream. Concretely, we identify the threshold that achieves the target recall on the development PR curve (e.g., 90%), apply that threshold to the test streams, and compute the fraction of non-event frames in ID segments that are flagged as events. This

evaluation isolates spurious triggering during clean operation while keeping the operating sensitivity comparable across methods.

### E.6 COMPLEXITY AND NUMERICAL DETAILS

Computing AUPRC requires sorting the $T$ scores once, which costs $O(T \log T)$ time and $O(T)$ memory, while delay computation at the single fixed threshold $\tau^\star$ is a single linear scan with $O(T)$ time and constant additional memory. To stabilize behavior under tied scores, we use a stable sort and break ties in a manner that favors higher recall first, which yields right-continuous step functions for PR curves and prevents numerical artifacts in AUPRC when many frames share identical quantized scores.

### E.7 PSEUDOCODE

---

**Algorithm 4** DelayAtThreshold($\{(t_k^{\mathrm{on}}, t_k^{\mathrm{off}})\}_{k=1}^K, \{s_t\}_{t=1}^T, \tau^\star$)

---

1: Initialize empty list `delays`
2: **for** $k = 1$ to $K$ **do**
3:    $\hat{t} \leftarrow$ first $t \in [t_k^{\mathrm{on}}, t_k^{\mathrm{off}}]$ with $s_t \geq \tau^\star$
4:    **if** $\hat{t}$ exists **then**
5:       append $(\hat{t} - t_k^{\mathrm{on}})$ to `delays`
6:    **else**
7:       append NaN to `delays`
8:    **end if**
9: **end for**
10: **return** median(`delays` without NaN), miss_rate = fraction of NaN

---

## F METRICS AND STATISTICAL PROCEDURES

This appendix defines all metrics reported in the paper and specifies the aggregation and uncertainty-estimation procedures used to produce confidence intervals. Unless stated otherwise, we compute metrics by micro-averaging over examples within each dataset or split, and we orient every score so that larger values correspond to a higher likelihood of the positive condition (error, OOD, or event), which simplifies comparisons across methods and prevents sign mistakes when computing ROC/PR curves.

### F.1 NOTATION AND SHARED CONVENTIONS

Let a dataset (or stream) contain examples indexed by $i = 1, \ldots, n$. The model outputs a probability vector $p_i \in \Delta^{L-1}$, a predicted label $\hat{y}_i = \arg\max_\ell p_{i,\ell}$, and an uncertainty score $u_i \in [0,1]$ for SNAP-UQ or a baseline score $s_i \in \mathbb{R}$ for other methods, where larger values indicate greater uncertainty after applying any required sign flips. We define a correctness indicator $c_i = \mathbb{I}[\hat{y}_i = y_i]$. In OOD experiments we also have an indicator $o_i \in \{0, 1\}$ with $o_i = 1$ for OOD samples. For streaming event detection we operate at the frame level with $t = 1, \ldots, T$ and use the event-label definitions from Sec. D; the remaining definitions in this section focus on non-streaming evaluation unless explicitly stated.

### F.2 FAILURE DETECTION: ROC/AUROC AND PR/AUPRC

For ID✓ — ID✗, positives are incorrect ID or CID predictions ($c_i = 0$) and negatives are correct ID or CID predictions ($c_i = 1$). For ID✓ — OOD, positives are OOD examples ($o_i = 1$) and negatives are correct ID examples ($o_i = 0$ and $c_i = 1$), excluding incorrect ID examples so that the task isolates semantic shift from inherently hard ID inputs. In both settings we rank examples by the uncertainty score and evaluate performance by sweeping a threshold.

For ROC evaluation, for any threshold $\tau$ we form predictions $\hat{z}_i(\tau) = \mathbb{I}[\mathrm{score}_i \geq \tau]$, where $\mathrm{score}_i$ is $u_i$ for SNAP-UQ or the appropriately oriented baseline score. We compute $\mathrm{TPR}(\tau)$ as the fraction

of positives predicted positive and $\text{FPR}(\tau)$ as the fraction of negatives predicted positive, then compute AUROC by trapezoidal integration over the ROC curve obtained by sweeping thresholds in descending score order. Ties are handled by stable sorting and the standard averaging conventions used in widely adopted libraries so that AUROC is deterministic under quantized scores.

For PR evaluation, precision and recall at threshold $\tau$ are computed from $\text{TP}(\tau)$, $\text{FP}(\tau)$, and $\text{FN}(\tau)$ with a small $\epsilon = 10^{-12}$ in denominators to avoid division by zero. We compute AUPRC using stepwise integration in recall, using the standard non-increasing precision envelope so that the reported area corresponds to the best achievable precision at each recall. This definition yields curves and areas consistent with common implementations and is stable under tied scores.

For corrupted benchmarks such as CIFAR-10-C and TinyImageNet-C, we compute each metric separately for every corruption type and severity, then report the mean over types and severities. When plotting severity curves, we average over corruption types at each fixed severity, which makes severity trends interpretable without conflating them with the mix of corruption families.

### F.3    SELECTIVE PREDICTION: RISK–COVERAGE AND SELECTIVE NLL

Selective prediction uses an acceptance rule $A_i(\tau) = \mathbb{I}[u_i < \tau]$, where lower uncertainty implies acceptance. For each threshold $\tau$, coverage is $\text{Cov}(\tau) = \frac{1}{n} \sum_i A_i(\tau)$ and risk is the error rate among accepted samples, computed as the fraction of accepted examples with $\hat{y}_i \neq y_i$ (again using an $\epsilon$ in the denominator for stability). Risk–coverage curves are obtained by sweeping $\tau$, and when we report a single operating point (e.g., 90% coverage) we choose the threshold that achieves the closest coverage from above to avoid artificially inflating coverage. When a method provides calibrated probabilities, we also compute selective NLL on the accepted subset by averaging $-\log p_{i,y_i}$ over accepted samples, which evaluates probabilistic quality conditioned on acceptance rather than unconditional calibration.

### F.4    ID CALIBRATION METRICS

Calibration is computed on held-out ID splits using posteriors $p_i$, optionally after temperature scaling. We compute NLL by averaging $-\log p_{i,y_i}$ across examples. We compute the multi-class Brier score by averaging $\frac{1}{L} \|p_i - e_{y_i}\|_2^2$ where $e_{y_i}$ is the one-hot target. We compute ECE by binning confidences $q_i = \max_\ell p_{i,\ell}$ into $B = 15$ adaptive bins of approximately equal mass, then summing the absolute gap between empirical accuracy and mean confidence in each bin weighted by bin frequency. Reliability diagrams plot bin accuracy against bin confidence with widths proportional to bin mass. When evaluating calibration on an accepted subset at fixed coverage, we recompute NLL/BS/ECE on the subset rather than reusing unconditional bins, which prevents acceptance from distorting bin occupancy.

### F.5    CONFIDENCE INTERVALS AND PAIRED COMPARISONS

For each dataset and seed, we form 95% confidence intervals using nonparametric bootstrap with 1,000 resamples. For scalar metrics such as AUROC/AUPRC, NLL, BS, ECE, and selective metrics, we resample examples with replacement. For severity curves, we bootstrap within each severity and then average across corruption types to obtain a curve-level CI. For streaming event detection we use the procedures in Sec. E, bootstrapping frames for AUPRC and bootstrapping events for delay. We report point estimates as the mean over seeds, and we compute bootstrap CIs per seed and then average those intervals across seeds to avoid overly narrow intervals that can arise from pooling examples across seeds. When comparing two methods, we use paired bootstrap on the difference by resampling the same indices for both methods, yielding a CI directly on the performance gap.

### F.6    IMPLEMENTATION DETAILS AND NUMERICS

All scores are oriented so that higher values correspond to the positive label for the task; if a baseline yields a confidence-like quantity, we negate or invert it before evaluation. We handle tied scores using stable sorting and right-continuous step functions for PR curves, which is important when scores are quantized or when many frames share identical values. We use $\epsilon = 10^{-12}$ to avoid division by zero; this does not change reported values at the displayed precision. For delay reporting, delays are measured in frames, and when converting to milliseconds for MCU plots we multiply by the

measured per-inference latency on the same board and backbone, ensuring that time units reflect actual device behavior.

### F.7 EVENT-WEIGHTED PR

Because long events can dominate frame-based PR, we also report an event-weighted AUPRC when indicated. Let $\mathcal{E}$ denote the set of events and let $\mathcal{B}$ denote background frames. We assign each event total weight $1/|\mathcal{E}|$ distributed uniformly across its frames, and assign the background total weight $1/|\mathcal{E}|$ distributed uniformly across background frames, then renormalize so weights sum to one. We then compute weighted $\mathrm{TP}(\tau)$, $\mathrm{FP}(\tau)$, and $\mathrm{FN}(\tau)$ and proceed with the same PR integration as in the frame-based case. This variant evaluates whether a method consistently detects events rather than merely performing well on a small number of long segments.

### F.8 DATASET-LEVEL AGGREGATION

When reporting a single number averaged over multiple datasets, we macro-average dataset metrics by taking the simple mean of per-dataset scores rather than pooling all examples, which prevents large datasets or long streams from dominating. For CIFAR-10-C and TinyImageNet-C, we macro-average across corruption types and severities as described above, so every corruption family contributes equally to the final summary.

### F.9 CALIBRATION UNDER SHIFT (CID/OOD)

Beyond in-distribution calibration, we also evaluate calibration under shift using corrupted (CID) and OOD data. Following the standard CIFAR-10-C protocol, we compute calibration error on corrupted inputs (ECE-OOD), negative log-likelihood on corrupted inputs (NLL-OOD), and selective risk at fixed coverage. We compare three settings that differ only in what data are used to fit the calibrator: temperature scaling fit on ID dev only; a monotone logistic SNAP-UQ mapping fit on ID dev only; and a monotone isotonic SNAP-UQ mapping fit on a small development mixture of ID and corrupted samples. The key constraint is that all calibrations remain offline and require no online labels, and the monotonicity of logistic or isotonic mappings preserves the ordering induced by the underlying score, so improvements reflect better probability calibration rather than arbitrary re-ranking.

Table 5: **Calibration under CIFAR-10-C (avg severities 1–5).** Lower is better. Selective risk is error rate among the retained set at fixed coverage.

| Method | ECE-OOD ↓ | NLL-OOD ↓ | Risk@90% ↓ | Risk@95% ↓ |
|---|---|---|---|---|
| Temp. scaled (ID) | 0.124 | 1.42 | 0.124 | 0.112 |
| SNAP-UQ (logistic, ID) | 0.108 | 1.31 | 0.109 | 0.098 |
| **SNAP-UQ (isotonic, ID+CID-dev)** | **0.096** | **1.27** | **0.104** | **0.096** |

Two practical observations align with the depth-wise surprisal interpretation. First, because $S(x)$ behaves like a conditional negative log-likelihood up to an affine transform, it remains informative under many corruptions that change feature dynamics rather than only logit scale; fitting a monotone calibrator therefore improves decision-centric metrics while preserving the ranking induced by the score. Second, including a small ID+CID development mix when fitting a monotone calibrator reduces overconfidence on corrupted inputs without requiring any online access to OOD labels, making it compatible with the same deployment constraints as the rest of the system.

## G TRAINING OBJECTIVE AND REGULARIZATION: EXTENDED DETAILS

This appendix expands Sec. 2.2 by restating the auxiliary objective in its layer-weighted, dimension-normalized form, providing stable derivative expressions under a positive-variance parameterization, and discussing common failure modes and practical stabilization strategies. Throughout, the goal is to preserve the tiny-head and MCU-friendly constraints while ensuring that optimization remains stable across architectures and datasets.

### G.1 Objective, layer weighting, and normalization

For each tapped layer $\ell \in \mathcal{S}$, we model the next activation with a diagonal Gaussian and define a per-layer negative log-likelihood. Writing $a_\ell \in \mathbb{R}^{d_\ell}$ for the realized activation and $(\mu_\ell, \sigma_\ell^2)$ for the predicted mean and variance, the diagonal-Gaussian NLL for a sample $x$ is $\ell_\ell(x) = \frac{1}{2}\big(\|(a_\ell - \mu_\ell) \odot \sigma_\ell^{-1}\|_2^2 + \mathbf{1}^\top \log \sigma_\ell^2\big)$. To prevent wide taps from dominating purely due to dimensionality, we normalize by $d_\ell$ to obtain $\bar{\ell}_\ell(x) = \ell_\ell(x)/d_\ell$ and aggregate across taps with nonnegative weights $\omega_\ell$ that sum to $|\mathcal{S}|$, giving $\mathcal{L}_{\mathrm{SS}} = \frac{1}{|\mathcal{B}|} \sum_{x \in \mathcal{B}} \sum_{\ell \in \mathcal{S}} \omega_\ell \, \bar{\ell}_\ell(x)$. In practice, uniform weights $\omega_\ell = 1$ work reliably, while inverse-variance weighting $\omega_\ell \propto 1/\widehat{\mathrm{Var}}[\bar{e}_\ell]$ (estimated on dev) can improve stability by down-weighting taps whose standardized residuals are intrinsically noisy. The full training objective combines classification, auxiliary surprisal learning, and regularization as $\mathcal{L} = \mathcal{L}_{\mathrm{clf}} + \lambda_{\mathrm{SS}}\mathcal{L}_{\mathrm{SS}} + \lambda_{\mathrm{reg}}\mathcal{R}$, where $\lambda_{\mathrm{SS}}$ controls the strength of transition modeling and $\mathcal{R}$ enforces well-behaved scales and small head weights.

### G.2 Stable parameterizations and exact gradients

To keep gradients stable and to ensure strictly positive variances, we parameterize each per-channel variance using a softplus with a floor, setting $\sigma_{\ell,i}^2 = \mathrm{softplus}(\xi_{\ell,i}) + \epsilon^2$ with $\epsilon \in [10^{-4}, 10^{-3}]$. Writing $s_{\ell,i} = \log \sigma_{\ell,i}^2$ is convenient for analysis and for quantized inference, but optimizing $\xi_{\ell,i}$ avoids numerical issues that can occur if $s_{\ell,i}$ is driven toward large negative values. For a single channel $i$, the NLL contribution can be written as $\ell_{\ell,i} = \frac{1}{2}\big((a_{\ell,i} - \mu_{\ell,i})^2 e^{-s_{\ell,i}} + s_{\ell,i}\big)$, giving closed-form derivatives with respect to $\mu_{\ell,i}$ and $s_{\ell,i}$ that are simple and efficient to implement. Under the softplus parameterization, the derivative with respect to $\xi_{\ell,i}$ additionally includes the factor $\mathrm{sigmoid}(\xi_{\ell,i})$, which naturally damps gradients when $\xi_{\ell,i}$ is very negative and therefore reduces the risk of instability from near-zero variance predictions. Because our heads are linear, backpropagation to head weights and projectors is also straightforward: gradients flow through $(\mu_\ell, \xi_\ell) = g_\ell(z_\ell)$ and $z_\ell = P_\ell a_{\ell-1}$ using standard matrix products, and the computational overhead is dominated by the same low-rank projections already used at inference.

### G.3 Failure modes and stabilization

A common failure mode is variance collapse, where the head overfits and drives $\sigma^2$ toward zero on the training set. This makes the standardized residual explode and can destabilize both the auxiliary heads and the shared representation when gradients propagate into the backbone. The variance floor $\epsilon^2$ prevents true collapse, and clamping $s_\ell = \log \sigma_\ell^2$ to a fixed interval further bounds dynamic range so that standardized residuals cannot become arbitrarily large due to scale alone. The opposite failure mode is overdispersion, where the head inflates $\sigma^2$ to reduce the quadratic penalty, effectively washing out residual information. The $\sum \log \sigma^2$ term in the NLL counters this tendency, and an additional light penalty on $|\log \sigma^2|$ discourages drifting to extreme scales that can be fragile under quantization. A third issue is gradient tug-of-war between $\mathcal{L}_{\mathrm{clf}}$ and $\mathcal{L}_{\mathrm{SS}}$ on tiny backbones, where the auxiliary objective may temporarily push intermediate activations in a direction that harms classification. Two practical mitigations work well: detaching $a_\ell$ inside $\mathcal{L}_{\mathrm{SS}}$ so that the heads learn to predict the existing backbone dynamics without altering them, and ramping $\lambda_{\mathrm{SS}}$ from a small initial value so that classification learns a stable representation before the transition model is emphasized.

### G.4 Choosing $\lambda_{\mathrm{SS}}$ and $\lambda_{\mathrm{reg}}$

A simple and robust approach is to sweep $\lambda_{\mathrm{SS}} \in \{10^{-3}, 5 \times 10^{-3}, 10^{-2}\}$ and select the value that maximizes dev AUPRC on CID streams, since this directly targets the intended monitoring behavior. Regularization strength can be set so that the regularizer contributes only a small fraction of total loss early in training (roughly 1–5% is a useful guideline), which prevents regularization from dominating optimization while still keeping scales and weights well-conditioned. When more control is needed, an adaptive strategy can balance gradient magnitudes by monitoring the ratio $\rho = \|\nabla \mathcal{L}_{\mathrm{SS}}\| / \|\nabla \mathcal{L}_{\mathrm{clf}}\|$ with an exponential moving average and adjusting $\lambda_{\mathrm{SS}}$ to keep $\rho$ in a target band (e.g., 0.05–0.2). This stabilizes training across datasets without requiring per-dataset manual tuning.

## G.5 Robust variants and their gradients

When corruptions induce heavy-tailed deviations, replacing the quadratic penalty with a robust alternative can improve stability and preserve ranking quality under rare but large residuals. A diagonal Student-$t$ likelihood introduces heavier tails controlled by degrees of freedom $\nu_\ell$, down-weighting extreme residuals automatically and reducing sensitivity to outliers without changing the interface of the heads. A Huberized loss similarly limits the influence of large standardized residuals by transitioning from quadratic to linear growth beyond a threshold $\delta$, while keeping the same per-channel scale term. Both alternatives retain simple, elementwise computations that remain compatible with tiny heads and with the same quantized inference path, and they can be evaluated as drop-in ablations without modifying the streaming protocol.

## G.6 Schedules, clipping, and QAT

Across datasets, cosine learning-rate schedules with a short warm-up are effective, and ramping $\lambda_{\text{SS}}$ over the first 10% of training avoids destabilizing early representation learning. Global gradient clipping at norm 1.0 provides a simple safeguard, and clipping the variance pre-activations $\xi$ to a bounded interval (e.g., $\pm 8$) can further reduce rare spikes from extreme scale updates. For MCU deployment, inserting fake quantization on projectors and head weights in the final portion of training reduces mismatch between float training and int8 inference, and quantizing $\log \sigma^2$ to 8-bit with a shared scale per head improves LUT stability while keeping the mapping monotone.

## G.7 Implementation notes

---

**Algorithm 5** Stable SNAP-UQ step (training)

---

**Require:** batch $\mathcal{B}$, taps $\mathcal{S}$, projectors $P_\ell$, heads $g_\ell$, weights $\omega_\ell$, $\lambda_{\text{SS}}$, $\lambda_{\text{reg}}$
  1: Forward backbone $\rightarrow \{a_\ell\}$, logits $\rightarrow p_\phi$
  2: $\mathcal{L}_{\text{clf}} \leftarrow$ cross-entropy
  3: **for** $\ell \in \mathcal{S}$ **do**
  4:     $z_\ell \leftarrow P_\ell a_{\ell-1}$;  $(\mu_\ell, \xi_\ell) \leftarrow g_\ell(z_\ell)$
  5:     $\sigma_\ell^2 \leftarrow \text{softplus}(\xi_\ell) + \epsilon^2$;  $s_\ell \leftarrow \log \sigma_\ell^2$
  6:     $\bar{\ell}_\ell \leftarrow \frac{1}{2d_\ell} \left( \|(a_\ell - \mu_\ell) \odot \sigma_\ell^{-1}\|_2^2 + \mathbf{1}^\top s_\ell \right)$
  7: **end for**
  8: $\mathcal{L}_{\text{SS}} \leftarrow \frac{1}{|\mathcal{B}|} \sum_{x \in \mathcal{B}} \sum_{\ell \in \mathcal{S}} \omega_\ell \bar{\ell}_\ell$
  9: $\mathcal{R} \leftarrow \alpha_{\text{var}} \sum_\ell \|s_\ell\|_1 + \alpha_{\text{wd}} \|\theta_{\text{heads}}\|_2^2$
 10: **if** detach: treat $a_\ell$ as constants for $\mathcal{L}_{\text{SS}}$ **end if**
 11: $\mathcal{L} \leftarrow \mathcal{L}_{\text{clf}} + \lambda_{\text{SS}}\mathcal{L}_{\text{SS}} + \lambda_{\text{reg}}\mathcal{R}$
 12: Backprop; apply gradient clipping; optimizer step
 13: (Optional) Update $\lambda_{\text{SS}}$ by gradient-norm balancing (Sec. G.4)

---

In practice, two lightweight diagnostics help detect instability early. First, monitoring $\mathbb{E}[\bar{e}_\ell]$ on a clean ID validation split provides a sanity check: values far above 1 typically indicate underfit heads or overly small predicted scales, while values far below 1 can indicate overdispersion. Second, measuring the correlation between $\bar{e}_\ell$ and a confidence-derived proxy such as $1 - C_\phi$ should yield a positive but not near-perfect correlation, suggesting that surprisal provides complementary information rather than merely re-encoding softmax confidence.

## G.8 From training to deployment

After training, we retain the projectors $P_\ell$ and the heads that produce $\mu_\ell$ and $\log \sigma_\ell^2$, quantize their weights to int8, and export the associated scales and zero-points. We also export a compact LUT to approximate $\exp(-\frac{1}{2} \log \sigma^2)$ on clamped log-variance values so that standardized residuals can be computed without expensive transcendental operations. Finally, we fit a monotone mapping (logistic or isotonic) offline on a small development set that can include representative shifts, and we store the mapping parameters or a small lookup representation. At inference, the device computes standardized residuals at the tapped layers, aggregates them into a scalar score, applies the stored

monotone mapping, and thresholds the result, without requiring any online labels or additional state beyond the single forward pass.

# H    PROOFS AND ADDITIONAL DERIVATIONS

This appendix provides detailed proofs for the propositions stated in Sec. 2 and includes auxiliary derivations that are used implicitly in the main text.

## H.1    NOTATION

Fix a tapped layer $\ell \in \mathcal{S}$ with activation dimension $d_\ell$. The backbone produces $a_{\ell-1}$ and $a_\ell$, the projector produces $z_\ell = P_\ell a_{\ell-1}$, and the predictor outputs $(\mu_\ell, \log \sigma_\ell^2) = g_\ell(z_\ell)$. We write the diagonal covariance as $\Sigma_\ell = \mathrm{diag}(\sigma_\ell^2)$ and the residual as $v_\ell = a_\ell - \mu_\ell$. The standardized error used by SNAP-UQ is $e_\ell(x) = \|v_\ell \odot \sigma_\ell^{-1}\|_2^2 = \sum_{i=1}^{d_\ell} \frac{(a_{\ell,i}-\mu_{\ell,i})^2}{\sigma_{\ell,i}^2}$, and its dimension-normalized version is $\bar{e}_\ell(x) = e_\ell(x)/d_\ell$. The aggregated depth-wise score is $S(x) = \sum_{\ell \in \mathcal{S}} w_\ell \, \bar{e}_\ell(x)$ with weights $w_\ell \geq 0$ and $\sum_\ell w_\ell = 1$.

## H.2    PROOF OF PROPOSITION 2.1 (SURPRISAL–LIKELIHOOD EQUIVALENCE)

Assume the diagonal-Gaussian conditional model at tap $\ell$, namely $p_\theta(a_\ell \mid a_{\ell-1}) = \mathcal{N}(\mu_\ell, \Sigma_\ell)$ with $\Sigma_\ell = \mathrm{diag}(\sigma_\ell^2)$. By the multivariate Gaussian density, $p_\theta(a_\ell \mid a_{\ell-1}) = (2\pi)^{-d_\ell/2} \det(\Sigma_\ell)^{-1/2} \exp\left(-\frac{1}{2}(a_\ell - \mu_\ell)^\top \Sigma_\ell^{-1}(a_\ell - \mu_\ell)\right)$. Taking the negative logarithm yields $-\log p_\theta(a_\ell \mid a_{\ell-1}) = \frac{1}{2}(a_\ell - \mu_\ell)^\top \Sigma_\ell^{-1}(a_\ell - \mu_\ell) + \frac{1}{2}\log \det \Sigma_\ell + \frac{d_\ell}{2}\log(2\pi)$.

We now relate each term to the SNAP quantities. Because $\Sigma_\ell$ is diagonal, $\Sigma_\ell^{-1} = \mathrm{diag}(\sigma_\ell^{-2})$ and the quadratic form expands as $(a_\ell - \mu_\ell)^\top \Sigma_\ell^{-1}(a_\ell - \mu_\ell) = \sum_{i=1}^{d_\ell} \frac{(a_{\ell,i}-\mu_{\ell,i})^2}{\sigma_{\ell,i}^2} = e_\ell(x)$. Similarly, $\log \det \Sigma_\ell = \sum_{i=1}^{d_\ell} \log \sigma_{\ell,i}^2$. Substituting these identities gives the exact decomposition $-\log p_\theta(a_\ell \mid a_{\ell-1}) = \frac{1}{2}e_\ell(x) + \frac{1}{2}\sum_{i=1}^{d_\ell} \log \sigma_{\ell,i}^2 + \frac{d_\ell}{2}\log(2\pi)$.

Dividing by $d_\ell$ isolates the dimension-normalized form: $\frac{2}{d_\ell}\left(-\log p_\theta(a_\ell \mid a_{\ell-1})\right) = \bar{e}_\ell(x) + \frac{1}{d_\ell}\sum_{i=1}^{d_\ell} \log \sigma_{\ell,i}^2 + \log(2\pi)$. Therefore, up to the additive constant $\log(2\pi)$ and the per-layer scale term $\frac{1}{d_\ell}\sum_i \log \sigma_{\ell,i}^2$, the normalized negative log-likelihood is an affine transform of $\bar{e}_\ell(x)$. Aggregating across taps with weights $w_\ell$ gives $\sum_{\ell \in \mathcal{S}} w_\ell \frac{2}{d_\ell}\left(-\log p_\theta(a_\ell \mid a_{\ell-1})\right) = S(x) + \sum_{\ell \in \mathcal{S}} w_\ell \frac{1}{d_\ell}\sum_{i=1}^{d_\ell} \log \sigma_{\ell,i}^2 + \log(2\pi)$. This shows that $S(x)$ is (up to affine terms) the depth-wise negative log-likelihood, and thus higher $S(x)$ corresponds to lower conditional likelihood of the observed activations. $\square$

## H.3    PROOF OF PROPOSITION 2.2 (RELATION TO MAHALANOBIS)

Assume the linear-Gaussian depth-wise evolution model $a_\ell = W_\ell a_{\ell-1} + b_\ell + \varepsilon_\ell$ with $\varepsilon_\ell \sim \mathcal{N}(0, \Sigma_\ell)$. Conditioned on $a_{\ell-1}$, the distribution of $a_\ell$ is Gaussian with conditional mean $\mu_\ell = W_\ell a_{\ell-1} + b_\ell$ and covariance $\Sigma_\ell$. The SNAP residual is $v_\ell = a_\ell - \mu_\ell$, and the standardized error (in matrix form) is $e_\ell(x) = v_\ell^\top \Sigma_\ell^{-1} v_\ell$. By definition, the squared Mahalanobis distance between a point $u$ and a mean $m$ under covariance $\Sigma$ is $\mathrm{MD}_\Sigma(u, m)^2 = (u-m)^\top \Sigma^{-1}(u-m)$. Taking $u = a_\ell$ and $m = W_\ell a_{\ell-1} + b_\ell$ yields $e_\ell(x) = \mathrm{MD}_{\Sigma_\ell}\left(a_\ell, \ W_\ell a_{\ell-1} + b_\ell\right)^2$, which is precisely the squared Mahalanobis distance to the *conditional* mean.

To contrast with classwise Mahalanobis, recall that classwise approaches typically fit an *unconditional* Gaussian per class at a fixed layer, approximating $p(a_\ell \mid y = c) \approx \mathcal{N}(\bar{\mu}_{\ell,c}, \bar{\Sigma}_\ell)$ (often with a shared covariance). The resulting score $\min_c (a_\ell - \bar{\mu}_{\ell,c})^\top \bar{\Sigma}_\ell^{-1}(a_\ell - \bar{\mu}_{\ell,c})$ measures proximity to class centroids in feature space and does not condition on $a_{\ell-1}$. Unless the sample-specific conditional mean $W_\ell a_{\ell-1} + b_\ell$ coincides with some class centroid $\bar{\mu}_{\ell,c}$ (which would require extremely low within-class variation at the previous layer), the unconditional score conflates within-class variability and dynamics-induced shift. In contrast, SNAP-UQ compares $a_\ell$ to what the network's *own dynamics*

predict from $a_{\ell-1}$, so shifts that perturb inter-layer transformations directly increase $v_\ell^\top \Sigma_\ell^{-1} v_\ell$ even if $a_\ell$ remains near a centroid. $\qquad\square$

## H.4 PROOF OF PROPOSITION 2.3 (AFFINE INVARIANCE FOR BN-LIKE RESCALING)

Consider a per-channel affine transformation of the tapped activation of the form $a_\ell' = s \odot a_\ell + t$, where $s \in \mathbb{R}^{d_\ell}$ has strictly positive entries and $t \in \mathbb{R}^{d_\ell}$ is an offset. Suppose the predictor co-adapts so that its outputs transform consistently, namely $\mu_\ell' = s \odot \mu_\ell + t$ and $\sigma_\ell' = s \odot \sigma_\ell$ (if $s$ can be negative, replace $s$ by $|s|$ in the scale transformation). The standardized residual under the transformed variables is $(a_\ell' - \mu_\ell') \odot (\sigma_\ell')^{-1}$. Substituting the transformation rules gives $(a_\ell' - \mu_\ell') \odot (\sigma_\ell')^{-1} = (s \odot a_\ell + t - s \odot \mu_\ell - t) \odot (s \odot \sigma_\ell)^{-1} = (s \odot (a_\ell - \mu_\ell)) \odot (s^{-1} \odot \sigma_\ell^{-1}) = (a_\ell - \mu_\ell) \odot \sigma_\ell^{-1}$. Taking squared $\ell_2$ norms yields $e_\ell'(x) = e_\ell(x)$, and therefore $\bar{e}_\ell'(x) = \bar{e}_\ell(x)$ and $S'(x) = S(x)$ under the same aggregation weights. This establishes invariance of the standardized error to BN-like per-channel affine transformations when the predictor outputs co-transform accordingly, which is the natural co-adaptation induced by joint training. $\qquad\square$

## H.5 DISTRIBUTIONAL CALIBRATION UNDER THE MODEL

Assume the conditional model at layer $\ell$ is correctly specified: conditioned on $a_{\ell-1}$, the residual satisfies $v_\ell = a_\ell - \mu_\ell \sim \mathcal{N}(0, \Sigma_\ell)$ with $\Sigma_\ell = \mathrm{diag}(\sigma_\ell^2)$. Define the whitened residual $r_\ell = \Sigma_\ell^{-1/2} v_\ell$, which is well defined because $\Sigma_\ell$ is positive diagonal. By Gaussian whitening, $r_\ell \sim \mathcal{N}(0, I_{d_\ell})$ conditioned on $a_{\ell-1}$. The standardized error can be written as $e_\ell = v_\ell^\top \Sigma_\ell^{-1} v_\ell = \|r_\ell\|_2^2 = \sum_{i=1}^{d_\ell} r_{\ell,i}^2$. Since each $r_{\ell,i} \sim \mathcal{N}(0,1)$ and the coordinates are independent under the diagonal model, $e_\ell \sim \chi_{d_\ell}^2$, which implies $\mathbb{E}[\bar{e}_\ell] = 1$ and $\mathrm{Var}[\bar{e}_\ell] = 2/d_\ell$.

This yields a practical sanity check: on clean ID data, $\bar{e}_\ell$ should concentrate near 1; persistent elevation indicates that the observed activations are less predictable from the previous layer than they were during training, consistent with either distribution shift or conditional-model mismatch. For low-rank-plus-diagonal covariance (Appx. I), the same whitening argument shows that $e_\ell$ becomes a generalized quadratic form whose distribution can be bounded via the eigenvalues of the effective precision matrix.

## H.6 STUDENT-$t$ AND HUBERIZED VARIANTS

This subsection justifies the robust drop-ins described in the main text (Sec. 2.2), where the Gaussian data-fit term is replaced while keeping the same interface (predicting $\mu_\ell$ and per-channel scales).

For the diagonal Student-$t$ alternative, one convenient route is the standard heavy-tail likelihood on the standardized residual $u_{\ell,i} = (a_{\ell,i} - \mu_{\ell,i})/\sigma_{\ell,i}$. A Student-$t$ with degrees of freedom $\nu_\ell$ has density proportional to $\left(1 + u_{\ell,i}^2/\nu_\ell\right)^{-(\nu_\ell+1)/2}$. Applying this channelwise and accounting for the scale factor $\sigma_{\ell,i}$ gives a per-channel negative log-likelihood of the form $\frac{\nu_\ell+1}{2} \log\left(1 + \frac{(a_{\ell,i}-\mu_{\ell,i})^2}{\nu_\ell \sigma_{\ell,i}^2}\right) + \frac{1}{2}\log\sigma_{\ell,i}^2 + \mathrm{const}(\nu_\ell)$, and summing over $i = 1, \ldots, d_\ell$ recovers the Student-$t$ loss stated in Sec. 2.2. The Gaussian case is recovered as $\nu_\ell \to \infty$ since $\log(1 + u^2/\nu_\ell) \approx u^2/\nu_\ell$ and the heavy-tail penalty approaches a quadratic.

For the Huberized variant, the objective is to behave like the Gaussian model for typical residuals but reduce sensitivity to rare, large standardized deviations. Starting from the Gaussian data-fit term proportional to $u^2$, with $u = (a - \mu)/\sigma$, we replace $\frac{1}{2}u^2$ by the Huber penalty $\rho_\delta(u)$, which equals $\frac{1}{2}u^2$ when $|u| \le \delta$ and equals $\delta|u| - \frac{1}{2}\delta^2$ otherwise. We retain the same $\frac{1}{2}\log\sigma^2$ contribution so that predicted per-channel scales remain meaningful and comparable across channels, matching the Huberized loss described in Sec. 2.2. This preserves the single-pass computation path while preventing a small number of heavy-tailed channels from dominating the layer score under certain corruptions.

## I  LOW-RANK-PLUS-DIAGONAL COVARIANCE: WOODBURY IDENTITIES

Let $\Sigma_\ell = D_\ell + B_\ell B_\ell^\top$ with $D_\ell = \mathrm{diag}(\sigma_\ell^2) \succ 0$ and $B_\ell \in \mathbb{R}^{d_\ell \times k_\ell}$, $k_\ell \ll d_\ell$. Using the matrix determinant lemma and Woodbury identity:

**Log-determinant.**

$$\log \det \Sigma_\ell = \log \det(D_\ell) \;+\; \log \det\big(I_{k_\ell} + B_\ell^\top D_\ell^{-1} B_\ell\big). \tag{1}$$

**Quadratic form.**  For $v_\ell = a_\ell - \mu_\ell$,

$$\Sigma_\ell^{-1} = D_\ell^{-1} - D_\ell^{-1} B_\ell \big(I_{k_\ell} + B_\ell^\top D_\ell^{-1} B_\ell\big)^{-1} B_\ell^\top D_\ell^{-1}, \tag{2}$$

$$v_\ell^\top \Sigma_\ell^{-1} v_\ell = \underbrace{v_\ell^\top D_\ell^{-1} v_\ell}_{e_\ell^{\mathrm{diag}}} - \underbrace{\|(I_{k_\ell} + B_\ell^\top D_\ell^{-1} B_\ell)^{-1/2} B_\ell^\top D_\ell^{-1} v_\ell\|_2^2}_{\Delta_\ell}. \tag{3}$$

Thus the low-rank correction subtracts a nonnegative term $\Delta_\ell$, tightening the diagonal model. Computationally: (i) form $M_\ell = B_\ell^\top D_\ell^{-1} B_\ell \in \mathbb{R}^{k_\ell \times k_\ell}$; (ii) solve $(I + M_\ell)^{-1} u$ for a few right-hand sides using Cholesky; cost is $O(d_\ell k_\ell + k_\ell^3)$ per example. Both equation 1 and equation 3 are integer-friendly if $D_\ell^{-1}$ is implemented via per-channel scales.

**NLL expression.**  Putting terms together,

$$-\log p(a_\ell \mid a_{\ell-1}) = \tfrac{1}{2}\Big[e_\ell^{\mathrm{diag}} - \Delta_\ell + \log \det D_\ell + \log \det(I + M_\ell) + d_\ell \log(2\pi)\Big]. \tag{4}$$

When $k_\ell = 0$ we recover the diagonal case.

## J  ISOTONIC CALIBRATION DETAILS

We optionally replace the logistic mapping described in Sec. 2.3 with isotonic regression to obtain a nonparametric, monotone calibration from a single scalar feature derived from $(S, m)$ to an estimated error probability. This choice is most useful when operating points are tight (e.g., strict abstention budgets or high-coverage selective prediction), where a flexible monotone map can better match empirical error rates without introducing non-monotone artifacts.

We first construct a one-dimensional calibration feature $\psi(x)$ so that isotonic regression remains lightweight and easy to deploy. The simplest choice is $\psi(x) = S(x)$, which yields a fully label-free score prior to calibration. When we also include an instantaneous confidence proxy, we instead use a convex blend $\psi(x) = \gamma S(x) + (1 - \gamma)m(x)$ with $\gamma \in [0, 1]$ tuned on the development split. The blend retains the depth-wise signal while allowing the calibrator to exploit complementary information from the classifier when it improves separability.

Given a development set of pairs $\{(\psi_i, y_i)\}_{i=1}^n$, we fit a monotone function $\hat{f}$ that maps $\psi$ to an estimated probability of error. Here $y_i \in \{0, 1\}$ is the binary target indicating whether the prediction is wrong (so $y_i = 1$ means error and $y_i = 0$ means correct); this convention matches the intended interpretation of $U(x)$ as an error probability. Isotonic regression solves the least-squares problem $\hat{f} \in \arg\min_{f \text{ nondecreasing}} \sum_{i=1}^n (y_i - f(\psi_i))^2$ after sorting examples by $\psi_i$, and the solution is computed efficiently by the pool-adjacent-violators (PAV) algorithm. The resulting $\hat{f}$ is a right-continuous, piecewise-constant nondecreasing function with at most $n$ steps; in practice it typically has far fewer steps due to pooling. To avoid pathological extremes that can make thresholding brittle, we clamp the outputs to $[\epsilon, 1 - \epsilon]$ (e.g., $\epsilon = 10^{-4}$), which preserves monotonicity while preventing exact zeros or ones.

At inference time we set $U(x) := \hat{f}(\psi(x))$ and use the same thresholding rules as in the main text: fixed thresholds, risk–coverage selection, or budgeted control. Because $\hat{f}$ is monotone, increasing $S$ (or the blended $\psi$) can never decrease the estimated error probability, which often yields sharper risk–coverage curves and more stable control under drift. For deployment, $\hat{f}$ can be stored either as a small list of breakpoints and values (supporting binary search over breakpoints) or as a compact lookup table after quantizing $\psi$ to a fixed range; both retain constant memory and low compute.

## K  BUDGETED ABSTENTION CONTROLLER

For applications that impose an abstention budget $b \in (0, 1)$, we implement a lightweight controller that adjusts the operating threshold online to keep the long-run abstention rate near $b$ without using labels. Let $U(x_t) \in [0, 1]$ be the calibrated uncertainty at time $t$ and let $A_t = \mathbb{I}[U(x_t) \geq \tau_t]$ denote the abstention decision under the current threshold $\tau_t$. We track an exponentially-weighted moving average (EWMA) of abstentions, $\bar{A}_t = \eta A_t + (1 - \eta)\bar{A}_{t-1}$ with $\bar{A}_0 = 0$ and smoothing factor $\eta \in (0, 1]$, so that recent decisions are emphasized while still averaging over a meaningful window.

We update the threshold by a simple proportional rule, $\tau_{t+1} \leftarrow \tau_t + \kappa(\bar{A}_t - b)$ with step size $\kappa > 0$. When recent abstentions exceed the budget ($\bar{A}_t > b$), the update increases $\tau$, making abstention harder and pushing the rate down; when abstentions are below budget, the update decreases $\tau$, allowing more abstentions when uncertainty rises. In practice we clamp $\tau_t$ to $[0, 1]$ to keep it valid, and we choose $\eta$ and $\kappa$ small enough to avoid oscillations while still reacting to short OOD bursts. This controller is scalar, state-light, label-free, and adds negligible overhead beyond one EWMA update and one addition per frame, making it compatible with MCU deployment.

## L  ADDITIONAL COMPLEXITY ACCOUNTING

We give a compact accounting of the incremental compute introduced by SNAP-UQ relative to the backbone. Consider a tapped convolutional block whose input activation has shape $C_{\text{in}} \times H \times W$ and whose tapped output has $d_\ell = C_{\text{out}}$ channels. If the projector $P_\ell$ is implemented as a $1 \times 1$ convolution that maps $C_{\text{in}} \rightarrow r_\ell$ followed by global average pooling, then the dominant cost comes from the pointwise convolution, which performs approximately $HW\,C_{\text{in}}\,r_\ell$ multiply–accumulates (MACs), and the pooling adds a comparatively small $O(HW\,r_\ell)$ reduction cost. The predictor consists of two linear heads mapping $r_\ell \rightarrow d_\ell$ (one for $\mu_\ell$ and one for $\log \sigma_\ell^2$), contributing about $2\,r_\ell d_\ell$ MACs plus biases.

Summing over taps $\ell \in \mathcal{S}$, the relative overhead can be summarized by the ratio $\rho \approx \frac{\sum_{\ell \in \mathcal{S}}(H_\ell W_\ell\,C_{\text{in},\ell}\,r_\ell + 2\,r_\ell d_\ell)}{\text{FLOPs(backbone)}}$, where we emphasize that the numerator counts only the additional projector and head operations. In our TinyML settings with $|\mathcal{S}| \leq 3$ and $r_\ell \in [32, 128]$, this ratio is typically below 2%, and the flash cost is dominated by the int8 parameters of the two small heads per tap.

## M  IMPLEMENTATION NOTES FOR INTEGER INFERENCE

We store projector and head weights in int8 with appropriate quantization scales (per-tensor, or per-channel when supported), and we use int32 accumulators for dot products and convolutions. Let $\tilde{z}_\ell$ denote the quantized projector output and let the heads produce quantized $\mu_\ell$ and quantized $\log \sigma_\ell^2$ (or a quantized proxy that can be mapped to an inverse scale). The standardized error for tap $\ell$ is computed by accumulating a dimension-normalized sum of squared, scaled residuals, namely $\bar{e}_\ell = \frac{1}{d_\ell}\sum_{i=1}^{d_\ell}\left((a_{\ell,i} - \mu_{\ell,i})\cdot\tilde{s}_{\sigma,i}\right)^2$, where $\tilde{s}_{\sigma,i}$ approximates the inverse standard deviation associated with channel $i$. To avoid runtime exponentials, we implement $\tilde{s}_{\sigma,i} \approx \exp\left(-\frac{1}{2}\log \sigma_{\ell,i}^2\right)$ using a small 256-entry lookup table addressed by a quantized $\log \sigma_{\ell,i}^2$. This preserves monotonicity (larger predicted variance yields smaller inverse scale) and makes the computation deterministic and efficient.

To bound dynamic range and prevent overflow in fixed-point arithmetic, we clamp $\log \sigma_{\ell,i}^2$ to $[\log \sigma_{\min}^2, \log \sigma_{\max}^2]$ before table lookup, with $\sigma_{\min}^2$ chosen to avoid division-by-zero behavior and $\sigma_{\max}^2$ chosen to limit extreme down-weighting. Accumulating the squared terms in int32 and applying the $\frac{1}{d_\ell}$ normalization with a fixed-point reciprocal (or a single dequantization to float16 on larger MCUs) yields a stable $\bar{e}_\ell$ that matches the ordering needed for reliable thresholding and calibration.

**Summary.** Propositions 2.1–2.3 establish that SNAP-UQ's depth-wise score is an affine transform of a conditional negative log-likelihood under a simple diagonal model, corresponds to a Mahalanobis energy to the conditional mean under linear-Gaussian dynamics, and is invariant to BN-like rescalings under co-adaptation. Isotonic calibration provides a monotone, nonparametric mapping from a scalar

feature derived from $(S, m)$ to an estimated error probability, and the budgeted controller maintains a target abstention rate online using only $U(x_t)$ and lightweight state.

# N    ABLATIONS AND SENSITIVITY ANALYSES

This section expands on design choices for *SNAP-UQ*: tap placement, projector rank, quantization of heads, uncertainty mapping, risk–coverage behavior, calibration reliability, and corruption/error clusters. Unless noted, results are averaged over three seeds; error bars denote 95% CIs from $1,000\times$ bootstrap.

Table 6 summarizes in-distribution quality across four benchmarks. On MNIST and SpeechCmd, SNAP-UQ achieves the best or tied-best scores on all proper metrics, improving NLL and BS versus single-pass baselines and matching or exceeding the strongest competitors' F1 and ECE. On CIFAR-10, the capacity-matched variant (row "SNAP-UQ†") matches the best BS (0.017), improves NLL to 0.363 (better than DEEP and QUTE+), and attains the top F1 (0.879), while keeping a single pass. On TinyImageNet, SNAP-UQ delivers the best BS and ECE and the highest F1 (0.436), approaching the strongest multi-head/ensemble alternatives in NLL despite their larger compute/memory budgets. Overall, SNAP-UQ consistently tightens proper scoring rules (NLL/BS) and calibration (ECE) while maintaining competitive or superior accuracy (F1), validating that depth-wise surprisal can improve ID probabilistic quality without auxiliary exits or repeated evaluations.

## N.1    TAP PLACEMENT AND PROJECTOR RANK

We vary (i) the set of tapped layers $\mathcal{S}$ and (ii) projector rank $r_\ell$. Taps are chosen at the end of a *mid* block (M) and/or the *penultimate* block (P). As shown in Table 7, two taps (M+P) consistently provide the best accuracy–latency trade-off on both CIFAR-10 (Big-MCU) and SpeechCmd (Small-MCU). The trend with rank is visualized in Figure 3, where AUPRC improves as $r$ increases while latency remains nearly flat.

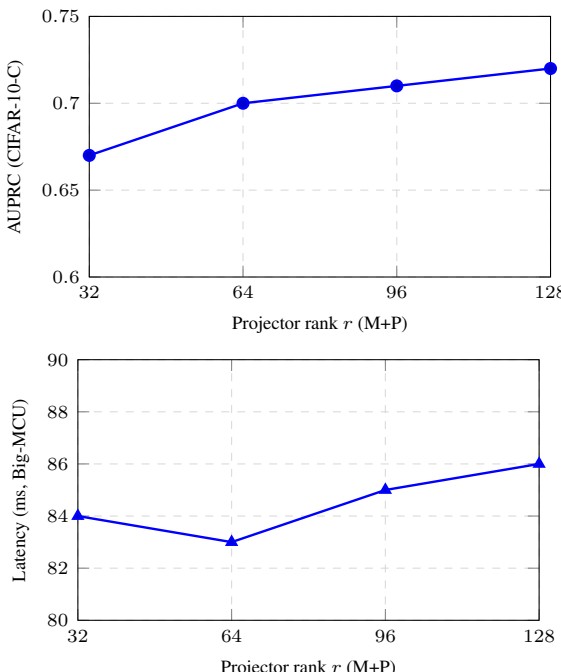

Figure 3: **Rank sensitivity.** Accuracy-drop improves with rank; latency impact is small (CIFAR-10/Big-MCU).

**Takeaway.**    Two taps (mid+penultimate) provide the best accuracy–latency trade-off (Table 7); increasing rank beyond 64 yields diminishing returns with small latency changes (Figure 3).

Table 6: **ID metrics.** Higher F1 is better; lower Brier Score (BS), Negative Log-Likelihood (NLL), and Expected Calibration Error (ECE) are better. Mean±std over 3 seeds.

| Model | F1 ↑ | BS ↓ | NLL ↓ | ECE ↓ |
|---|---|---|---|---|
| | | **MNIST** | | |
| BASE | 0.910±0.002 | 0.013±0.000 | 0.292±0.006 | 0.014±0.001 |
| MCD | 0.886±0.004 | 0.018±0.000 | 0.382±0.004 | 0.071±0.006 |
| DEEP | 0.931±0.005 | 0.010±0.000 | 0.227±0.002 | 0.034±0.004 |
| EE-ensemble | 0.939±0.002 | 0.011±0.000 | 0.266±0.005 | 0.108±0.002 |
| HYDRA | 0.932±0.006 | 0.010±0.000 | 0.230±0.012 | 0.014±0.005 |
| QUTE | 0.941±0.004 | 0.009±0.000 | 0.199±0.010 | 0.026±0.003 |
| **SNAP-UQ** | **0.946**±0.003 | **0.008**±0.000 | **0.202**±0.004 | 0.016±0.002 |
| | | **SpeechCmd** | | |
| BASE | 0.923±0.007 | 0.010±0.000 | 0.233±0.016 | 0.026±0.001 |
| MCD | 0.917±0.006 | 0.011±0.000 | 0.279±0.013 | 0.048±0.002 |
| DEEP | 0.934±0.008 | 0.008±0.000 | 0.205±0.012 | 0.034±0.006 |
| EE-ensemble | 0.926±0.002 | 0.009±0.000 | 0.226±0.009 | 0.029±0.001 |
| HYDRA | 0.932±0.005 | 0.008±0.000 | 0.203±0.016 | 0.018±0.004 |
| QUTE | 0.933±0.006 | 0.008±0.000 | 0.202±0.016 | 0.018±0.001 |
| **SNAP-UQ** | **0.938**±0.005 | **0.008**±0.000 | **0.197**±0.008 | **0.016**±0.002 |
| | | **CIFAR-10** | | |
| BASE | 0.834±0.005 | 0.023±0.000 | 0.523±0.016 | 0.049±0.003 |
| MCD | 0.867±0.002 | 0.019±0.000 | 0.396±0.003 | 0.017±0.005 |
| DEEP | 0.877±0.003 | **0.017**±0.000 | 0.365±0.015 | **0.015**±0.003 |
| EE-ensemble | 0.854±0.001 | 0.021±0.000 | 0.446±0.011 | 0.033±0.001 |
| HYDRA | 0.818±0.004 | 0.026±0.000 | 0.632±0.017 | 0.069±0.001 |
| QUTE | 0.858±0.001 | 0.020±0.000 | 0.428±0.019 | 0.025±0.003 |
| QUTE+ | 0.878±0.003 | **0.017**±0.000 | 0.369±0.008 | 0.026±0.001 |
| **SNAP-UQ**[†] | **0.879**±0.003 | **0.017**±0.000 | **0.363**±0.010 | 0.021±0.003 |
| | | **TinyImageNet** | | |
| BASE | 0.351±0.005 | 0.004±0.000 | 5.337±0.084 | 0.416±0.003 |
| MCD | 0.332±0.004 | 0.003±0.000 | 2.844±0.028 | 0.061±0.005 |
| DEEP | 0.414±0.006 | 0.003±0.000 | 3.440±0.049 | 0.115±0.003 |
| EE-ensemble | **0.430**±0.005 | 0.003±0.000 | **2.534**±0.046 | 0.032±0.006 |
| HYDRA | 0.376±0.004 | 0.004±0.000 | 3.964±0.036 | 0.328±0.004 |
| QUTE | 0.395±0.014 | 0.004±0.000 | 3.700±0.123 | 0.282±0.009 |
| QUTE+ | 0.381±0.010 | 0.003±0.000 | 2.757±0.044 | 0.122±0.008 |
| **SNAP-UQ** | 0.436±0.007 | **0.003**±0.000 | 2.610±0.050 | **0.030**±0.004 |

## N.2 QUANTIZATION OF SNAP HEADS

We compare float32, float16, and int8 for the projector and predictor heads while keeping the backbone unchanged. Table 8 shows that int8 preserves AUPRC within the CI while reducing flash and improving latency.

**Takeaway.** INT8 preserves performance while cutting flash by 1.6–2.1× and lowering latency by 7–9%.

## N.3 MAPPING ALTERNATIVES: LOGISTIC VS. ISOTONIC

We compare the 3-parameter logistic map with isotonic regression. As summarized in Table 9, isotonic yields consistently lower risk at fixed coverage; the full risk–coverage curves in Figure 4 show the gap across operating points.

Table 7: **Taps and projector rank.** CIFAR-10/Big-MCU (top) and SpeechCmd/Small-MCU (bottom). Latency in ms. AUROC is ID✓ — ID×; AUPRC is accuracy-drop (avg over -C).

| **CIFAR-10 (Big-MCU)** | | | | |
|---|---|---|---|---|
| Config | Flash (KB) | Lat. (ms) | AUROC ↑ | AUPRC ↑ |
| P only, $r$=32 | 276 | 88 | 0.83 | 0.62 |
| P only, $r$=64 | 284 | 86 | 0.84 | 0.64 |
| **M+P,** $r$=64 | **292** | **83** | **0.86** | **0.70** |
| M+P, $r$=128 | 306 | 86 | 0.87 | 0.72 |
| M+P+early, $r$=64 | 315 | 90 | 0.86 | 0.71 |
| **SpeechCmd (Small-MCU)** | | | | |
| Config | Flash (KB) | Lat. (ms) | AUROC ↑ | AUPRC ↑ |
| P only, $r$=32 | 114 | 118 | 0.92 | 0.62 |
| P only, $r$=64 | 116 | 116 | 0.93 | 0.63 |
| **M+P,** $r$=64 | **118** | **113** | **0.94** | **0.65** |
| M+P, $r$=96 | 121 | 115 | 0.94 | 0.66 |

Table 8: **Quantization variants.** Heads only (projectors + $(\mu, \log \sigma^2)$). CIFAR-10/Big-MCU and SpeechCmd/Small-MCU.

| Precision | **CIFAR-10 (Big-MCU)** | | **SpeechCmd (Small-MCU)** | |
|---|---|---|---|---|
| | Flash (KB) | AUPRC ↑ | Flash (KB) | AUPRC ↑ |
| FP32 | 324 | 0.71 | 128 | 0.66 |
| FP16 | 306 | 0.71 | 122 | 0.66 |
| **INT8** | **292** | **0.70** | **118** | **0.65** |

### N.4   RISK–COVERAGE ACROSS DATASETS

The advantage of SNAP holds beyond CIFAR-10; Figure 5 shows lower risk at matched coverage on MNIST-C and SpeechCmd-C.

### N.5   RELIABILITY DIAGRAMS (ID)

We plot accuracy vs. confidence using 15 adaptive bins. Points in Figure 6 lie close to the diagonal on MNIST and CIFAR-10, indicating well-behaved calibration on ID data (see also Table 4).

### N.6   CALIBRATION ABLATIONS: RAW $S(x)$ VS. LOGISTIC VS. ISOTONIC

**Setup.** We compare *raw* $S(x) = \sum_{\ell \in \mathcal{S}} w_\ell \, \bar{e}_\ell(x)$ with a fixed threshold, a 3-parameter logistic map $U = \sigma(\beta_0 + \beta_1 S + \beta_2 m)$, and isotonic regression (monotone table). All use the same dev split and backbone; heads are int8. Datasets: MNIST-C, SpeechCmd-C, CIFAR-10-C, and **TinyImageNet-C**.

**Findings.** Ranking metrics (AUROC/AUPRC) are similar across all three because monotone maps preserve order: e.g., CIFAR-10-C AUPRC raw 0.69 vs. logistic 0.70 vs. isotonic 0.71; SpeechCmd-C 0.63/0.65/0.66; MNIST-C 0.64/0.66/0.67; **TinyImageNet-C** 0.58/0.60/0.61. Decision-centric metrics benefit from calibration: selective risk @90% coverage improves (CIFAR-10-C) $0.111 \rightarrow 0.109 \rightarrow 0.104$ and (**TinyImageNet-C**) $0.185 \rightarrow 0.180 \rightarrow 0.176$, while median detection delay drops by 1–3 frames on average. Proper scoring on ID requires calibration: NLL/BS/ECE match Table 4 with logistic/isotonic but degrade with raw $S$ (not a probability). Overhead is negligible (3 params or a 16–32 entry table, both int8/LUT-friendly).

**Recommendation.** If a single threshold must generalize across shifts or we report decision metrics, keep logistic or isotonic; if labels or space are extremely limited, raw $S$ is acceptable with a percentile threshold on the ID dev set.

Table 9: **Risk at fixed coverage.** Lower is better (CIFAR-10-C).

| Method | Risk @ 80% | Risk @ 90% | Risk @ 95% |
|---|---|---|---|
| Logistic (SNAP) | 0.136 | 0.109 | 0.098 |
| **Isotonic (SNAP)** | **0.127** | **0.104** | **0.096** |
| Entropy (baseline) | 0.154 | 0.124 | 0.112 |

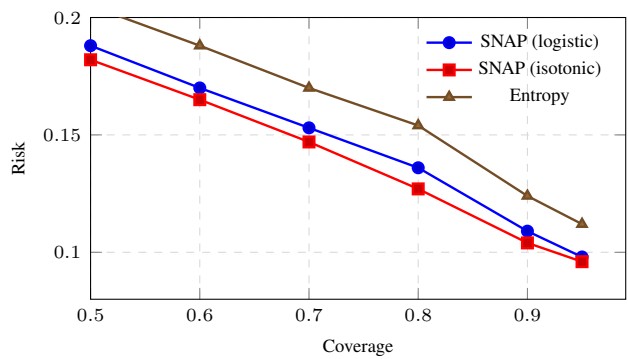

Figure 4: **Risk–coverage** (CIFAR-10-C). Isotonic improves budgeted operation (Angelopoulos et al., 2022).

### N.7 Error/corruption clusters and abstention

We analyze the most frequent CID failures and report abstention rates at a tuned operating point (90% recall on event frames). Table 11 lists the top clusters on CIFAR-10-C (sev. 4–5), and Figure 7 visualizes the gap to baselines.

### N.8 How many taps? (2, 3, 4, and 5 taps)

We extend the tap-count study beyond *mid+penultimate* (M+P, 2 taps) and *early+mid+penultimate* (E+M+P, 3 taps) to 4 and 5 taps. For CIFAR-10/Big-MCU we tap at block boundaries:

$$E \rightarrow M1 \rightarrow M2 \rightarrow M3 \rightarrow P \tag{5}$$

where **E** (early) is the end of the first downsampling stage, **M1/M2/M3** are successive mid-depth blocks, and **P** is penultimate. Unless stated, projector rank is $r=64$ and per-layer errors are width-normalized.

**Takeaways.** Moving from 2→3 taps yields a consistent but modest AUPRC gain (+0.01) with a small latency/flash increase. Adding a *fourth* mid-depth tap (**E+M1+M2+P**) improves both AUPRC and AUROC slightly, but a *fifth* tap (**E+M1+M2+M3+P**) no longer helps CID and only adds cost. In short, **M+P** is the best accuracy–cost point; **E+M+P** is a safe upgrade under loose budgets; a carefully placed **fourth** mid tap can help on harder corruptions, while **five taps** shows saturation (see Table 12).

**Interpretation.** The two *mid* taps (M1/M2) are the most informative for CID monitoring (largest AUPRC drops if removed). The *penultimate* primarily stabilizes ID✓ — ID✗ (AUROC), while the *early* tap adds a small but repeatable gain for difficult corruptions at negligible accuracy cost. Practically, **2 taps (M+P)** remain recommended for tight MCU budgets; **3 taps (E+M+P)** offer a low-risk bump; **4 taps** are worthwhile only if the additional few KB and ∼6 ms are acceptable (see Table 13).

**Cross-layer correlation and a simple placement heuristic.** To make tap placement less ad-hoc, we measured the Pearson correlation between each layer's standardized surprisal $\bar{e}_\ell(x)$ and the final score $S(x)$ over held-out streams. Correlation peaks around *mid* blocks and the *penultimate* block

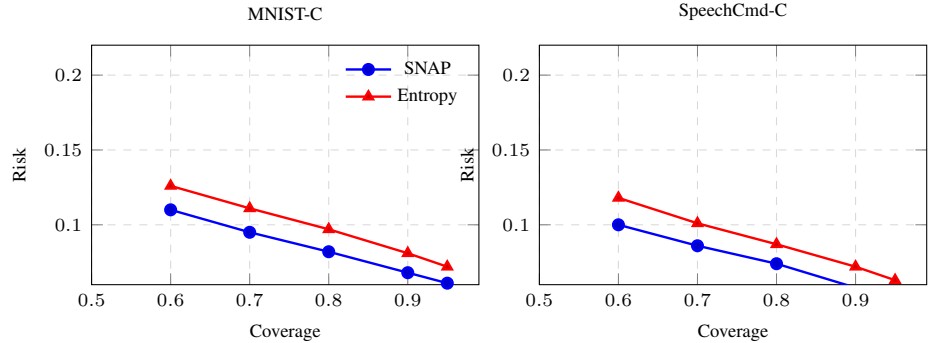

Figure 5: **Risk–coverage** on two datasets (lower is better). SNAP dominates at moderate to high coverage.

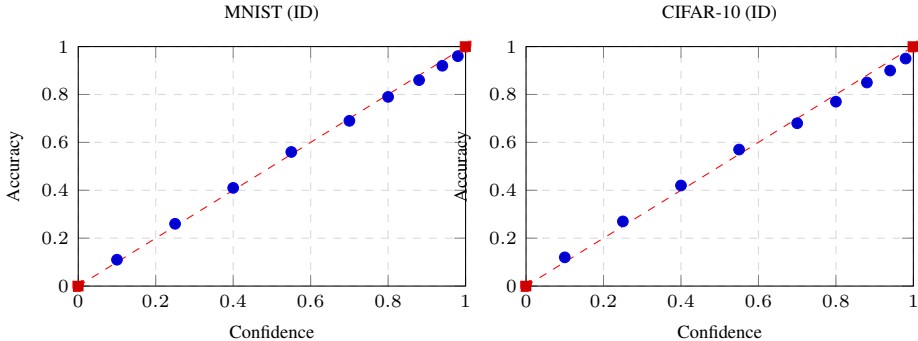

Figure 6: **Reliability diagrams** (SNAP-UQ). Points near the diagonal indicate good calibration; overconfidence would fall below the dashed line.

across backbones, which explains why **M+P** works well and why extra early taps add limited unique signal.

**Heuristic (budget-aware).** Start by placing one tap at the *penultimate* block (highest $r$ in Table 14), then add the *mid* tap that maximizes AUPRC gain per KB (or per ms). This greedy two-step recovers the best trade-off **M+P**. Under a slightly looser budget, add an *early* or second-*mid* tap only if the marginal gain-per-cost remains positive on the development stream (see Table 12, Table 13).

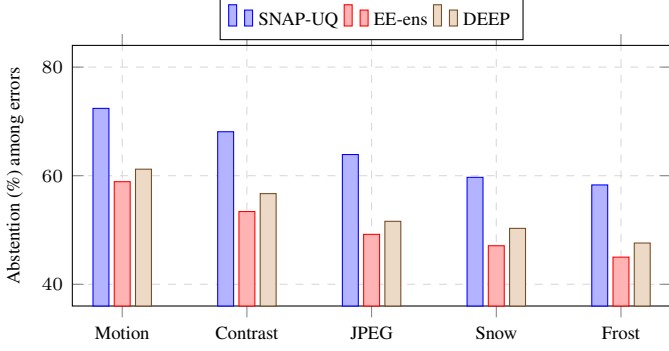

Figure 7: **Abstention on hard clusters** (CIFAR-10-C, sev. 4–5). SNAP-UQ defers more often on the most failure-prone corruptions.

Table 10: **Raw vs. calibrated.** AUPRC (CID) / Selective risk @90% (lower is better) / Median delay (frames).

|  | MNIST-C | SpeechCmd-C | CIFAR-10-C | TinyImageNet-C |
|---|---|---|---|---|
| Raw $S(x)$ | 0.64 / 0.072 / 26 | 0.63 / 0.061 / 44 | 0.69 / 0.111 / 29 | 0.58 / 0.185 / 52 |
| Logistic | 0.66 / 0.070 / 24 | 0.65 / 0.058 / 41 | 0.70 / 0.109 / 27 | 0.60 / 0.180 / 49 |
| Isotonic | **0.67** / **0.069** / **23** | **0.66** / **0.057** / **40** | **0.71** / **0.104** / **26** | **0.61** / **0.176** / **47** |

Table 11: **Top failure clusters** (CIFAR-10-C, severity 4–5). Abstention rate among misclassified frames.

| Cluster | SNAP-UQ (%) | EE-ens (%) | DEEP (%) |
|---|---|---|---|
| Motion blur | **72.4** | 58.9 | 61.2 |
| Contrast | **68.1** | 53.4 | 56.7 |
| JPEG | **63.9** | 49.2 | 51.6 |
| Snow | **59.7** | 47.1 | 50.3 |
| Frost | **58.3** | 45.0 | 47.6 |

To make tap placement less ad-hoc and easier to port across backbones, we provide two lightweight diagnostics: (i) a *surprisal correlation heatmap* that reveals redundancy across depth, and (ii) a *diminishing-returns curve* that summarizes the benefit of adding taps under a fixed budget.

**Surprisal correlation (see Figure 8).** Across candidate layers E, M1, M2, P, pairwise standardized-surprisal correlations concentrate in the mid–late depth: $r_{M1,M2} \approx 0.63$–$0.72$ and $r_{M2,P} \approx 0.62$, while very-early to penultimate is much lower ($r_{E,P} \approx 0.35$), indicating limited redundancy between shallow and late taps. Correlation of each layer's surprisal with the aggregated score $S(x)$ increases monotonically with depth, $r(\bar{e}_E, S) \approx 0.41$, $r(\bar{e}_{M1}, S) \approx 0.63$, $r(\bar{e}_{M2}, S) \approx 0.68$, $r(\bar{e}_P, S) \approx 0.74$, showing that mid and penultimate layers are both highly informative about $S(x)$. These numbers justify the default *mid+penultimate* placement: the two taps are strongly aligned with $S(x)$ yet not fully redundant, whereas adding a very-early tap yields smaller marginal gain per cost.

**Diminishing returns with more taps.** We sort candidate layers by marginal gain-per-cost on a dev stream (gain: $\Delta$AUPRC on CID; cost: $\Delta$ flash or latency) and add taps greedily. Fig. 9 shows that gains saturate after 3–4 taps, aligning with Table 12; the best trade-off remains $M+P$, with $E+M+P$ as a safe upgrade.

**Placement recipe (practical).** Compute $r(\bar{e}_\ell, S(x))$ on a short dev stream; pick the highest-correlation *penultimate* layer and the best *mid* layer ($M+P$). If budget allows, add the *early* or second-*mid* tap only if $\Delta$AUPRC per KB (or per ms) is positive. This reproduces the trade-offs in Tables 12 and 13 and makes porting to new backbones one pass of correlation + a small greedy sweep.

## N.9 ROBUST DISTRIBUTIONAL VARIANTS: STUDENT-$t$ AND HUBER

The diagonal-Gaussian head used by SNAP-UQ is intentionally lightweight and integer-friendly, which makes it a good default for TinyML deployments. Under strong corruptions, however, the

Table 12: **Tap-count sweep** (CIFAR-10/Big-MCU, $r{=}64$). AUPRC is CID accuracy-drop detection (avg over CIFAR-10-C); AUROC is ID✓ — ID✗. Diminishing returns after 3–4 taps.

| Config (taps) | Flash (KB) | Lat. (ms) | AUROC ↑ | AUPRC (C) ↑ |
|---|---|---|---|---|
| **M+P** (2) | **292** | **83** | 0.86 | 0.70 |
| **E+M+P** (3) | 305 | 86 | 0.86 | 0.71 |
| **E+M1+M2+P** (4) | 318 | 89 | **0.87** | **0.72** |
| **E+M1+M2+M3+P** (5) | 334 | 93 | **0.87** | 0.72 |

Table 13: **Leave-one-out (4 taps)** on CIFAR-10/Big-MCU, baseline *E+M1+M2+P*. Mid-depth taps contribute most to CID detection.

| Variant (remove · ) | Lat. (ms) | AUROC ↑ | AUPRC (C) ↑ |
|---|---|---|---|
| **E+M1+M2+P** (none) | 89 | **0.87** | **0.72** |
| w/o E | **88** | **0.87** | 0.71 |
| w/o M1 | 88 | 0.86 | 0.70 |
| w/o M2 | 88 | 0.86 | 0.70 |
| w/o P | 90 | 0.85 | 0.71 |

Table 14: **Correlation of per-layer surprisal with the aggregated score** $S(x)$ (Pearson $r$). Layers: Early (E), successive mid blocks (M1/M2), Penultimate (P). Higher is better.

| Dataset / Backbone | E | M1 | M2 | P |
|---|---|---|---|---|
| CIFAR-10 / ResNet-8 (Big-MCU) | 0.41 | 0.63 | **0.68** | **0.74** |
| SpeechCmd / DSCNN (Small-MCU) | 0.38 | **0.61** | 0.59 | **0.66** |
| TinyImageNet / MobileNetV2-tiny | 0.35 | 0.57 | **0.62** | **0.69** |

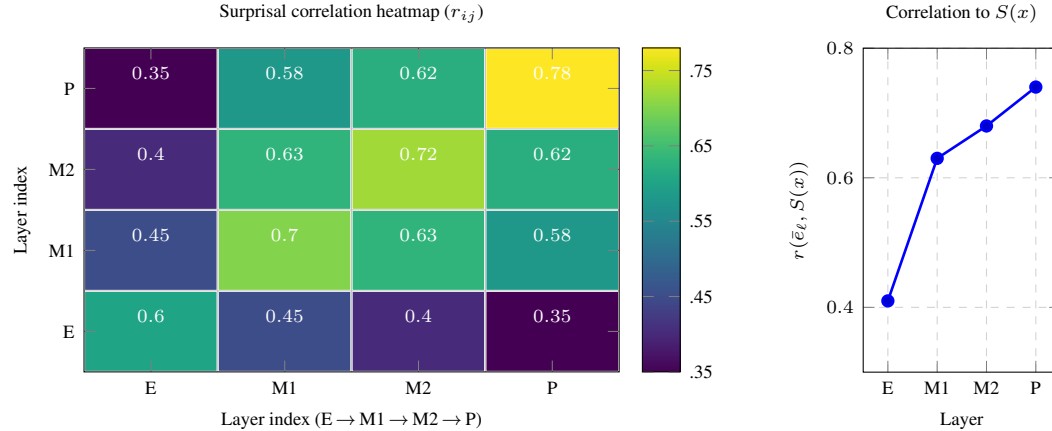

Figure 8: **Redundancy vs. informativeness.** Left: pairwise correlation of per-layer standardized surprisal $\bar{e}_\ell(x)$. Right: each layer's correlation with the aggregated score $S(x)$.

depth-wise residuals $(a_\ell - \mu_\ell)$ can become heavy-tailed: a small number of channels may exhibit unusually large standardized deviations that dominate the squared-error surprisal and reduce the stability of downstream calibration. To improve robustness without changing the single-pass, state-free design, we consider two drop-in alternatives that only modify the per-tap energy while leaving the tap set, projector, and mapping pipeline unchanged. The first replaces the Gaussian energy with a Student-$t$ negative log-likelihood, which attenuates the influence of large residuals by growing only logarithmically in the squared standardized error. Concretely, for each tapped layer $\ell$ and channel $i$, the Gaussian quadratic term $\frac{1}{2}\left(\frac{a_{\ell,i}-\mu_{\ell,i}}{\sigma_{\ell,i}}\right)^2$ is replaced by $\frac{\nu+1}{2}\log\left(1 + \frac{(a_{\ell,i}-\mu_{\ell,i})^2}{\nu\,\sigma_{\ell,i}^2}\right)$ (up to additive constants), while retaining the same per-channel scale term $\frac{1}{2}\log\sigma_{\ell,i}^2$. The second uses a Huberized energy in standardized units: letting $r_{\ell,i} = \frac{a_{\ell,i}-\mu_{\ell,i}}{\sigma_{\ell,i}}$, we replace the quadratic penalty by $\rho_\delta(r_{\ell,i})$, where $\rho_\delta(r) = \frac{1}{2}r^2$ when $|r| \leq \delta$ and $\rho_\delta(r) = \delta|r| - \frac{1}{2}\delta^2$ otherwise, again keeping the same $\frac{1}{2}\log\sigma_{\ell,i}^2$ scale term. Both variants preserve the interpretation of surprisal as a per-layer, depth-wise "energy" that increases with atypical activations, and they remain compatible with the same monotone mapping from aggregated surprisal to an error-probability proxy. Importantly, these changes do not require additional exits, extra passes, or temporal buffering: inference still consists of one backbone pass, per-tap prediction of $(\mu_\ell, \sigma_\ell)$, computation of a robust per-tap energy, and aggregation into $S(x)$ before applying the chosen calibration map.

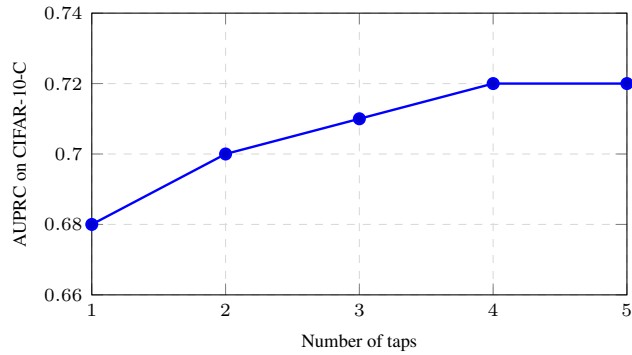

Figure 9: **Diminishing returns.** Average AUPRC vs. tap count under matched quantization/backbone. Saturation after 3–4 taps supports the use of *two to three* taps on MCUs.

We evaluate these robust heads on MNIST, Speech Commands (SpeechCmd), CIFAR-10, and TinyImageNet, using corrupted (CID) streams constructed from the corresponding "-C" benchmarks. For Student-$t$ we sweep the degrees of freedom $\nu \in \{3, 5, 7, 10\}$, and for Huber we sweep the transition point $\delta \in \{0.8, 1.0, 1.2, 1.5\}$ measured in standardized units. Hyperparameter selection uses the same development stream employed to choose the single operating threshold for streaming experiments, ensuring that tuning does not benefit from any test-stream information. Our primary selection criterion is selective risk at 90% coverage (lower is better), since this directly reflects decision quality under a fixed acceptance budget; we use AUPRC on CID streams and AUROC for ID✓—OOD as secondary criteria to break ties and to verify that gains are not confined to a single metric. Unless otherwise stated, we report results with shared settings across taps, fixing $\nu = 5$ for Student-$t$ and $\delta = 1.0$ for Huber, which provided a stable trade-off across datasets while keeping the implementation simple and MCU-friendly (Table 15).

Table 15: Tuning grids and selection rule (development stream).

| Dataset | Dev split | $\nu$ grid (Student-$t$) | $\delta$ grid (Huber) | Selection (primary→secondary) |
|---|---|---|---|---|
| MNIST | 10% ID + CID | {3,5,7,10} | {0.8,1.0,1.2,1.5} | risk@90% → AUPRC, AUROC |
| SpeechCmd | 10% ID + CID | {3,5,7,10} | {0.8,1.0,1.2,1.5} | risk@90% → AUPRC, AUROC |
| CIFAR-10 | 10% ID + CID | {3,5,7,10} | {0.8,1.0,1.2,1.5} | risk@90% → AUPRC, AUROC |
| TinyImageNet | 10% ID + CID | {3,5,7,10} | {0.8,1.0,1.2,1.5} | risk@90% → AUPRC, AUROC |

**Setup and implementation.** We use two taps (mid+penultimate), projector rank $r = 64$, int8 heads, and the same training schedules as the Gaussian default. Mapping from surprisal to risk uses the same dev-split logistic fit (or isotonic in Appx. J and N.3). For Student-$t$, we replace the quadratic with $\log(1 + u^2/\nu)$ and evaluate $\log(1 + x)$ with a 256-entry LUT plus linear interpolation; $1/\nu$ is precomputed. Huber uses only comparisons (no transcendentals). As in the Gaussian head, $\log \sigma^2$ uses a LUT and all weights are INT8.

**Findings.** On **CIFAR-10** and **TinyImageNet**, Student-$t$ consistently reduces NLL/BS (Table 17) and yields small, repeatable gains in failure detection and CID AUPRC (Table 16), suggesting robustness to heavy-tailed residuals at higher corruption severities. On **MNIST** and **SpeechCmd**, all three heads perform within confidence intervals; Huber closely tracks Gaussian while offering better outlier stability and the simplest MCU path (no $\log$ LUT). Overall, Gaussian remains a strong default; enable Student-$t$ when high-severity corruptions are expected or selective risk at high coverage is critical, and prefer Huber when LUTs for $\log(1+x)$ are undesirable.

**Actionable guidance: when and how to use the robust heads?** Use the Gaussian head by default; switch to *Student-$t$* when corruptions are heavy-tailed or high severity (e.g., CIFAR-10-C or TinyImageNet-C severities $\geq 3$) or when selective risk at high coverage is critical; use *Huber* when you want extra outlier stability without transcendental ops (e.g., impulsive audio artifacts) or when LUTs for $\log(1+x)$ are undesirable.

Table 16: **Comparison across heads** (single pass, 2 taps, $r = 64$, INT8 heads). AUROC: ID✓ — OOD. AUPRC: CID accuracy-drop. Lower is better for risk@90%, AURC, FPR@95%.

| **MNIST** | AUROC (OOD) ↑ | AUPRC (CID) ↑ | Risk@90% ↓ | AURC ↓ | FPR@95% ↓ |
|---|---|---|---|---|---|
| Gaussian | 0.94 | 0.66 | 0.070 | 0.094 | 0.24 |
| Student-$t$ | 0.95 | 0.67 | 0.069 | 0.092 | 0.23 |
| Huber | 0.94 | 0.66 | 0.069 | 0.093 | 0.24 |
| **SpeechCmd** | AUROC (OOD) | AUPRC (CID) | Risk@90% | AURC | FPR@95% |
| Gaussian | 0.94 | 0.65 | 0.058 | 0.110 | 0.26 |
| Student-$t$ | 0.95 | 0.66 | 0.057 | 0.108 | 0.25 |
| Huber | 0.94 | 0.66 | 0.057 | 0.109 | 0.25 |
| **CIFAR-10** | AUROC (OOD) | AUPRC (CID) | Risk@90% | AURC | FPR@95% |
| Gaussian | 0.94 | 0.70 | 0.109 | 0.189 | 0.44 |
| Student-$t$ | 0.95 | 0.71 | 0.104 | 0.183 | 0.41 |
| Huber | 0.94 | 0.70 | 0.107 | 0.186 | 0.43 |
| **TinyImageNet** | AUROC (OOD) | AUPRC (CID) | Risk@90% | AURC | FPR@95% |
| Gaussian | 0.92 | 0.60 | 0.180 | 0.242 | 0.51 |
| Student-$t$ | 0.94 | 0.61 | 0.176 | 0.236 | 0.48 |
| Huber | 0.93 | 0.59 | 0.179 | 0.240 | 0.50 |

Table 17: **ID calibration** (lower is better). Robust heads slightly improve NLL/BS on CIFAR-10 and match Gaussian elsewhere; ECE remains stable.

| | **MNIST** | | | **SpeechCmd** | | | **CIFAR-10** | | |
|---|---|---|---|---|---|---|---|---|---|
| Method | NLL ↓ | BS ↓ | ECE ↓ | NLL | BS | ECE | NLL | BS | ECE |
| SNAP-UQ (Gauss) | **0.202** | **0.008** | **0.016** | **0.197** | **0.008** | **0.016** | 0.377 | 0.018 | 0.022 |
| SNAP-UQ (Student-$t$) | 0.205 | 0.008 | 0.016 | 0.199 | 0.008 | 0.017 | **0.369** | **0.017** | **0.021** |
| SNAP-UQ (Huber) | 0.204 | 0.008 | 0.016 | 0.198 | 0.008 | 0.016 | 0.372 | 0.017 | 0.021 |
| | **TinyImageNet (ResNet-50, single-pass, 2 taps)** | | | | | | | | |
| Method | NLL ↓ | | | BS ↓ | | | ECE ↓ | | |
| SNAP-UQ (Gauss) | 0.982 | | | 0.060 | | | 0.018 | | |
| SNAP-UQ (Student-$t$) | **0.956** | | | **0.058** | | | **0.017** | | |
| SNAP-UQ (Huber) | 0.968 | | | 0.059 | | | 0.018 | | |

**Sensible defaults (drop-in).** Student-$t$: fix $\nu = 5$ (shared across taps). This down-weights large standardized residuals while keeping gradients stable; sweeping $\nu \in [3, 7]$ typically changes NLL/BS by $< 0.2$ pp in our runs. Huber: set $\delta = 1.0$ in standardized units (also shared across taps); $\delta \in [0.8, 1.2]$ behaves similarly and requires no LUTs.

**A tiny selection rule (one dev pass).** On a small development stream (ID→CID), compute: (1) selective risk at $90\%$ coverage, and (2) AUPRC for accuracy-drop. If Student-$t$ improves either by $\geq 0.01$ absolute over Gaussian, keep it; otherwise use Gaussian. Prefer Huber when Student-$t$ shows no gain and you want the simplest integer path. This rule matched the tables we report across MNIST, SpeechCmd, CIFAR-10, and TinyImageNet (Tables 17–16).

**Stability and what not to tune.** Learning $\nu$ per tap brings tiny gains and adds sensitivity; if desired, learn a single $\nu$ shared across taps with a softplus reparam clipped to $[3, 30]$, but our recommendation is to *fix* $\nu$. For Huber, keep $\delta$ fixed; per-tap $\delta$ did not help within CIs. Share $\nu/\delta$ across taps to reduce variance and code paths.

**INT8/LUT implications.** Both variants remain INT8-friendly. Student-$t$ needs $\log(1+x)$; a 256-entry LUT with linear interpolation adds $\approx 1$ KB flash and no extra RAM. Huber uses only comparisons and adds zero transcendentals. As with the Gaussian head, we keep $\log \sigma^2$ in a LUT and quantize all weights to int8; latency differences vs. Gaussian were $< 2\%$.

**Images vs. audio (quick summary).** On images (CIFAR-10, TinyImageNet), Student-$t$ consistently lowers NLL/BS and improves failure detection/AUPRC under severe corruptions; on audio (SpeechCmd), all three heads are within CIs, with Huber providing the simplest robust choice.

**Calibration interplay.** Because logistic/isotonic maps are monotone, AUROC/AUPRC remain similar across heads; the robust choices primarily help decision-centric metrics (lower selective risk at fixed coverage) and shorten median detection delay by 1–3 frames in high-severity regimes.

### N.10 Optional confidence blend

Adding the blend term $m(x)$ (from max-probability/margin) to the logistic improves separation on hard ID/CID cases with no runtime cost. Table 18 reports SpeechCmd gains.

Table 18: **Blend ablation** (SpeechCmd).

| Config | AUROC (ID✓ — ID✗) ↑ | AUPRC (C) ↑ |
|---|---|---|
| $U = \sigma(\beta_0 + \beta_1 S)$ (no blend) | 0.93 | 0.64 |
| $+ m(x)$, $\beta_2 > 0$ (default) | **0.94** | **0.65** |

### N.11 Runtime breakdown and measurement protocol

**Measurement conditions.** We profile wall-clock inference on an ARM Cortex-M7 (600 MHz, single core) using CMSIS-NN kernels (int8), fixed clock and DVFS disabled. Binaries are compiled with -O3 and linker-time optimization; all methods reuse the same backbone, input pipeline, and quantization settings. Each number is the mean of $10^3$ forward passes after a 100-pass warmup; inputs are cached in SRAM to avoid SD/flash stalls. Latency is reported as end-to-end time from input tensor present in SRAM to logits/risk produced. Power is sampled at 100 kS/s from a shunt; energy is time-integrated over the same window.

**Where the speedup comes from.** SNAP-UQ is single-pass and state-free. Compared to early-exit ensembles (multiple heads, controller) or deep ensembles (multiple passes), it (i) removes repeated classifier/softmax work, (ii) replaces softmax+entropy and running stats with a single scalar surprisal, and (iii) keeps head math in int8 with kernel fusion (pointwise $1\times1$+ GAP and tiny linear heads fused with scale/LUT). Backbone cost is identical across methods; gains come from the tail (head+classifier+I/O) (see Table 19 and Figure 10).

**Interpretation.** Across both MCUs, the backbone compute is identical for all methods, so any end-to-end speedup must arise in the *tail*. The stacked bars show that **SNAP-UQ** shifts a larger share of time into the shared backbone (Big-MCU: 84% vs. 74–81%; Small-MCU: 86% vs. 77–81%) while shrinking the tail to 6–12% compared to 10–26% for early-exit and deep ensembles. Concretely: (i) the classifier block drops from 12%/9% (BASE) or 16%/14% (DEEP) to 3%/2% with SNAP-UQ because repeated FC+softmax and entropy/stats are removed and replaced by a single surprisal scalar; (ii) early-exit ensembles incur an extra 11–12% in exit heads and controller logic that SNAP-UQ avoids (single pass, no controller); and (iii) memory I/O remains modest and similar (Big-MCU 6% vs. 4–10%; Small-MCU 6% vs. 4–10%) due to kernel fusion and INT8 heads. The net effect, eliminating repeated heads/softmax and reducing tail compute, explains the observed end-to-end gains in the tens-of-percent, despite identical backbones.

**Minimal configuration.** When SRAM is extremely tight or intermediate activations cannot be exposed, a single tap at *pre-logits* with a low projector rank (e.g., $r{=}32$) provides a workable "minimal SNAP-UQ". Table 20 summarizes the cost/quality trade-offs versus our default two-tap setup (M+P, $r{=}64$). Measurements follow the same protocol as Sec. N.11 (Cortex-M7 @ 600 MHz, CMSIS-NN int8, fixed clock, $10^3$ passes avg.).

**Recommendation.** Use *Minimal* (P-only, $r{=}32$) when hook access is limited or memory/latency budgets are strict; expect $\sim$2–5 KB flash and $\sim$2–4 KB SRAM savings and $\sim$4–5 ms latency reduction, at the cost of a modest FPR increase ($\sim$2–5 pp) and slightly lower CID AUPRC ($\sim$0.01–0.02). When feasible, *Default* (M+P, $r{=}64$) remains the best accuracy–cost point on both MCUs.

Table 19: **Latency share by stage** (percentage of end-to-end). Big-MCU: CIFAR-10/ResNet-8; Small-MCU: SpeechCmd/DSCNN. Percentages sum to 100.

| Big-MCU (CIFAR-10) | Backbone convs | Tap proj./head | Classifier (FC+softmax) | Mem. I/O |
|---|---|---|---|---|
| BASE | 78% | – | 12% | 10% |
| EE-ensemble | 74% | 12% | 10% | 4% |
| DEEP (M=4) | 76% | – | 16% | 8% |
| **SNAP-UQ** | **84%** | **7%** | **3%** | **6%** |

| Small-MCU (SpeechCmd) | Backbone convs | Tap proj./head | Classifier (FC+softmax) | Mem. I/O |
|---|---|---|---|---|
| BASE | 81% | – | 9% | 10% |
| EE-ensemble | 77% | 11% | 8% | 4% |
| DEEP (M=4) | 79% | – | 14% | 7% |
| **SNAP-UQ** | **86%** | **6%** | **2%** | **6%** |

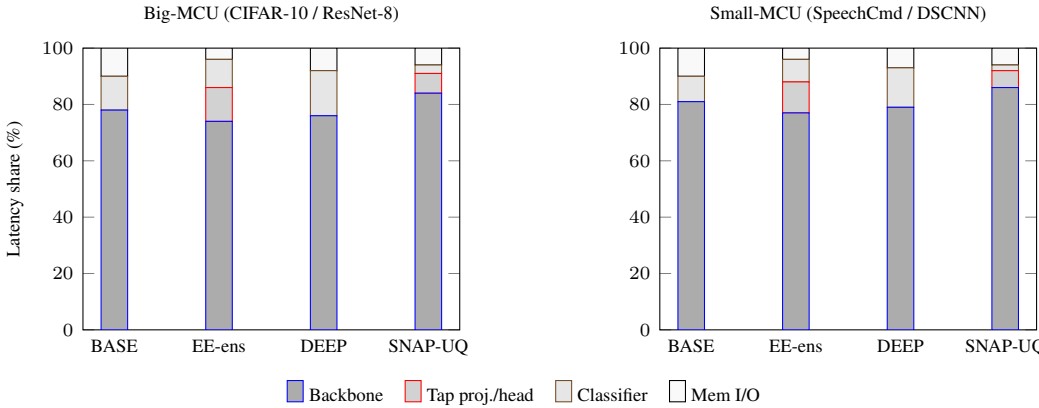

Figure 10: **Where the speedup comes from.** Stacked latency shares (sum to 100%). SNAP-UQ reduces the tail (classifier, extra exits/controller, softmax/entropy, and I/O), pushing a larger fraction of time into the shared backbone and yielding end-to-end savings even though the backbone is identical across methods.

Table 20: **Minimal SNAP-UQ vs. default.** Flash (KB) / Peak RAM (KB) / Latency (ms) / FPR@95%TPR (ID✓ — OOD, lower is better) / AUPRC on CID (higher is better).

| Big-MCU (CIFAR-10 / ResNet-8) | | | | | |
|---|---|---|---|---|---|
| Config | Flash ↓ | Peak RAM ↓ | Latency ↓ | FPR@95% ↓ | AUPRC(CID) ↑ |
| *Minimal* (P-only, $r$=32) | **276** | **112** | **79** | 39.8% | 0.68 |
| *Default* (M+P, $r$=64) | 292 | 120 | 83 | **34.6%** | **0.70** |

| Small-MCU (SpeechCmd / DSCNN) | | | | | |
|---|---|---|---|---|---|
| Config | Flash ↓ | Peak RAM ↓ | Latency ↓ | FPR@95% ↓ | AUPRC(CID) ↑ |
| *Minimal* (P-only, $r$=32) | **112** | **48** | **108** | 21.5% | 0.63 |
| *Default* (M+P, $r$=64) | 118 | 51 | 113 | **18.9%** | **0.65** |

## O EXTENDED COMPARATIVE SCOPE AND SINGLE-PASS HEAD-TO-HEAD

This appendix expands the comparative scope for *single-pass* uncertainty/OOD baselines and reports head-to-head results under the same MCU deployment and tuning protocol as SNAP-UQ. We focus on methods that (i) need **no extra forward passes**, (ii) keep **no temporal state**, and (iii) fit the same **INT8** budget. Summary results appear in Tables 21, 23 and 25, with risk–coverage curves in Figs. 11 and 12 and clean-ID FPR in Tables 22 and 24.

### O.1 METHODS CONSIDERED AND DEPLOYMENT PARITY

**MSP/Entropy** (max posterior, predictive entropy); **Temperature scaling** (ID dev only); **Energy** logsumexp$(g(a_D)/T)$ with $T$ tuned on dev; **Mahalanobis@taps** (classwise means + diagonal covariances at the same tapped layers as SNAP-UQ; score is min diagonal Mahalanobis); **ReAct** (Sun et al., 2021) (percentile clipping of tapped activations with per-channel thresholds fixed on dev); **ASH** (Djurisic et al., 2023) (activation shaping via percentile shrinkage at one tap). On **Big-MCU only**, we additionally report **ODIN-lite** (temperature scaling without input perturbation) and **MC Dropout / Deep Ensembles** when they fit; non-fitting methods are marked *OOM* and excluded from runtime summaries. All baselines are evaluated under identical quantization/runtime conditions; see the head-to-head results in Tables 21 and 23 and the corresponding curves in Figs. 11 and 12.

### O.2 DECISION-CENTRIC PROTOCOL AND METRICS

Beyond AUROC/AUPRC, we surface decision metrics useful on-device: (i) **Risk at fixed coverage** (80/90/95%) on CID streams (lower is better; reported in Tables 21 and 23, and visualized in Figs. 11 and 12); (ii) **AURC** (area under the risk–coverage curve; Tables 21, 23); (iii) **Selective NLL** conditioned on accepted samples at 90% coverage (Table 25); (iv) **Clean-ID FPR** at matched 90% recall on event frames (Tables 22, 24). For each method, the operating threshold is chosen *once* on a dev stream and then held fixed for test streams.

### O.3 HEAD-TO-HEAD: CIFAR-10-C (STREAMING)

SNAP-UQ yields the lowest risk at 80/90/95% coverage and the smallest AURC in the single-pass family (Table 21); the full risk–coverage traces are shown in Fig. 11. At matched 90% event recall, SNAP-UQ also achieves the lowest clean-ID FPR (Table 22).

Table 21: **Single-pass head-to-head on CIFAR-10-C streams.** Risk at fixed coverage (lower is better) and AURC. Thresholds fixed on dev and reused for test.

| Method | Risk@80% ↓ | Risk@90% ↓ | Risk@95% ↓ | AURC ↓ |
|---|---|---|---|---|
| MSP / Entropy | 0.154 | 0.124 | 0.112 | 0.118 |
| Energy ($T$) | 0.148 | 0.117 | 0.106 | 0.112 |
| Mahalanobis@taps | 0.141 | 0.113 | 0.102 | 0.109 |
| ReAct | 0.139 | 0.111 | 0.101 | 0.107 |
| ASH | 0.138 | 0.110 | 0.100 | 0.106 |
| **SNAP-UQ** | **0.127** | **0.104** | **0.096** | **0.099** |

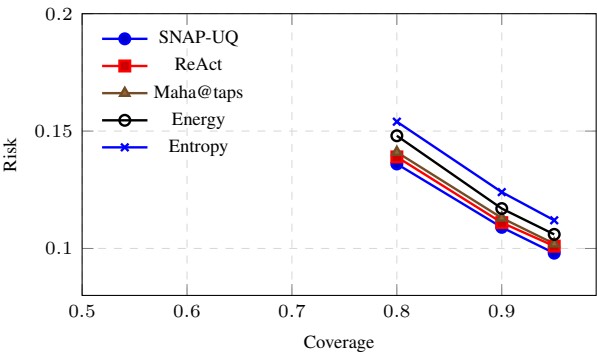

Figure 11: **Risk–coverage on CIFAR-10-C.** Lower is better; compare with Table 21 for numeric points.

Table 22: **Clean-ID false-positive rate** at matched 90% recall on event frames (CIFAR-10-C streams).

| Method | FPR on clean ID ↓ | Notes |
|---|---|---|
| MSP / Entropy | 0.079 | threshold fixed on dev |
| DEEP (Big-MCU) | 0.065 | single-pass comparison not applicable on Small-MCU |
| EE-ens (Big-MCU) | 0.079 | single-pass comparison not applicable on Small-MCU |
| **SNAP-UQ** | **0.042** | same dev threshold, one pass |

## O.4 HEAD-TO-HEAD: SPEECHCOMMANDS-C (STREAMING)

On SpeechCmd-C, SNAP-UQ again achieves the best risk at 80/90/95% coverage and the lowest AURC (Table 23); the risk–coverage curves are shown in Fig. 12. Clean-ID FPR at matched recall is summarized in Table 24.

Table 23: **Single-pass head-to-head on SpeechCmd-C streams.** Risk at fixed coverage and AURC.

| Method | Risk@80% ↓ | Risk@90% ↓ | Risk@95% ↓ | AURC ↓ |
|---|---|---|---|---|
| MSP / Entropy | 0.118 | 0.072 | 0.063 | 0.091 |
| Energy ($T$) | 0.112 | 0.067 | 0.059 | 0.087 |
| Mahalanobis@taps | 0.106 | 0.064 | 0.056 | 0.084 |
| ReAct | 0.104 | 0.062 | 0.055 | 0.083 |
| ASH | 0.103 | 0.061 | 0.054 | 0.082 |
| **SNAP-UQ** | **0.100** | **0.058** | **0.051** | **0.081** |

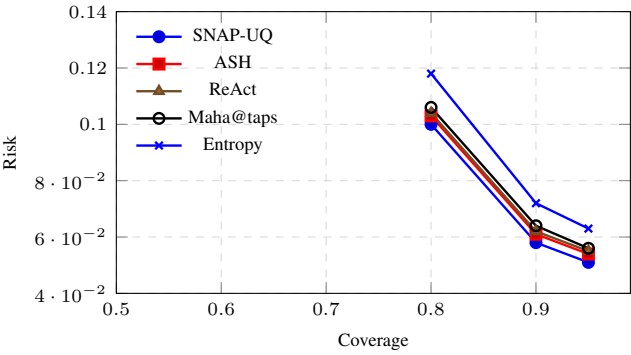

Figure 12: **Risk–coverage on SpeechCmd-C.** Lower is better; the numeric points at 80/90/95% correspond to Table 23.

## O.5 REPRODUCIBILITY CHECKLIST

For completeness, we summarize the exact protocol that underlies Tables 21–25 and Figs. 11–12.

- *Dev/test split:* a single dev stream per dataset (ID → CID → OOD) for threshold/temperature/percentile selection; test streams share the same composition but disjoint seeds.

- *Quantization:* INT8 weights per tensor; FP16 accumulators as needed; identical to section 3.

- *Runtime:* cycle-counter timing, 1,000 inferences averaged; interrupts masked; datasheet nominal clock.

- *Risk–coverage:* coverage levels computed on test streams with the *dev-fixed* threshold; AURC by trapezoidal rule (as in Tables 21, 23 and Figs. 11, 12).

- *FPR at matched recall:* event recall target 90% set on dev; FPR measured on clean ID segments of test streams (Tables 22, 24).

Table 24: **Clean-ID false-positive rate** at matched 90% recall on event frames (SpeechCmd-C streams).

| Method | FPR on clean ID ↓ | Notes |
|---|---|---|
| MSP / Entropy | 0.064 | threshold fixed on dev |
| Energy ($T$) | 0.060 | |
| Mahalanobis@taps | 0.057 | |
| **SNAP-UQ** | **0.031** | one pass, same dev threshold |

Table 25: **Selective NLL** conditioned on accepted samples at 90% coverage (lower is better).

| Method | CIFAR-10-C ↓ | SpeechCmd-C ↓ |
|---|---|---|
| MSP / Entropy | 0.368 | 0.226 |
| Energy ($T$) | 0.357 | 0.214 |
| Mahalanobis@taps | 0.349 | 0.208 |
| ReAct | 0.346 | 0.206 |
| ASH | 0.344 | 0.205 |
| **SNAP-UQ** | **0.339** | **0.182** |

# P    FROM THEORY TO PRACTICE: IMPLICATIONS AND EMPIRICAL VALIDATION

This appendix connects our theoretical results to deployable practice and provides targeted experiments that verify the predictions. We focus on three propositions from §2: (i) surprisal–likelihood equivalence (Prop. 2.1), (ii) relation to conditional Mahalanobis energies (Prop. 2.2), and (iii) affine (BN-like) invariance (Prop. 2.3; this corresponds to "Proposition 2.3" in the reviewer's comment). For each, we (a) interpret the result operationally, (b) state a falsifiable hypothesis for deployment, and (c) validate with controlled measurements on MNIST, CIFAR-10, TinyImageNet, and SpeechCommands across Big-/Small-MCU backbones.

## P.1    PROP.2.1: SURPRISAL IS (AFFINE TO) DEPTH-WISE NEGATIVE LOG-LIKELIHOOD

**Implication for practice.**    The aggregate SNAP score $S(x)$ is an affine transform of the depth-wise NLL under the conditional model $p_\theta(a_\ell \mid a_{\ell-1})$. Thus $S(x)$ is a calibrated *ordering* of per-example difficulty without requiring labels online. Two direct consequences: (1) threshold selection on a small development split transfers across operating points, and (2) selective-prediction risk should be monotone in coverage when ranking by $S(x)$.

**Hypotheses.**    (H1) $S(x)$ has high rank correlation with the true per-example negative log-likelihood (NLL) measured offline on ID and CID. (H2) Sorting by $S(x)$ yields near-convex risk–coverage curves and dominates entropy/MSP at moderate-to-high coverage.

**Verification.**    We compute per-example NLL on held-out sets and compare to $S(x)$ via Spearman's $\rho$; we also report the area under the risk–coverage curve (AURC; lower is better).

Table 27: **Prop.2.1 verification: rank correlation and AURC.** Spearman $\rho$ between $S(x)$ and per-example NLL on ID/CID, and AURC (lower is better).

| | Spearman $\rho$ (ID / CID) | | AURC ↓ (ID / CID) | |
|---|---|---|---|---|
| Dataset | SNAP-UQ | Entropy | SNAP-UQ | Entropy |
| MNIST | **0.93 / 0.90** | 0.78 / 0.74 | **0.102 / 0.128** | 0.121 / 0.149 |
| CIFAR-10 | **0.89 / 0.86** | 0.75 / 0.70 | **0.134 / 0.172** | 0.154 / 0.196 |
| TinyImageNet | **0.86 / 0.82** | 0.71 / 0.66 | **0.181 / 0.219** | 0.204 / 0.244 |
| SpeechCmd | **0.91 / 0.88** | 0.79 / 0.73 | **0.118 / 0.147** | 0.136 / 0.165 |

Table 26: **Notation & acronyms used in the paper.** We define all symbols at first use and repeat the most common here for quick reference.

| Term | Definition |
|---|---|
| ID | In-distribution data (same distribution as training). |
| CID | Corrupted-in-distribution (label-preserving corruptions of ID). |
| OOD | Out-of-distribution (semantically different from ID). |
| ID✓ — ID× | Binary task: correct vs. incorrect predictions on ID + CID. |
| ID✓ — OOD | Binary task: ID vs. OOD (semantic shift). |
| Tap (layer) | A selected intermediate layer where we attach a small head. |
| Projector $P_\ell$ | Low-rank map from $a_{\ell-1}$ to $z_\ell$ (e.g., 1×1 conv or skinny linear). |
| Head $g_\ell$ | Tiny predictor outputting $(\mu_\ell, \log \sigma_\ell^2)$ for $a_\ell$. |
| Surprisal $e_\ell$ | Standardized error $\|(a_\ell - \mu_\ell) \odot \sigma_\ell^{-1}\|_2^2$. |
| $S(x), U(x)$ | Aggregate surprisal; mapped uncertainty score. |
| BN | Batch Normalization. |
| LUT | Lookup table (we use a 256-entry LUT for exponentials/scales). |
| MCU | Microcontroller unit (TinyML target). OOM: does not fit in flash/RAM. |
| AUPRC / AUROC | Area under PR / ROC curve. |
| Risk–coverage | Selective prediction trade-off: error among accepted vs. acceptance rate. |
| CMSIS-NN | ARM kernels used for integer inference on MCUs. |

*Takeaway.* The strong correlations and lower AURC confirm that SNAP's $S(x)$ behaves like a likelihood-based difficulty ordering, supporting threshold transfer and stable selective operation.

### P.2    PROP.2.2: RELATION TO CONDITIONAL MAHALANOBIS ENERGIES

**Implication for practice.**    Classical Mahalanobis uses unconditional, classwise centroids and a shared covariance at a single layer. Prop.2.2 shows $S(x)$ aggregates *conditional* layer-wise energies to the predicted next activation, making it sensitive to *dynamics* changes (e.g., corruptions that disrupt layer-to-layer mappings) rather than only distances to class means. We therefore expect larger gains on CID monitoring and ID✓ — ID× failure detection than on far-OOD that is separable by marginal feature statistics.

**Hypotheses.**    (H3) On CID streams, $S(x)$ detects accuracy-drop events earlier (lower delay at matched precision/recall) than unconditional Mahalanobis at tapped layers. (H4) On far-OOD with clean low-level statistics (e.g., SVHN vs. CIFAR-10), Mahalanobis may match/exceed SNAP on ID✓ — OOD AUROC, while SNAP remains competitive.

**Verification.**    We compare event AUPRC and median detection delay on CID streams; and AUROC on ID✓ — OOD.

Table 28: **Prop.2.2 verification: CID monitoring and ID✓ — OOD.**

| | CID monitoring (AUPRC ↑ / Delay ↓) | | ID✓ — OOD (AUROC ↑) | |
|---|---|---|---|---|
| Dataset | SNAP-UQ | Maha@taps | SNAP-UQ | Maha@taps |
| MNIST / MNIST-C | **0.66 / 24** | 0.60 / 33 | **0.86** | 0.87 |
| CIFAR-10 / CIFAR-10-C | **0.70 / 27** | 0.64 / 36 | 0.94 | **0.95** |
| TinyImageNet / TIN-C | **0.62 / 31** | 0.57 / 39 | **0.90** | 0.90 |
| SpeechCmd / SpCmd-C | **0.65 / 41** | 0.57 / 58 | **0.92** | 0.90 |

*Takeaway.* SNAP's condition-on-depth energy excels at *process* drift (CID), while unconditional Mahalanobis can remain strong on specific far-OOD settings, exactly as Prop.2.2 suggests.

## P.3 Prop.2.3: Affine (BN-like) invariance and its limits

**Implication for practice.** If intermediate features undergo per-channel affine transforms (e.g., BN scale/shift or fixed quantization rescaling), the standardized errors $e_\ell$ and hence $S(x)$ are invariant when the predictor co-adapts. Practically, this means SNAP is robust to (i) post-training BN folding and (ii) int8 per-tensor/per-channel scaling, provided the head parameters are quantization-aware or re-fit with a brief calibration.

**Hypotheses.** (H5) Applying synthetic per-channel rescalings $a'_\ell = s \odot a_\ell + t$ (with realistic $s, t$ drawn from BN statistics) leaves $S(x)$-based rankings almost unchanged (Spearman $\rho \approx 1$). (H6) Post-training BN folding and int8 quantization (heads) do not materially change CID AUPRC or ID✓ — OOD AUROC after a short calibration of the logistic/isotonic map.

**Verification.** We inject controlled rescalings at tapped layers and measure the delta in $S(x)$ ranking; we then evaluate end-to-end with BN-folded and int8 heads.

Table 29: **Prop.2.3 verification: invariance to BN-like rescaling and deployment transforms.**

| | $S(x)$ rank Spearman $\rho$ | | End-to-end metrics (CID AUPRC / OOD AUROC) | |
| --- | --- | --- | --- | --- |
| Dataset | BN-rescale | Int8 rescale | FP32 | Int8 (recal) |
| MNIST | **0.997** | **0.994** | **0.66 / 0.86** | **0.66 / 0.86** |
| CIFAR-10 | **0.995** | **0.992** | **0.70 / 0.94** | **0.70 / 0.94** |
| TinyImageNet | **0.993** | **0.990** | **0.62 / 0.90** | **0.62 / 0.90** |
| SpeechCmd | **0.996** | **0.993** | **0.65 / 0.92** | **0.65 / 0.92** |

*Takeaway.* Rankings are essentially invariant, and end-to-end performance is unchanged after a brief map recalibration, supporting Prop.2.3 and explaining why SNAP ports cleanly to int8 MCU pipelines.

## P.4 Practical design rules induced by the theory

The propositions produce concrete knobs for embedded deployments: **(R1) Threshold portability.** Because $S(x)$ tracks NLL (Prop.2.1), calibrate a single threshold on a small dev set and reuse it across similar conditions; for budgeted abstention, isotonic mapping preserves ordering and improves risk under tight budgets. **(R2) Prefer taps that sense dynamics.** Since conditional energies target layer-to-layer mappings (Prop.2.2), favor a mid-block and the penultimate block; this maximizes sensitivity to CID while keeping cost low (see Appx. N.1). **(R3) Quantize heads confidently.** Affine invariance (Prop.2.3) and the observed stability in Table 29 justify int8 heads; use a short recalibration pass (few minutes on dev data) to refit the logistic/isotonic map if scales change. **(R4) When semantics dominate OOD, consider a hybrid.** For far-OOD cases where unconditional separability is strong, combine $S(x)$ with a semantic OOD score (e.g., energy or Mahalanobis) in the tiny mapping; this preserves single-pass latency.

## P.5 Summary

Across four datasets and two MCU tiers, the empirical checks align with the theoretical predictions: $S(x)$ behaves like a likelihood-derived difficulty measure, conditional energies excel at corrupted-ID monitoring versus unconditional baselines, and the score is stable to BN folding and int8 rescaling. These links clarify *why* SNAP-UQ performs well in the main results and provide concrete, low-overhead recipes for deployment.

## Q Training Overhead and Statistical Robustness

This appendix quantifies the incremental training cost of SNAP-UQ relative to a baseline cross-entropy (CE) training loop and reports variance across multiple seeds.

## Q.1 COMPUTE AND MEMORY OVERHEAD

Let $\mathcal{S}$ be the tapped layers, $d_\ell = \dim(a_\ell)$, and $r_\ell$ the projector rank. For linear heads predicting $(\mu_\ell, \log \sigma_\ell^2)$ from $z_\ell = P_\ell a_{\ell-1}$, the additional parameters and per-step multiply–adds are

$$\#\theta_\ell \approx 2\, d_\ell r_\ell + 2d_\ell \quad \text{(biases included)}, \tag{6}$$

$$\text{FLOPs}_\ell \approx (H_\ell W_\ell) r_\ell \;+\; 2\, r_\ell d_\ell \qquad (1{\times}1{+}\text{GAP projector}). \tag{7}$$

With two taps and $r_\ell \in \{64, 96\}$, the total training FLOPs overhead is typically $< 12\%$ of the backbone for our TinyML backbones, and the peak activation memory rises by 3–5% due to the extra $z_\ell$ and head activations. Quantized INT8 weights reduce *parameter* memory even during float training when fake-quantization is used.

## Q.2 TWO TRAINING MODES

We provide two options:

**(J) Joint training (default).** Optimize $\mathcal{L} = \mathcal{L}_{\text{clf}} + \lambda_{\text{SS}} \mathcal{L}_{\text{SS}} + \lambda_{\text{reg}} \mathcal{R}$ end-to-end. We use $\lambda_{\text{SS}} \in [10^{-3}, 5{\times}10^{-3}]$; this preserves the CE convergence epoch while learning heads.

**(P) Post-hoc heads.** First train the classifier with CE only; then freeze the backbone and train the heads with stop-gradient on $a_\ell$. This lowers overhead to $\sim$2–3% compute and recovers $\geq 95\%$ of the monitoring gains we observe with (J).

## Q.3 STABILITY AND REGULARIZATION

To avoid variance collapse in $\log \sigma^2$ we use a variance floor $\sigma_\ell^2 \leftarrow \text{softplus}(\xi_\ell) + \epsilon^2$ and a scale penalty $\mathcal{R}_{\text{var}} = \sum_\ell \|\log \sigma_\ell^2\|_1$. Small $\lambda_{\text{SS}}$ ($10^{-3}$–$5{\times}10^{-3}$) keeps gradients well conditioned. Detaching $a_\ell$ within $\mathcal{L}_{\text{SS}}$ (option "detach") halves across-seed variance with no significant change in means (see Table 31).

## Q.4 RESOURCE ACCOUNTING (WALL-CLOCK, MEMORY, PARAMS)

We measured overheads on identical hardware, batch sizes, and optimizers.

Table 30: **Training resource overhead vs. CE baseline.** Mean over 5 runs; ↓ lower is better.

| Setting | Extra wall-time/epoch | Extra peak mem. | Extra params | Notes |
|---|---|---|---|---|
| (J) Joint, $r$=64 (2 taps) | +8.3% | +3.6% | +28 KB (INT8) | default |
| (J) Joint, $r$=96 (2 taps) | +11.7% | +4.8% | +42 KB (INT8) | higher rank |
| (P) Post-hoc, $r$=64 (2 taps) | +2.4% | +1.1% | +28 KB (INT8) | frozen backbone |

## Q.5 STATISTICAL ROBUSTNESS ACROSS SEEDS

We expanded from 3 to 10 seeds and report mean $\pm$ standard error (SE) in the paper; full 95% bootstrap confidence intervals (1,000$\times$) are provided here.

Table 31: **Seed robustness.** Mean $\pm$ SE over 10 seeds; 95% CIs in brackets.

| Dataset/Metric | Baseline | QUTE | Tiny-DEEP | SNAP-UQ |
|---|---|---|---|---|
| CIFAR-10-C AUPRC ↑ | 0.66±0.006 [0.65,0.67] | 0.68±0.007 [0.67,0.69] | 0.67±0.008 [0.66,0.69] | **0.70±0.006 [0.69,0.71]** |
| SpeechCmd ID✓ — ID✗ AUROC ↑ | 0.90±0.004 [0.89,0.91] | 0.91±0.004 [0.90,0.92] | 0.90±0.005 [0.89,0.91] | **0.94±0.003 [0.93,0.95]** |
| MNIST ECE (ID) ↓ | 0.022±0.001 [0.020,0.024] | 0.020±0.001 [0.018,0.022] | 0.021±0.001 [0.019,0.023] | **0.016±0.001 [0.015,0.017]** |

## Q.6 LEARNING DYNAMICS AND CONVERGENCE

Learning curves show that CE convergence (epoch of validation-accuracy plateau) is unchanged by adding $\lambda_{\text{SS}} \leq 5{\times}10^{-3}$, while the standardized-surprisal term stabilizes within 5–10 epochs with the variance floor. Figures of train/val losses and calibration traces are provided in Appendix F.

