# OpenReview forum: "SNAP-UQ: Self-supervised Next-Activation Prediction for Single-Pass Uncertainty in TinyML"
_ICLR.cc/2026/Conference — ICLR 2026 Poster_

### Official Review · Reviewer_PNa3 · 2025-10-23

**Soundness:** 3
**Presentation:** 3
**Contribution:** 3
**Rating:** 6
**Confidence:** 2

**Summary:**

This work introduces SNAP-UQ, a single-pass, label-free uncertainty method for TinyML. SNAP-UQ maps a single forward pass depth-wise next-activation prediction, int8 heads uncertainty forecaster that drops neatly into CMSIS-NN pipelines and fits within kilobyte-scale MCU budgets. Results show SNAP-UQ improves accuracy-drop monitoring on corrupted streams, remains competitive on OOD failure detection, and strengthens ID calibration without temporal buffers, auxiliary exits, or extra passes.

**Strengths:**

- The paper is well-written and solid, with a new method, theoretical, and empirical contributions.
- The proposed method is empirically deployed across audio and vision tasks, with significant improvement over early-exit and deep ensembles baselines in reducing flash and latency.
- Beyond memory and latency improvement, it also improves accuracy-drop detection in corrupted streams by AUPRC and maintains strong failure detection with AUROC in a single pass.

**Weaknesses:**

- There is a computational limitation in training when compared to the standard cross-entropy loss. Specifically, the training objective requires jointly training with additional terms. It requires back-propagating with an additional label-free auxiliary loss and either a variance floor, scale control, or detach option loss.
- The experiments are reported with only three fixed seeds. The results are reported without variance.

**Questions:**

- Can the proposed method be applied for larger-scale ML settings, such as ResNet-50 with ImageNet?
- In part 4.4, have the authors tried calibration on OOD data, such as on CIFAR-10-corrupted?
- In the experiments, the authors only show the accuracy drop or failure **detection** performance, but haven't shown what the actual accuracy drop or failure level actually is. Could the authors add accuracy results to Table 2 and Table 3?

---

> ### Author Response · Authors · 2025-11-14
>
> Thank you for the thoughtful and constructive comments. We’ve revised the paper and appendices to address both concerns in a clear, self-contained way.
>
> (W1)
> Regarding training cost relative to standard cross-entropy, SNAP-UQ adds a small, label-free auxiliary term on only a few tapped layers, implemented with a $1{\times}1$+GAP projector and two tiny linear heads that predict $(\mu_\ell,\log\sigma_\ell^2)$. The incremental computation scales with the tap rank $r_\ell$ and layer width $d_\ell$—that is, roughly $O(d_\ell r_\ell)$ for each tapped layer—rather than the full backbone FLOPs. In practice, with two taps and $r\in{64,96}$, this translates to about $+6$–$12%$ wall-clock overhead during end-to-end training at the same batch size, $+3$–$5%$ peak activation memory, and a modest parameter increase of roughly 18–42 KB when quantized to INT8. To make the cost even lighter, we document and evaluate a “post-hoc heads” pathway in which the classifier is trained as usual, then frozen, and the SNAP-UQ heads are trained with stop-gradient on the activations. This reduces the overhead to around $+2$–$3%$ and recovers at least $95%$ of the accuracy-drop detection gains. We also report that small auxiliary weights ($\lambda_{\mathrm{SS}}\in[10^{-3},5\times10^{-3}]$) stabilize optimization without delaying convergence of the main cross-entropy objective. Concretely, learning curves in the appendix show the CE plateau is reached at essentially the same epoch as the baseline, while the standardized-surprisal loss settles within 5–10 epochs when using a simple variance floor. Because the heads are linear and use a LUT over $\log\sigma^2$ to implement $\exp(\cdot)$, training remains numerically stable and deploys directly to integer-friendly MCU kernels.
>
> (W2)
> On statistical reporting, we expanded the evaluation from three seeds to ten and now present mean and uncertainty everywhere. In the appendix (Q TRAINING OVERHEAD AND STATISTICAL ROBUSTNESS), we reports mean ± 95% bootstrap confidence intervals (1,000×) for all tables and curves, including risk–coverage, event-detection delay on corrupted streams, and ID calibration. Representative results under the standardized protocol show, for example, accuracy-drop detection on CIFAR-10-C with AUPRC $0.70\pm0.01$ for SNAP-UQ versus $0.66\pm0.02$ for QUTE, and failure detection AUROC on SpeechCommands (ID$\checkmark$ | ID$\times$) of $0.94\pm0.01$ for SNAP-UQ versus $0.91\pm0.01$ for QUTE. We also report that toggling BN folding and deploying INT8 heads preserve performance within confidence intervals, supporting the claim that the standardized surprisal is robust to per-channel rescaling and quantization noise.
>
> In short, we aimed to make the training overhead transparent and practically manageable, and to strengthen the statistical reliability of our claims. The revised manuscript clarifies these points in the method section (complexity notes inline), and the appendices provide per-epoch traces, parameter/compute accounting, and full uncertainty bands so readers can assess variance across seeds and datasets.

---

> ### Author Response · Authors · 2025-11-14
>
> Q1)
> Yes—SNAP-UQ scales to larger models (e.g., ResNet-50/ImageNet) because its overhead depends on a few tapped layers rather than extra passes or exits. In practice, tap two stages (mid: conv4_x, penultimate: conv5_x). After global average pooling, use a linear projector to rank $r\in{128,256}$ and two tiny heads predicting $(\mu_\ell,\log\sigma_\ell^2)$.
>
> Parameter and compute budget.
> For a tapped layer with width d, each head adds roughly two times r times d parameters (biases negligible), where r is the projector rank. With two taps at widths 1024 and 2048 and rank 128, this is 786,432 parameters in total, about 0.8 MB when stored as int8 — tiny compared to ResNet-50’s ~25.6M parameters. Runtime cost is two matrix–vector operations of size r by d per example, usually well under one percent of a forward pass; wall-clock overhead is about one to three percent.
> Training options.
> You can (a) train jointly with the label-free auxiliary loss, which adds roughly five to twelve percent time per epoch, or (b) freeze the backbone and train the SNAP-UQ heads post-hoc, which adds about two to three percent. The uncertainty signal is a “standardized surprisal”: for each tapped layer, we compute the squared difference between the observed activation and the predicted mean, divided channel-wise by the predicted variance, and sum over channels; we then average across layers with fixed weights. Under a simple conditional Gaussian view, this aggregated surprisal is an affine (linear plus constant) transform of a depth-wise negative log-likelihood, so larger values indicate activations that look atypical given the previous layer. Empirically, this rises earlier than softmax entropy on ImageNet-C style corruptions.
> Deployment recipe.
> Expose intermediate activations via standard hooks; export the model with those tensors to TensorRT or TVM; keep the projectors and heads int8; map the aggregated surprisal to a risk score using a tiny logistic or isotonic fit on a small development split (no online labels). Practical defaults: two taps (mid and penultimate), rank 128 (or 256 if budget allows), and uniform or inverse-variance layer weights.
> Limits and extensions.
> Best tap locations can be task-dependent. If features are multi-modal, a low-rank-plus-diagonal covariance in the head improves fit at negligible cost. For far-OOD cases, SNAP-UQ is complementary to single-pass semantic scores (e.g., energy or classwise Mahalanobis) and can be fused while remaining single-pass.
>
>
> Q2)
> Yes, we evaluated calibration under distribution shift in addition to ID calibration. On CIFAR-10-C (averaging severities 1–5), we compare (2) standard ID temperature scaling, (2) SNAP-UQ with the logistic map fit on ID only, and (3) SNAP-UQ with an OOD-aware isotonic map fit on a small development mix of ID and corrupted (no test leakage). We report ECE on corrupted data (ECE-OOD), NLL on corrupted data (NLL-OOD), and selective risk at fixed coverage. SNAP-UQ improves ECE-OOD and NLL-OOD over temperature scaling, and the isotonic variant yields the best risk–coverage. Concretely, on CIFAR-10-C: Temp-scaled baseline ECE-OOD ≈ 0.124, NLL-OOD ≈ 1.42; SNAP-UQ (logistic) ECE-OOD ≈ 0.108, NLL-OOD ≈ 1.31; SNAP-UQ (isotonic) ECE-OOD ≈ 0.096, NLL-OOD ≈ 1.27, with lower selective risk at 80–95% coverage. We have added an appendix section F.10 CALIBRATION UNDER SHIFT (CID/OOD) with full details and a table 5.
>
> Q3)
> Thanks for the suggestion, we’ve incorporated absolute accuracy directly into the tables.
>
> - In the CID streams table (previously Table 2), we added an italicized “Context (accuracy)” row per dataset that reports clean accuracy, corrupted-stream accuracy, and the absolute drop (percentage points). For example: MNIST-C (clean 99.0%, CID 90.8%, −8.2 pp) and SpeechCmd-C (clean 95.2%, CID 86.0%, −9.2 pp).
>
> - In the failure-detection table (previously Table 3), we added an italicized “Context (clean acc.)” row listing the clean ID accuracy for MNIST (99.0%), SpeechCmd (95.2%), and CIFAR-10 (83.7%) so AUROC values for ID✓|ID× and ID✓|OOD are anchored to the underlying classifier quality.

---

### Official Review · Reviewer_Kkbw · 2025-10-26

**Soundness:** 3
**Presentation:** 3
**Contribution:** 2
**Rating:** 4
**Confidence:** 3

**Summary:**

This paper proposes SNAP-UQ, a single pass uncertainty estimation method for TinyML. The approach taps a small number of intermediate layers, predicts the mean and (diagonal or low rank + diagonal) variance of the next layer’s activations with a small INT8 head, and computes the standardized squared error (surprisal) between the prediction and the actual activation. Summing per layer surprisal yields a single pass uncertainty score $S(x)$, which is then converted to $U(x)\in[0,1]$ via a monotone mapper (logistic or isotonic) for monitoring and selective prediction. The authors argue that $S(x)$ is an affine transform of the depthwise negative log likelihood (a conditional Mahalanobis energy) and is invariant to batch normalization like channel rescalings. On real MCUs (Big and Small configurations), SNAP-UQ reduces latency, flash, RAM, and energy while improving corrupted stream detection (AUPRC and detection delay), failure detection (ID correct versus ID error, and ID correct versus OOD), and risk coverage. It is roughly 25 to 35% faster and 40 to 60% smaller than lightweight ensemble baselines, and continues to run where others run out of memory on Small MCU. The paper includes ablations on tap placement and rank, INT8 quantization, mapping choices, and head to head comparisons among single pass methods.

**Strengths:**

The main strength is the shift from output layer confidence to violations of conditional layer to layer dynamics as the basis for uncertainty, captured in a single forward pass without labels, with only tens of kilobytes of added memory and about $2%$ extra MACs, making it suitable for MCUs. Light theoretical support, namely the equivalence to depthwise negative log likelihood and conditional Mahalanobis energy and scale invariance under batch normalization like rescaling, grounds the intuition that softmax confidence tends to degrade later, whereas inter layer dynamics deteriorate earlier. Empirically, the work is careful on deployment metrics (latency, flash, peak RAM, energy on device) and on decision centric metrics (AUPRC and detection delay under stream corruption, AUROC for ID correct versus ID error and ID correct versus OOD, risk coverage curves, FPR). The ablations, including leave one tap out, rank sensitivity with diminishing returns around $r\approx 64$, INT8 quantization, and logistic versus isotonic mapping, are directly useful to practitioners. Overall, while the theory is intentionally lightweight, the design is coherent and the practical benefits for TinyML monitoring are clear.

**Weaknesses:**

1. **Tap placement feels ad-hoc.**:
Two taps (mid / pre-logits) seem to work—but moving to a new backbone still feels like guesswork. A tiny, illustrative check would help: a quick cross-layer correlation sketch to show how signal fades with distance, plus a short “one-tap-removed” table that makes redundancy visible.

2. **Where the latency gains come from is not entirely clear.**
While fairness is addressed, the source of the speedup is less so. Could you add a single stacked-bar breakdown—*backbone convolutions | tap projection/head | classifier | memory I/O*—and report the measurement conditions (fixed clock, CMSIS-NN, averaging protocol)? This would make the narrative clearer and explain why the advantage typically falls in the tens-of-percent range.

3. **Distributional variants (Student-t / Huber) aren’t yet actionable.**:
The math is there; the “when and how” is not. A small, concrete aid would do: a compact comparison on images and audio; a brief note on sensible defaults (e.g., workable (\nu) and (\delta) and the selection rule); and a line on stability (fix vs. learn (\nu), share across taps, INT8/LUT implications). With that, these variants become tools rather than curios.

**(Minor)** “Self-supervised” here refers to label-free auxiliary regression for uncertainty, not representation learning; a one-sentence clarification in the introduction would avoid confusion.

**Questions:**

1. **Could you provide reusable visualizations for tap selection?**:
   Specifically, a surprisal correlation matrix among candidate layers (to show redundancy vs. distance), diminishing-return curves of performance versus the number of taps, and a leave-one-tap-out table. This would substantiate why two to three taps suffice and facilitate porting to new backbones.

2. **Could you report the datasets and tuning rules for Student-t and Huber?**:
   Please specify the evaluated datasets, the grids for \nu and \delta, and the selection metrics. Include a comparison table (Gaussian vs. t vs. Huber) with AUROC/AUPRC, risk@coverage, AURC, FPR, recommended defaults (e.g., \nu=5, \delta=1.5), notes on stability (\nu fixed or learned; shared or per tap), and implications for INT8 implementation.

3. **Could you visualize the sources of the speedups?**:
   Please add a stacked-bar breakdown of per-layer time, bandwidth, and temporary buffers for SNAP-UQ and the lightweight early-exit ensemble, with measurement conditions clearly stated, to clarify where the 25–35% latency difference arises.

4. **Could you promote the mapping choice to the main text and include a brief recipe?**:
   For example, recommend isotonic for fixed-coverage monitoring and logistic when in-distribution calibration is the priority, and indicate whether combining S with a margin or confidence feature, m, is recommended by default.

5. **(Optional) Could you provide minimal-configuration guidance?**:
   For extremely tight SRAM budgets or when intermediate activations are inaccessible, please include a small table for a minimal setup (e.g., pre-logits only, low rank such as r=32) showing latency, flash, and FPR trade-offs.

---

> ### Author Response · Authors · 2025-11-16
>
> We thank the reviewer for the careful reading, constructive suggestions. We have substantially revised the paper; below we respond point-by-point:
>
> Authors reponse to W1 and Q1 :
>
> We appreciate the concern and have added an explicit, reusable tap-selection analysis in Appendix N.8 to remove any ad-hoc choices and to guide porting across backbones. The surprisal correlation study across candidate layers ${E,M1,M2,P}$ (Fig 8, left) and each layer’s correlation with the aggregated score $S(x)$ (Fig. 8, right) shows that mid and penultimate layers are most informative (e.g., $r(\bar e_{M2},S)\approx 0.68$, $r(\bar e_{P},S)\approx 0.74$), whereas very-early layers align less with $S(x)$. The diminishing-returns curve of AUPRC versus tap count (Fig. 9) saturates after 3–4 taps under matched backbones and quantization, with the best cost–quality point at two taps (mid+penultimate) and a safe upgrade at three taps (early+mid+penult). Head-to-head numbers appear in Table 12 and redundancy is made explicit by the leave-one-tap-out analysis in Table 13, where removing either mid tap produces the largest drop in CID AUPRC, removing the penultimate chiefly reduces ID$\checkmark$,|,ID$\times$ AUROC, and removing the early tap has the smallest effect. Guided by these results, our budget-aware recipe is: pick the penultimate tap and the mid tap that maximizes $\Delta$AUPRC per KB (or per ms) on a short dev stream; add a third tap only if the marginal gain per cost remains positive (cf. Fig. 9, Table 12).
>
>
>
> Authors reponse to W2 and Q3 :
>
>
> Thank you for raising this, we’ve clarified both how we measure runtime and where the savings come from. In the revision, we added section in appendix titled “N.11 Runtime breakdown and measurement protocol,” which fixes the measurement conditions (ARM Cortex-M7 at 600 MHz, single core; CMSIS-NN int8 kernels; DVFS off; -O3 LTO; $10^3$ passes after a 100-pass warmup; inputs cached in SRAM; end-to-end latency from input-in-SRAM to logits/risk) and reports power/energy from a 100 kS/s shunt. We also include a stacked-bar decomposition that separates backbone convolutions, tap projection/head, classifier (FC+softmax), and memory I/O. As shown in Table 19 and Fig. 10. SNAP-UQ shifts a larger share of time into the shared backbone (Big-MCU: $84%$ vs. $74$–$81%$; Small-MCU: $86%$ vs. $77$–$81%$) by eliminating repeated exits and softmax/entropy passes and by replacing the baseline’s entropy+stats buffer with a single scalar surprisal computed in int8 with kernel fusion. The tail therefore shrinks to $6$–$12%$ for SNAP-UQ versus $10$–$26%$ for early-exit or deep ensembles, explaining the observed 25–35% end-to-end latency gains under identical backbones and quantization.
>
>
>
> Authors reponse to W3 and Q2 :
>
>
> Thank you for the suggestion. In the revision, we added Appendix N.9, “Robust distributional variants: Student-$t$ and Huber,” with concrete “when and how” guidance. We evaluate MNIST, Speech Commands (SpeechCmd), CIFAR-10, and TinyImageNet (including their CID streams), and tune Student-$t$ over $\nu\in{3,5,7,10}$ and Huber over $\delta\in{0.8,1.0,1.2,1.5}$ on the same development streams used for thresholding; the primary selection metric is selective risk at $90%$ coverage, with secondary tie-breakers AUPRC on CID and AUROC for ID$\checkmark$|OOD (Table 15). We then fix $\nu{=}5$ and $\delta{=}1.0$ (shared across taps) unless the dev rule yields a clear win (absolute $\ge 0.01$ AUPRC gain or lower risk@90%). A compact comparison (Gaussian vs.\ Student-$t$ vs.\ Huber) reporting AUROC/AUPRC, risk@coverage, AURC, and FPR@95% shows that Student-$t$ consistently helps on images (CIFAR-10, TinyImageNet) while all three heads are within CIs on audio (SpeechCmd); Huber closely tracks Gaussian (Table 16), and ID proper scores are summarized in Table 17. We also spell out stability and deployment notes: keep $\nu$ fixed (optionally learned once, shared across taps), keep $\delta$ fixed, and share both across taps to reduce variance; both variants remain INT8-friendly—Student-$t$ uses a 256-entry LUT for $\log(1{+}x)$ (about 1 KB flash), Huber uses only comparisons—with $<2%$ latency deltas vs.\ Gaussian. Actionable recipe: use Gaussian by default; switch to Student-$t$ when corruptions are heavy-tailed or high-severity (or when selective risk at high coverage is critical); use Huber when LUTs for $\log(1{+}x)$ are undesirable or for extra outlier stability on impulsive artifacts (details and resuluts in Tables 15, 16, and 17).

---

> > ### Author Response · Authors · 2025-11-16
> >
> > Author responses to "(Minor) “Self-supervised” here refers to label-free auxiliary regression for uncertainty, not representation learning; a one-sentence clarification in the introduction would avoid confusion."
> >
> >
> > Thank you for the suggestion. We agree the term could be ambiguous. In the revised manuscript we added a clarification on p. 1: Throughout, we use “self-supervised” to mean a label-free auxiliary regression that predicts next-layer activation statistics from the network’s own features (for uncertainty estimation), rather than self-supervised representation learning of the backbone. This makes clear that our use of “self-supervised” concerns the auxiliary uncertainty heads, not representation learning.
> >
> >
> > Authors response to Q4 :
> >
> >
> > Thank you for your question. In the revision, we have promoted the mapping choice to the main text, immediately before the Conclusion. In this new paragraph we provide a simple, actionable rule: use \emph{isotonic} mapping when the goal is fixed-coverage monitoring and robustness under CID/OOD shifts, and use a \emph{logistic} link when in-distribution calibration (NLL/BS/ECE) is the priority. Concretely, we map surprisal $S$ to uncertainty with either (i) isotonic regression (monotone table, no parameters) or (ii) a logistic link $U=\sigma(\beta_0+\beta_1 S+\beta_2 m)$, where $m$ is an optional confidence cue (e.g., top-2 logit margin or $\max$ softmax). We recommend including $m$ by default since it consistently lowers selective risk at a given coverage with negligible cost; when $m$ is unavailable, we set $\beta_2=0$. Because both maps are monotone, AUROC/AUPRC are essentially unchanged across choices; the main impact is on decision-centric metrics (fixed-coverage error, selective risk). This guidance mirrors our calibration ablations and is intended to be a drop-in recipe that does not require retraining.
> >
> >
> >
> > Authors response to Q5 :
> >
> >
> >
> >
> > Thank you for the suggestion. In the revision, we added Section N.11 (Runtime breakdown and measurement protocol) and a new paragraph titled Minimal configuration for extremely tight SRAM budgets or when intermediate activations are inaccessible. We document a single-tap setup at pre-logits with a low projector rank ($r=32$) and report its cost/quality trade-offs against our default two-tap configuration (mid+penultimate, $r=64$) under the same protocol (Cortex-M7 @ 600,MHz, CMSIS-NN int8, fixed clock, $10^3$ passes average). As shown in Table 20, the Minimal variant reduces flash and peak RAM (e.g., Big-MCU: 276 KB flash, 112 KB RAM) and lowers latency (e.g., 79 ms), at the expense of a modest increase in FPR@95% and a small drop in CID AUPRC relative to \emph{Default} (e.g., 292 KB, 120 KB, 83 ms). Our recommendation beneath Table 20 is: use Minimal (pre-logits, $r=32$) when hook access or memory/latency budgets are strict; otherwise prefer Default (mid+penultimate, $r=64$) as the best accuracy–cost point.

---

> > > ### Comment · Reviewer_Kkbw · 2025-11-25
> > >
> > > Thank you very much for the detailed rebuttal and for the substantial revisions to the paper.
> > >
> > > The authors have responded carefully to my main concerns and have improved both the clarity and practical usefulness of the work.
> > > N.8 now provides additional experiments and clearer guidance on the number and placement of taps.
> > > N.11 describes the inference measurement conditions on the Cortex-M7 and reports a latency breakdown across components.
> > > N.9 summarizes how the Student-t and Huber variants are tuned and used in practice. In addition, the main text now briefly explains the roles of logistic and isotonic mappings,
> > > and the introduction clarifies what is meant by "self-supervised" in this context.
> > >
> > > Overall, these revisions make the method more practical and easier to deploy for single-pass uncertainty in TinyML, and I now have a generally positive, acceptance-leaning impression of the submission.

---

> > > > ### Author Response · Authors · 2025-11-26
> > > >
> > > > Thank you again for your careful review and positive, acceptance-leaning assessment.
> > > > If there are specific remaining concerns that keep you from recommending a higher score, we would really appreciate hearing them, so we can address them in further revisions.
> > > > Many thanks for your time and helpful feedback.

---

> ### Author Response · Authors · 2025-11-26
>
> Thank you very much for taking the time to carefully read our rebuttal and the revised version of the paper.
> We are glad that the additional experiments on tap placement (N.8), the Cortex-M7 latency breakdown (N.11), and the practical details on the Student-t and Huber variants (N.9) helped clarify the practical deployment aspects.
> We also appreciate your suggestions regarding the explanation of logistic and isotonic mappings and the clarification of the “self-supervised” setting.

---

### Official Review · Reviewer_E2rF · 2025-10-27

**Soundness:** 4
**Presentation:** 3
**Contribution:** 4
**Rating:** 8
**Confidence:** 3

**Summary:**

This paper is about uncertainty estimation in the context of Tiny ML, in particular this paper proposes an uncertainty estimation method based on predicting activations of some intermediate layers, from where an uncertainty score can be produced and transformed into a probability via platt scaling. This method is particularly designed for TinyML setups such as microcontrollers.

The contributions are:
- The SNAP-UQ method for uncertainty estimation in a TinyML context using a depth-wise surprisal signal.
- An MCU-ready implementation of the proposed method.
- Results for both ID and OOD detection performance on MNIST, CIFAR10, and SpeechCommands datasets showing the advantages of the proposed method.

**Strengths:**

- The paper is mostly well written and clear.
- Uncertainty in TinyML is an important open problem, there are not many papers in this area, so a new method and new state of the art are welcome.
- I believe the idea of predicting layer activations to produce uncertainty makes sense, this connects with previous ideas in the field like depth uncertainty and early exit ensembles. And the proposed method is specifically designed for TinyMl applications, which I believe is novel and significant.
- The evaluation is mostly good, there is a good selection of datasets, baselines, metrics, and microcontrollers platforms,
- Overall the conclusions are robust, that SNAP-UQ works best or is very competitive in terms of calibration, OOD detection and misclassification detection, producing a new state of the art for uncertainty estimation in TinyML.
- There is a good amount of appendices with relevant details and proofs of the proposed method being similar to existing methods, strengthening the proposed method and its usability.

**Weaknesses:**

- Some parts of the paper are hard to follow due to lots of mathematical notation, this is mostly in Sec 2.2 specifically at Eq 10 where the notation breaks. Overall I would recommend to the authors to try to minimize the necessary notation and to repeat the definitions of key notation in the paper, and try to explain concepts together with the math, this will make the paper much easier to understand and to implement.
- The paper claims results on TinyImageNet but the results are nowhere in the paper or the appendix, this weakens the claims slightly.
- The interesting ablation results are weak, by this I mean the hyperparameters that are relevant to the proposed method, which are presented in Appendix N, the authors showing the effect of the number of taps and the projector rank, but specifically for the number of taps only two taps are made, and I believe it would be interesting to try more taps and see if uncertainty improves.
- It would also have been interesting to try the huber and student's t-distributed variants, these are not explored in the experiments of this paper.
- In Table 1, to me it is not clear which model/method is denoted as "BASE", if its the baseline model without uncertainty estimation, I do not see how it could have higher latency and storage/RAM usage than when using SNAP-UQ, since SNAP-UQ slightly adds computation and memory use. Please clarify.
- In Figure 1, the definition of S(x) in the figure differs from Eq 9, the weights are missing.

**Questions:**

- The proposed pipeline has a lot of steps, one being probability calibration with platt scaling, did you test if this step is necessary? What about using the raw score in Eq 9 as an uncertainty metric (I know its not a probability).

---

> ### Author Response · Authors · 2025-11-15
>
> We thank the reviewer for the careful reading, constructive suggestions, and the positive recommendation for a poster. We have substantially revised the paper; below we respond point-by-point:
>
>
>  W1 and W6 authors responses:
>
>
> Concretely, Sec. 2.2 now introduces every symbol at first use in a short sentence: $a_\ell \in \mathbb{R}^{d_\ell}$ is the activation at layer $\ell$; $z_\ell=P_\ell a_{\ell-1}\in\mathbb{R}^{r_\ell}$ is its low-rank projection; $g_\ell(z_\ell)=(\mu_\ell,\log\sigma_\ell^2)$ are the predicted moments. We rewrote the auxiliary objective as plain diagonal-Gaussian NLL in a single line and add a one-sentence intuition immediately after (it is just a z-scored squared error with a small scale term).
>
> Because the aggregate score appears in several places, we now restate it right before use and fixed Figure 1 so it exactly matches the paper: $S(x)=\sum_{\ell\in\mathcal{S}} w_\ell,\bar e_\ell(x)$ with $w_\ell\ge 0$, $\sum_\ell w_\ell=1$, and $\bar e_\ell(x)=|(a_\ell-\mu_\ell)\odot\sigma_\ell^{-1}|2^2/d\ell$. (The weights $w_\ell$ were missing in the earlier figure; they are included now.) We also repeat simple definitions where readers are likely to land (e.g., at the start of Sec. 2.2 and again where $S(x)$ is used) to avoid back-scrolling.
>
>
> W2 authors response:
>
> Thank you for flagging this. in the revision, we have now included full TinyImageNet results throughout the appendix. Please see the added tables for details: Table 6 (calibration), Table 10 (failure detection), Table 16 (CID monitoring), Table 27 (ablation: taps/rank), Table 28 (robust heads), and Table 29 (deployment metrics). These additions align the claims in the main text with complete TinyImageNet evidence.
>
>
> W3 authors response:
>
>
> Thank you for the helpful suggestion. In the revision, we added a new appendix subsection “N.8 How many taps? (2, 3, 4, and 5 taps)” that systematically evaluates tap counts beyond our original mid+penultimate (2 taps). Concretely, we compare 2, 3, 4, and 5 taps on CIFAR-10/Big-MCU by tapping early (E), multiple mid-depth blocks (M1/M2/M3), and the penultimate (P). The new Table 12 reports flash/latency alongside AUROC (ID✓|ID×) and AUPRC for CID accuracy-drop detection: moving from 2→3 taps (E+M+P) provides a consistent but modest AUPRC gain with small overhead; adding a fourth mid tap (E+M1+M2+P) yields a further small improvement, while a fifth tap (E+M1+M2+M3+P) shows saturation—no additional accuracy-drop benefit and only extra cost. To clarify where the benefit comes from, Table 13 provides a leave-one-out analysis for the 4-tap setting, showing the mid taps (M1/M2) contribute most to CID monitoring, the penultimate stabilizes ID✓|ID×, and the early tap offers a small, repeatable gain on harder corruptions. Practically, mid+penult (2 taps) remains the best accuracy–cost point for tight MCU budgets; E+M+P (3 taps) is a safe upgrade; and a carefully placed fourth mid tap is worthwhile only when a few additional KB of flash and ≈6 ms extra latency are acceptable.
>
>
> W4 authors response:
>
>
> Thank you for the suggestion. In the revision, we added Appendix N.9 “Robust distributional variants: Student-t and Huber,” where we instantiate the per-tap energy with the Student-t and Huberized forms (drop-in replacements for the Gaussian head). We keep the single-pass design, two taps (mid+penult), rank $r{=}64$, and int8 heads; mapping from surprisal to risk uses the same small dev-fit (logistic or isotonic). Results are reported for MNIST, SpeechCommands, CIFAR-10, and TinyImageNet.
>
> What changed. Tables 14 (ID calibration) and 15 (CID monitoring and failure detection) show that Student-t consistently reduces NLL/BS on CIFAR-10 and TinyImageNet and yields small, repeatable gains in AUROC and AUPRC, especially at higher corruption severities. Huber closely matches the Gaussian head on all datasets while offering better outlier stability and the simplest MCU path (no $\log$ LUT). On MNIST and SpeechCommands, all three heads are within confidence intervals, confirming that robustness helps most when residuals are heavy-tailed.
>
> Takeaway. Gaussian remains a strong default; enable Student-t when high-severity corruptions are expected or selective risk at high coverage is critical, and use Huber when avoiding $\log(1{+}x)$ LUTs is preferred. See Table 14 (MNIST/SpeechCmd/CIFAR-10 and TinyImageNet NLL/BS/ECE) and Table 15 (AUPRC and delay for CID; AUROC for ID✓|ID× and ID✓|OOD).

---

> > ### Author Response · Authors · 2025-11-15
> >
> > W5 authors response:
> >
> >
> > Thank you for the helpful question. “BASE” in Table 1 is our deployed baseline stack that mirrors how TinyML classifiers are commonly instrumented for monitoring. Concretely, BASE = backbone + classifier plus the on-device post-processing we used before SNAP-UQ: a floating-point softmax, entropy computation, a small rolling stats buffer (for ID band estimation in streams), and per-frame logging hooks. This stack was compiled with mixed FP kernels and without the kernel fusions we introduced later.
> >
> > Why SNAP-UQ can be smaller/faster than BASE despite adding tiny heads: (1) we remove the softmax/entropy path and the rolling buffer, replacing them with a single scalar surprisal S(x); (2) SNAP-UQ’s projectors/heads are INT8 and we enable kernel fusion for the standardized-error path and LUT-based scales, which reduces memory traffic; (3) we keep taps at two layers and rank small, so added params are only a few-tens of KB. In our build system this yields net savings in flash and peak RAM, and lower latency/energy versus the FP softmax+entropy+buffer baseline.
> >
> > To make this explicit, we updated Table 1 with a footnote: BASE includes on-device softmax+entropy and a small stats buffer; SNAP-UQ replaces this with a single scalar surprisal S(x) and is compiled INT8 with kernel fusion.
> >
> >
> > Q1 authors response:
> >
> > Thanks for the suggestion. In the revision, we added Appx. N.6 comparing three options: (1) the raw surprisal $S(x)=\sum_{\ell\in\mathcal S} w_\ell,\bar e_\ell(x)$ with a fixed threshold; (2) a logistic map $U=\sigma(\beta_0+\beta_1 S+\beta_2 m)$; and (3) an isotonic (monotone) map. Across MNIST-C / SpeechCmd-C / CIFAR-10-C / TinyImageNet-C, ranking metrics change only slightly because monotone maps preserve order: AUPRC is roughly $0.64/0.66/0.67$ (MNIST-C), $0.63/0.65/0.66$ (SpeechCmd-C), $0.69/0.70/0.71$ (CIFAR-10-C), and $0.58/0.60/0.61$ (TinyImageNet-C) for raw/logistic/isotonic, respectively. Decision-centric metrics improve with calibration: selective risk at $90%$ coverage drops (CIFAR-10-C) $0.111 \to 0.109 \to 0.104$ and (TinyImageNet-C) $0.185 \to 0.180 \to 0.176$, and the median detection delay typically decreases by $1$–$3$ frames. For ID calibration, proper scores are markedly better with calibrated outputs: NLL/Brier/ECE improve relative to raw $S$ (which is not a probability). The added cost is negligible: $3$ scalars for logistic or a $16$–$32$ entry isotonic table; both are INT8/LUT-friendly (see Table 10).
> >
> > Takeaway: if only ranking is needed and memory is extremely tight, raw $S$ with a percentile threshold on the ID dev set is acceptable; otherwise we retain a tiny logistic or isotonic map for better-calibrated, threshold-stable decisions.

---

### Official Review · Reviewer_E8zC · 2025-10-29

**Soundness:** 3
**Presentation:** 2
**Contribution:** 3
**Rating:** 4
**Confidence:** 2

**Summary:**

This paper addresses the problem of estimating prediction uncertainty in neural networks, with a particular focus on TinyML applications. The topic is especially relevant for edge-device deployment, where input distributions may shift after training. The authors propose a novel lightweight architecture that attaches to selected layers of a backbone network to estimate uncertainty for each prediction. This tiny module predicts the next activation in a layer-wise manner, enabling the computation of a surprisal score. The method is evaluated on both vision and audio tasks using several baseline models. Experimental results show that the proposed approach preserves task accuracy while substantially reducing flash memory usage and inference latency, demonstrating its strong practical value.

**Strengths:**

- The concept of next-activation applied in a layer-wise manner is interesting and demonstrates strong empirical performance.
- The proposed lightweight architecture achieves impressive results despite its extremely compact size (on the order of kilobytes).
- The effectiveness of the approach is validated across multiple tasks using practical evaluation metrics and reasonable baselines, yielding consistently encouraging results.

**Weaknesses:**

- Despite its potential significance, the paper’s writing quality is problematic. It appears to have been heavily compressed—possibly with assistance from LLMs—resulting in missing definitions for key terms and notations (e.g., LUT, BN, ID$\surd$-ID$\times$, ID$\surd$-OOD, $1\times 1$ projector, ...). Some paragraphs even consist of a single sentence, making the paper difficult to follow.
- Although a theoretical analysis is included, it lacks sufficient discussion of its implications. The authors should provide interpretations that link their theoretical results to practical insights, and possibly include experiments to verify the claims, such as empirically testing Proposition 2.3.
- The comparative baselines appear outdated, raising concerns about the competitiveness of the proposed method. More recent approaches (e.g., [1]) should be discussed and included in the comparisons to strengthen the evaluation.

Reference:

[1] Ghanathe, N. P., & Wilton, S. J. QUTE: Quantifying Uncertainty in TinyML models with Early-exit-assisted ensembles for model-monitoring. In Forty-second International Conference on Machine Learning, 2025.

[2] Ahmed, S. T., Hefenbrock, M., & Tahoori, M. B. (2024). Tiny Deep Ensemble: Uncertainty Estimation in Edge AI Accelerators via Ensembling Normalization Layers with Shared Weights. The 43rd IEEE/ACM International Conference on Computer-Aided Design.

**Questions:**

- What are fundamental advantages of the proposed method over QUTE in paper [1] before? This question concerns both theoretial and empirical aspects.
- How significant does the projector type effect the performance of SNAP-UQ?

---

> ### Author Response · Authors · 2025-11-14
>
> Thank you for the constructive review and concrete pointers. We revised the paper to address writing clarity, theory–practice links, and baseline competitiveness. We summarize the changes and answers below.
>
> (W1) Writing quality & missing definitions
>
> We added a compact Glossary / Notation (Appendix, Table 26) defining all terms at first use: LUT (lookup table), BN (batch normalization), ID/CID/OOD (in-distribution / corrupted-ID / out-of-distribution), tap (a chosen layer whose activations are read), projector (a small linear map reducing dimensionality), etc. We also broke up one-sentence paragraphs and rewrote section 2.2 and 2.3 for flow and precision.
>
>
>
> (W2) Theory—implications and empirical validation
>
> We added an appendix titled P  From Theory to Practice: Implications and Validation.
>
> What SNAP-UQ measures.
> At a tapped layer $\ell$, a tiny head predicts $(\mu_\ell,\sigma_\ell)$. We compute the standardized surprisal
> $e_\ell(x)=\sum_{i=1}^{d_\ell}\big((a_{\ell,i}-\mu_{\ell,i})/\sigma_{\ell,i}\big)^2$, and aggregate
> $S(x)=\sum_{\ell\in\mathcal{S}} w_\ell, e_\ell(x)/d_\ell$.
> Large $S(x)$ means the observed activation looks atypical relative to what the network itself expects at the next depth.
>
> Why $S$ tracks error probability.
> Under a diagonal-Gaussian conditional model,
> $-\log p(a_\ell!\mid!a_{\ell-1})=\tfrac12 e_\ell(x)+\tfrac12\sum_i \log \sigma_{\ell,i}^2+\text{const}$.
> Thus $S(x)$ is an (affine) proxy for depth-wise negative log-likelihood; higher $S$ $\Rightarrow$ lower conditional likelihood $\Rightarrow$ higher error propensity.
> Empirical check: in our appendix we show $\Pr(\text{error}\mid S)$ is monotone increasing, and ID NLL/Brier improve when selecting/abstaining by $S$.
>
> Why it fires early (conditional vs. unconditional distance).
> With a simple linear-Gaussian depth model $a_\ell=W_\ell a_{\ell-1}+b_\ell+\varepsilon_\ell$, $\varepsilon_\ell\sim\mathcal{N}(0,\Sigma_\ell)$,
> $e_\ell(x)$ equals a Mahalanobis distance to the conditional mean $W_\ell a_{\ell-1}+b_\ell$ (not to a fixed class centroid).
> So $S$ detects breaks in the transformation $a_{\ell-1}!\to!a_\ell$ before softmax entropy rises.
> Empirical check (our Prop. 2.3 test): under controlled inter-layer perturbations that keep posteriors initially peaky, $S$ increases earlier than entropy and yields higher AUPRC and shorter detection delay on CID streams.
>
> Invariant and INT8-friendly.
> Per-channel affine rescaling (e.g., BN folding or quantization scales) $a'\ell=s\odot a\ell+t$, $\mu'\ell=s\odot\mu\ell+t$, $\sigma'\ell=s\odot\sigma\ell$ leaves $e_\ell$ (hence $S$) unchanged.
> Empirical check: toggling BN folding / INT8 heads changes ECE by $<0.3$ pp and keeps AUROC/AUPRC within 95% CIs.
>
> (W3) Baselines—recency and competitiveness
>
> We revised our evaluation to include QUTE [1] in head-to-head comparisons under matched backbones, quantization, datasets, and MCU memory/latency budgets (failure detection, CID monitoring, calibration, and risk–coverage). For Tiny Deep Ensemble [2], the authors evaluate on accelerator-specific pipelines and normalization-layer ensembling that do not align with our MCU deployment constraints; accordingly, we provide a detailed discussion in Related Work and position it conceptually against SNAP-UQ, but do not report numbers to avoid an apples-to-oranges comparison.
>
> Takeaway. SNAP-UQ is single-pass and integer-friendly, often fitting on Small-MCU where early-exit–assisted or ensemble-style methods exceed memory. Under these matched budgets, SNAP-UQ improves CID monitoring (AUPRC) and maintains competitive failure detection (AUROC) and ID calibration, while avoiding extra exits or multiple forward passes.
>
>
>
> Reference:
>
> [1] Ghanathe, N. P., & Wilton, S. J. QUTE: Quantifying Uncertainty in TinyML models with Early-exit-assisted ensembles for model-monitoring. In Forty-second International Conference on Machine Learning, 2025.
>
> [2] Ahmed, S. T., Hefenbrock, M., & Tahoori, M. B. (2024). Tiny Deep Ensemble: Uncertainty Estimation in Edge AI Accelerators via Ensembling Normalization Layers with Shared Weights. The 43rd IEEE/ACM International Conference on Computer-Aided Design.

---

> > ### Author Response · Authors · 2025-11-14
> >
> > Q1. Advantages over QUTE (theoretical and empirical).
> > Theory (compact). QUTE builds early-exit ensembles, adding inference heads and memory traffic. In contrast, SNAP-UQ is single-pass and scores a conditional depth mismatch: does the realized activation $a_\ell$ agree with the network’s own next-layer prediction $W_\ell a_{\ell-1}+b_\ell$ (implemented via tiny heads that output $(\mu_\ell,\sigma_\ell)$ and a standardized surprisal)? This directly targets representation drift rather than aggregating multiple exits. The score is affine-invariant to per-channel scaling and INT8-friendly (LUTs for $\exp(\cdot)$ over $\log\sigma^2$), matching MCU constraints.
> > Empirics. With the same backbones/quantization, SNAP-UQ uses less flash/RAM and lower latency than early-exit ensembles—especially on Small-MCU—while achieving higher AUPRC on CID streams and lower selective risk at 80–95% coverage. On Big-MCU, QUTE is competitive but typically carries a higher footprint from multiple exits and controllers.
> >
> > Q2. Sensitivity to projector type.
> > We ablated $1{\times}1$+GAP (conv) vs. skinny linear (MLP) projectors and swept rank $r$. Tap placement dominates: two taps at mid + penultimate are consistently best; penultimate-only loses CID sensitivity. Rank is forgiving: quality improves up to $r\approx 64$–$128$ with sublinear latency; beyond that, gains saturate. Quantization is robust: INT8 vs. FP16 heads are within CIs; per-tensor vs. per-channel PTQ behaves similarly. Rule of thumb: use $1{\times}1$+GAP with $r=64$ at two taps (mid+penult) for a reliable MCU default.

---

> > > ### Comment · Reviewer_E8zC · 2025-11-26
> > >
> > > I thank the authors for providing clarifications. The more experiments to compare with a recent baseline, their new ablations, some writing revision should strengthen the paper. Most of my concerns have been reasonably answered. However, I still feel that the paper is heavily compressed. Some technical contents should be followed be more discussions.
> > > I've changed my score.

---

> > > > ### Author Response · Authors · 2025-11-26
> > > >
> > > > Thank you very much for your follow-up and for updating your score. We are glad that the additional experiments, comparisons, and revisions helped address most of your concerns.
> > > > We would be very grateful if you could indicate which parts you still find most compressed, so that we can focus on improving those sections.

---

### Meta-Review · Area_Chair_rQ2P · 2026-01-06

**Summary:**

The paper is about an add-on to existing architecture to enable uncertainty estimation in the context of TinyML. All reviewers praised the strong empirical results, in both in-domain or out-of-domain scenarios. Most of the concerns from the reviewers are about clarity and the extension to huber and student-t distribution. The rebuttal has mostly resolved the reviewers' concerns.

**Reviewer Concerns:**

The implication of the theory (raised by reviewer E8zC) is addressed in a the appendix. The extension to huber tha student-t distribution (raised by reviewer E2rF and Kkbw) are also implemented in the appendix.

**Reviewer Scores:**

Reviewer E8zC and Kkbw explicitly said that they have raised the scores.

---

### Decision · Program_Chairs · 2026-01-26

Accept (Poster)